# Variance-Reduced Forward-Reflected-Backward Splitting Methods for Nonmonotone Generalized Equations

**Quoc Tran-Dinh** [1]

## Abstract

We develop two novel stochastic variance-reduction methods to approximate solutions of a class of nonmonotone [generalized] equations. Our algorithms leverage a new combination of ideas from the forward-reflected-backward splitting method and a class of unbiased variance-reduced estimators. We construct two new stochastic estimators within this class, inspired by the well-known SVRG and SAGA estimators. These estimators significantly differ from existing approaches used in minimax and variational inequality problems. By appropriately choosing parameters, both algorithms achieve state-of-the-art oracle complexity of $\mathcal{O}(n + n^{2/3}\epsilon^{-2})$ for obtaining an $\epsilon$-solution in terms of the operator residual norm for a class of nonmonotone problems, where $n$ is the number of summands and $\epsilon$ signifies the desired accuracy. This complexity aligns with the best-known results in SVRG and SAGA methods for stochastic nonconvex optimization. We test our algorithms on some numerical examples and compare them with existing methods. The results demonstrate promising improvements offered by the new methods compared to their competitors.

## 1. Introduction

**[Non]linear equations and inclusions** are cornerstones of computational mathematics, finding applications in diverse fields like engineering, mechanics, economics, statistics, optimization, and machine learning, see, e.g., (Bauschke & Combettes, 2017; Burachik & Iusem, 2008; Facchinei & Pang, 2003; Phelps, 2009; Ryu & Yin, 2022; Ryu & Boyd, 2016). These problems, known as *generalized equations* (Rockafellar & Wets, 1997), are equivalent to *fixed-point problems*. The recent revolution in modern machine learning and robust optimization has brought renewed interest to generalized equations and their special case: minimax problem. They serve as powerful tools for handling Nash's equilibria and minimax models in generative adversarial nets, adversarial training, and robust learning, see (Arjovsky et al., 2017; Goodfellow et al., 2014; Madry et al., 2018; Namkoong & Duchi, 2016). Notably, most problems arising from these applications are nonmonotone, nonsmooth, and large-scale. This paper develops new and simple stochastic algorithms with variance reduction for solving this class of problems, equipped with rigorous theoretical guarantees.

### 1.1. Nonmonotone finite-sum generalized equations

The central problem we study in this paper is the following [possibly nonmonotone] *generalized equation* (also known as *composite inclusion*) (Rockafellar & Wets, 1997):

$$\text{Find } x^\star \in \mathbb{R}^p \text{ such that: } 0 \in Gx^\star + Tx^\star, \quad \text{(NI)}$$

where $G : \mathbb{R}^p \to \mathbb{R}^p$ is a single-valued operator, possibly nonlinear, and $T : \mathbb{R}^p \rightrightarrows 2^{\mathbb{R}^2}$ is a multivalued mapping from $\mathbb{R}^p$ to $2^{\mathbb{R}^p}$ (the set of all subsets of $\mathbb{R}^p$). In addition, we assume that $G$ is given in the following *large finite-sum*:

$$Gx := \frac{1}{n}\sum_{i=1}^{n}G_ix, \quad \text{(1)}$$

where $G_i : \mathbb{R}^p \to \mathbb{R}^p$ are given operators for all $i \in [n] := \{1, 2, \cdots, n\}$ and $n \gg 1$. This structure often arises from statistical learning, [generative] machine learning, networks, distributed systems, and data science. For simplicity of notation, we denote $\Psi := G + T$ and $\mathrm{dom}(\Psi) := \mathrm{dom}(G) \cap \mathrm{dom}(T)$, where $\mathrm{dom}(R)$ is the domain of $R$.

We highlight that the methods developed in this paper can be straightforwardly extended to tackle $Gx = \mathbb{E}_{\xi\sim\mathbb{P}}\big[\mathbf{G}(x,\xi)\big]$ as the expectation of a stochastic operator $\mathbf{G}$ involving a random vector $\xi$ defined on a probability space $(\Omega, \mathbb{P}, \Sigma)$.

### 1.2. Equivalent forms and special cases

The model (NI) covers many fundamental problems in optimization and related fields, including the following ones.

(a) **[Non]linear equation.** If $T = 0$, then (NI) reduces to the following *[non]linear equation*:

$$\text{Find } x^\star \in \mathrm{dom}(G) \text{ such that: } Gx^\star = 0. \quad \text{(NE)}$$

---

[1]Department of Statistics and Operations Research, The University of North Carolina at Chapel Hill, Chapel Hill, NC 27599, USA. Correspondence to: Quoc Tran-Dinh <quoctd@email.unc.edu>.

*Proceedings of the 42nd International Conference on Machine Learning*, Vancouver, Canada. PMLR 267, 2025. Copyright 2025 by the author(s).

Both (NI) and (NE) are also called *root-finding problems*. Clearly, (NE) is a special case of (NI). However, under appropriate assumptions on $G$ and/or $T$ (e.g., using the resolvent of $T$), one can also transform (NI) to (NE). Let $\text{zer}(\Psi) := \{x^\star \in \text{dom}(\Psi) : 0 \in \Psi x^\star\}$ and $\text{zer}(G) := \{x^\star \in \text{dom}(G) : Gx^\star = 0\}$ be the solution sets of (NI) and (NE), respectively, which are assumed to be nonempty.

(b) **Variational inequality problem (VIP).** If $T = \mathcal{N}_\mathcal{X}$, the normal cone of a nonempty, closed, and convex set $\mathcal{X}$ in $\mathbb{R}^p$, then (NI) reduces to the following VIP:

Find $x^\star \in \mathcal{X}$ such that: $\langle Gx^\star, x - x^\star \rangle \geq 0, \forall x \in \mathcal{X}$. (VIP)

If $T = \partial g$, the subdifferential of a convex function $g$, then (NI) reduces to a mixed VIP, denoted by MVIP. Both VIP and MVIP cover many problems in practice, including minimax problems and Nash's equilibria, see, e.g., (Burachik & Iusem, 2008; Facchinei & Pang, 2003; Phelps, 2009).

(c) **Minimax optimization.** Another important special case of (NI) (or MVIP) is the following minimax optimization (or saddle-point) problem, which has found various applications in machine learning and robust optimization:

$$\min_{u \in \mathbb{R}^{p_1}} \max_{v \in \mathbb{R}^{p_2}} \left\{ \mathcal{L}(u, v) := \varphi(u) + \mathcal{H}(u, v) - \psi(v) \right\}, \text{ (SP)}$$

where $\mathcal{H} : \mathbb{R}^{p_1} \times \mathbb{R}^{p_2} \to \mathbb{R}$ is a smooth function, and $\varphi$ and $\psi$ are proper, closed, and convex. Let us define $x := [u, v] \in \mathbb{R}^p$ as the concatenation of $u$ and $v$ with $p := p_1 + p_2$, $Gx := [\nabla_u \mathcal{H}(u, v), -\nabla_v \mathcal{H}(u, v)]$, and $Tx := [\partial \varphi(u), \partial \psi(v)]$. Then, the optimality condition of (SP) is written in the form of (NI). Since (VIP), and in particular, (SP) are special cases of (NI), our algorithms for (NI) in the sequel can be specified to solve these problems.

(d) **Fixed-point problem.** Problem (NE) is equivalent to the following fixed-point problem:

$$\text{Find } x^\star \in \text{dom}(F) \text{ such that: } x^\star = Fx^\star, \quad \text{(FP)}$$

where $F := \mathbb{I} - \lambda G$ with $\mathbb{I}$ being the identity operator and $\lambda > 0$. Since (FP) is equivalent to (NE), our algorithms for (NE) from this paper can also be applied to solve (FP).

### 1.3. Motivation
Our work is mainly motivated by the following aspects.

(i) *Recent applications.* Both (NE) and (NI) cover the minimax problem (SP) as a special case. This minimax problem, especially in nonconvex-nonconcave settings, has recently gained its popularity as it provides a powerful tool to model applications in generative adversarial networks (Arjovsky et al., 2017; Goodfellow et al., 2014), robust and distributionally robust optimization (Ben-Tal et al., 2009; Bertsimas & Caramanis, 2011; Levy et al., 2020), adversarial training (Madry et al., 2018), online optimization (Bhatia & Sridharan, 2020), and reinforcement learning (Azar et al., 2017; Zhang et al., 2021). Our work is motivated by those applications.

(ii) *Optimality certification.* Existing stochastic methods often target special cases of (NI) such as (NE) and (VIP). In addition, these methods frequently rely on a monotonicity assumption, which excludes many problems of current interest, e.g., (Alacaoglu et al., 2023; Alacaoglu & Malitsky, 2022; Beznosikov et al., 2023; Gorbunov et al., 2022a; Loizou et al., 2021). Furthermore, existing methods analyze convergence based on a [duality] **gap function** (Facchinei & Pang, 2003) or a **restricted gap function** (Nesterov, 2007). As discussed in (Cai et al., 2024; Diakonikolas, 2020), these metrics have limitations, particularly in nonmonotone settings. It is important to note that standard gap functions are not applicable to our settings due to Assumption 1.4. Regarding oracle complexity, several works, e.g., (Alacaoglu & Malitsky, 2022; Beznosikov et al., 2023; Gorbunov et al., 2022a; Loizou et al., 2021) claim an oracle complexity of $\mathcal{O}(n + \sqrt{n}\epsilon^{-1})$ to attain an $\epsilon$-solution, but this is measured using a restricted gap function. Again, as highlighted in (Cai et al., 2024; Diakonikolas, 2020), this certification does not translate to the operator residual norm and is inapplicable to nonmonotone settings. Therefore, a direct comparison between our results and these previous works is challenging due to these methodological discrepancies (see Table 1).

(iii) *New and simple algorithms.* Various existing stochastic methods for solving (VIP) and (NI) rely on established techniques. These include mirror-prox/averaging and extragradient-type schemes combined with the classic Robbin-Monro stochastic approximation (Robbins & Monro, 1951) (e.g., (Cui & Shanbhag, 2021; Iusem et al., 2017; Juditsky et al., 2011; Kannan & Shanbhag, 2019; Kotsalis et al., 2022; Yousefian et al., 2018)). Some approaches utilize increasing mini-batch sizes for variance reduction (e.g., (Iusem et al., 2017; Pethick et al., 2023)). Recent works have explored alternative variance-reduced methods for (NI) and its special cases (e.g., (Alacaoglu et al., 2023; Alacaoglu & Malitsky, 2022; Bot et al., 2019; Cai et al., 2022; Davis, 2022)). However, these methods primarily adapt existing optimization estimators to approximate the operator $G$ without significant differences. Our approach departs from directly approximating $G$. Instead, we construct an intermediate quantity $S_\gamma^k := Gx^k - \gamma Gx^{k-1}$ as a linear combination of two consecutive evaluations of $G$ (i.e. $Gx^k$ and $Gx^{k-1}$). We then develop stochastic variance-reduced estimators specifically for $S_\gamma^k$. Note that $S_\gamma^k$ alone is not new, but our idea of using it in stochastic methods is new. This idea allows us to design new and simple algorithms with a single loop for both (NE) and (NI) where the state-of-the-art oracle complexity is achieved (*cf.* Sections 3 and 4).

### 1.4. Basic assumptions
We tackle both (NE) and (NI) covered by the following basic assumptions (see (Bauschke & Combettes, 2017) for terminologies and concepts used in these assumptions).

**Assumption 1.1.** [*Well-definedness*] $\text{zer}(\Psi)$ of (NI) and

$\text{zer}(G)$ of (NE) are nonempty.

**Assumption 1.2.** [*Maximal monotonicity of $T$*] $T$ in (NI) is maximally monotone on $\text{dom}(T)$.

**Assumption 1.3.** [*Lipschitz continuity of $G$*] $G$ in (1) is $L$-averaged Lipschitz continuous, i.e. $\forall x, y \in \text{dom}(G)$:

$$\tfrac{1}{n}\sum_{i=1}^n \|G_i x - G_i y\|^2 \le L^2 \|x - y\|^2. \qquad (2)$$

**Assumption 1.4.** [*Weak-Minty solution*] There exist a solution $x^\star \in \text{zer}(\Psi)$ and $\kappa \ge 0$ such that $\langle Gx + v, x - x^\star \rangle \ge -\kappa\|Gx + v\|^2$ for all $x \in \text{dom}(\Psi)$ and $v \in Tx$.

While Assumption 1.1 is basic, Assumption 1.2 guarantees the single-valued and well-definiteness of the resolvent $J_T$ of $T$. In fact, this assumption can be relaxed to some classes of nonmonotone operators $T$, but we omit this extension. The $L$-averaged Lipschitz continuity (2) is standard and has been used in most deterministic, randomized, and stochastic methods. It is slightly stronger that the $L$-Lipschitz continuity of the sum $G$. The star-co-hypomonotonicity in Assumption 1.4 is significantly different from the star-strong monotonicity used in, e.g., (Kotsalis et al., 2022). It covers a class of nonmonotone operators $G$ (see Supp. Doc. A.2 for a concrete example). It is also weaker than the co-hypomotonicity, used, e.g., in (Cai et al., 2024).

### 1.5. Contribution and related work
Our primary goal is to develop a class of variance-reduction methods to solve both (NE) and (NI), their special cases such as (VIP) and (SP), and equivalent problems, like (FP).

**Our contribution.** Our main contribution consists of:

(a) We exploit the variable $S_\gamma^k$ in (FRQ) and introduce a class of unbiased variance-reduced estimators $\widetilde{S}_\gamma^k$ for $S_\gamma^k$, not for $G$, covered by Definition 2.1.

(b) We construct two instances of $\widetilde{S}_\gamma^k$ by leveraging the SVRG (Johnson & Zhang, 2013) and SAGA (Defazio et al., 2014) estimators, respectively that fulfill our Definition 2.1. These estimators are also of independent interest, and can be applied to develop other methods.

(c) We develop a variance-reduced forward-reflected-type method (VFR) to solve (NE) required $\mathcal{O}(n + n^{2/3}\epsilon^{-2})$ evaluations of $G_i$ to obtain an $\epsilon$-solution.

(d) We design a novel stochastic variance-reduced forward-reflected-backward splitting method (VFRBS) to solve (NI), also required $\mathcal{O}(n + n^{2/3}\epsilon^{-2})$ evaluations of $G_i$.

Table 1 below compares our work and some existing single-loop variance-reduction methods, but in either the co-coercive or monotone settings. Now, let us highlight the following points of our contribution. First, our intermediate quantity $S_\gamma^k$ can be viewed as a generalization of the forward-reflected-backward splitting (FRBS) operator (Malitsky & Tam, 2020) or an optimistic gradient operator (Daskalakis et al., 2018) used in the literature. However, the chosen

range $\gamma \in (1/2, 1)$ excludes these classical methods from recovering as special cases of $S_\gamma^k$. Second, since our SVRG and SAGA estimators are designed specifically for $S_\gamma^k$, they differ from existing estimators in the literature, including recent works (Alacaoglu et al., 2023; Alacaoglu & Malitsky, 2022; Bot et al., 2019). Third, both proposed algorithms are single-loop and straightforward to implement. Fourth, our algorithm for nonlinear inclusions (NI) significantly differs from existing methods, including deterministic ones, due to the additional term $\gamma^{-1}(2\gamma - 1)(y^k - x^k)$. For a comprehensive survey of deterministic methods, we refer to (Tran-Dinh, 2023). Fifth, our oracle complexity estimates rely on the metric $\mathbb{E}[\|Gx^k\|^2]$ or $\mathbb{E}[\|Gx^k + v^k\|^2]$ for $v^k \in Tx^k$, commonly used in nonmonotone settings. Unlike the monotone case, this metric cannot be directly converted to a gap function, see, e.g., (Alacaoglu et al., 2023; Alacaoglu & Malitsky, 2022). Our complexity bounds match the best known in stochastic nonconvex optimization using SAGA or SVRG without additional enhancements, e.g., utilizing a nested technique as in (Zhou et al., 2018).

**Related work.** Since both theory and solution methods for solving (NE) and (NI) are ubiquitous, see, e.g., (Bauschke & Combettes, 2017; Burachik & Iusem, 2008; Facchinei & Pang, 2003; Phelps, 2009; Ryu & Yin, 2022; Ryu & Boyd, 2016), especially under the monotonicity, we only highlight the most recent related works (see more in Supp. Doc. A).

(i) *Weak-Minty solution.* Assumption 1.4 is known as a weak-Minty solution of (NI) (in particular, of (NE)), which has been widely used in recent works, e.g., (Böhm, 2022; Diakonikolas et al., 2021; Lee & Kim, 2021; Pethick et al., 2022; Tran-Dinh, 2023a) for deterministic methods and, e.g., (Lee & Kim, 2021; Pethick et al., 2023; Tran-Dinh & Luo, 2025) for stochastic methods. This weak-Minty solution condition is weaker than the co-hypomonotonicity (Bauschke et al., 2020), which was used earlier in proximal-point methods (Combettes & Pennanen, 2004). Diakonikolas *et al.* exploited this condition to develop an extragradient variant (called EG+) to solve (NE). Following up works include (Böhm, 2022; Cai & Zheng, 2023; Luo & Tran-Dinh, 2022; Pethick et al., 2022; Tran-Dinh, 2023a). A recent survey in (Tran-Dinh, 2023) provides several deterministic methods that rely on this condition. This assumption covers a class of nonmonotone operators $G$ or $G + T$.

(ii) *Stochastic approximation methods.* Stochastic methods for both (NE) and (NI) and their special cases have been extensively developed, see, e.g., (Juditsky et al., 2011; Kotsalis et al., 2022; Pethick et al., 2023). Several methods exploited mirror-prox and averaging techniques such as (Juditsky et al., 2011; Kotsalis et al., 2022), while others relied on projection or extragradient schemes, e.g., (Cui & Shanbhag, 2021; Iusem et al., 2017; Kannan & Shanbhag, 2019; Pethick et al., 2023; Yousefian et al., 2018). Many of these algorithms use standard Robbin-Monro stochastic

Table 1: Comparison of recent existing single-loop variance-reduction methods and our algorithms

| Papers | Problem | Assumptions | Estimators | Residual Rates | Oracle Complexity |
|---|---|---|---|---|---|
| (Davis, 2022) | (NE) and (NI) | co-coercive/SQM | SVRG & SAGA | linear | $\mathcal{O}\big((L/\mu)\log(\epsilon^{-1})\big)$ |
| (Tran-Dinh, 2024) | (NE) and (NI) | co-coercive | a class | $\mathcal{O}(1/k^2)$ | $\mathcal{O}(n + n^{2/3}\epsilon^{-1})$ |
| (Cai et al., 2024) | (NE) and (NI) | co-coercive | SARAH | $\mathcal{O}(1/k^2)$ | $\mathcal{O}(n + n^{1/2}\log(n)\epsilon^{-1})$ |
| (Alacaoglu & Malitsky, 2022) | (VIP) | monotone | SVRG | ✗ | $\mathcal{O}(n + n^{1/2}\epsilon^{-1})$ |
| (Alacaoglu et al., 2023) | (VIP) | monotone | SVRG | ✗ | $\mathcal{O}(n\epsilon^{-1})$ |
| **Ours** | (NE) and (NI) | weak Minity | a class | $\mathcal{O}(1/k)$ | $\mathcal{O}(n + n^{2/3}\epsilon^{-2})$ |

**Notes: SQM** means "strong quasi-monotonicity"; **Residual Rate** is the convergence rate on $\mathbb{E}\big[\|Gx^k\|^2\big]$ or $\mathbb{E}\big[\|Gx^k + v^k\|^2\big]$ for $v^k \in Tx^k$; and **a class** is a class of variance-reduced estimators. The complexity of (Alacaoglu & Malitsky, 2022) and (Alacaoglu et al., 2023) marked by magenta is on a gap function, a different metric than in other works in Table 1. Thus, it is unclear how to compare them.

approximation with fixed or increasing batch sizes. Some other works generalized the analysis to a general class of algorithms such as (Beznosikov et al., 2023; Gorbunov et al., 2022a; Loizou et al., 2021) covering both standard stochastic approximation and variance reduction algorithms.

(iii) *Variance-reduction methods.* Variance-reduction techniques have been broadly explored in optimization, where many estimators were proposed, including SAGA (Defazio et al., 2014), SVRG (Johnson & Zhang, 2013), SARAH (Nguyen et al., 2017), and Hybrid-SGD (Tran-Dinh et al., 2019; 2022), and STORM (Cutkosky & Orabona, 2019). Researchers have adopted these estimators to develop methods for (NE) and (NI). For example, (Davis, 2022) proposed a SAGA-type methods for (NE) under a [quasi]-strong monotonicity. The authors in (Alacaoglu et al., 2023; Alacaoglu & Malitsky, 2022) employed SVRG estimators and developed methods for (VIP). Other works can be found in (Bot et al., 2019; Carmon et al., 2019; Chavdarova et al., 2019; Huang et al., 2022; Palaniappan & Bach, 2016; Yu et al., 2022). All of these results are different from ours. Some recent works exploited Halpern's fixed-point iterations and develop corresponding variance-reduced methods, see, e.g., (Cai et al., 2024; 2022). However, varying parameters or incorporating double-loop/inexact methods must be used to achieve improved theoretical oracle complexity. We believe that such approaches may be challenging to select parameters and to implement in practice. Finally, unlike optimization, it has been realized that using biased estimators such as SARAH or Hybrid-SGD/STORM for (NI) (including (SP) and (VIP)) is challenging due to the lack of an objective function, a key metric to prove convergence, and product terms like $\langle e^k, x^{k+1} - x^\star \rangle$ in convergence analyses, where $e^k$ is a bias rendered from $\widetilde{S}_\gamma^k$ (see Supp. Doc. A).

**Notation.** We use $\mathcal{F}_k := \sigma(x^0, x^1, \cdots, x^k)$ to denote the $\sigma$-algebra generated by $x^0, \cdots, x^k$ up to the iteration $k$. $\mathbb{E}_k\big[\cdot\big] = \mathbb{E}\big[\cdot \mid \mathcal{F}_k\big]$ denotes the conditional expectation w.r.t. $\mathcal{F}_k$, and $\mathbb{E}\big[\cdot\big]$ is the total expectation. We also use $\mathcal{O}(\cdot)$ to characterize convergence rates and oracle complexity. For an operator $G$, $\mathrm{dom}(G) := \{x : Gx \neq \emptyset\}$ denotes its domain, and $J_G$ denotes its resolvent.

**Paper organization.** Section 2 introduces $S_\gamma^k$ and defines a class of stochastic estimators for it. It also constructs two instances: SVRG and SAGA, and proves their key properties. Section 3 develops an algorithm for solving (NE) and establishes its oracle complexity. Section 4 designs a new algorithm for solving (NI) and proves its oracle complexity. Section 5 presents two concrete numerical examples. Proofs and additional results are deferred to Sup. Docs. A to E.

## 2. Forward-Reflected Quantity and Its Stochastic Variance-Reduced Estimators

We first define our forward-reflected quantity (FRQ) for $G$ in (NE) and (NI) using here. Next, we propose a class of unbiased variance-reduced estimators for FRQ. Finally, we construct two instances relying on the two well-known estimators: SVRG from (Johnson & Zhang, 2013) and SAGA from (Defazio et al., 2014).

### 2.1. The forward-reflected quantity

Our methods for solving (NE) and (NI) rely on the following intermediate quantity constructed from $G$ via two consecutive iterates $x^{k-1}$ and $x^k$ controlled by $\gamma \in [0, 1]$:

$$S_\gamma^k := Gx^k - \gamma Gx^{k-1}. \qquad \text{(FRQ)}$$

Here, $\gamma \in \big(\frac{1}{2}, 1\big)$ plays a crucial role in our methods in the sequel. Clearly, if $\gamma = \frac{1}{2}$, then we can write $S_{1/2}^k = \frac{1}{2}Gx^k + \frac{1}{2}(Gx^k - Gx^{k-1}) = \frac{1}{2}[2Gx^k - Gx^{k-1}]$ used in both the forward-reflected-backward splitting (FRBS) method (Malitsky & Tam, 2020) and the optimistic gradient method (Daskalakis et al., 2018). In deterministic unconstrained settings (i.e. solving (NE)), see (Tran-Dinh, 2023), FRBS is also equivalent to Popov's past-extragradient method (Popov, 1980), reflected-forward-backward splitting algorithm (Cevher & Vũ, 2021; Malitsky, 2015), and optimistic gradient scheme (Daskalakis et al., 2018). In the deterministic constrained case, i.e. solving (NI), these methods are different. Since $\gamma \in \big(\frac{1}{2}, 1\big)$, our methods below exclude these classical schemes. However, due to a similarity pattern of (FRQ) and FRBS, we still term our quantity $S_\gamma^k$ by the "**forward-reflected quantity**", abbreviated by FRQ.

### 2.2. Unbiased variance-reduced estimators for FRQ

Now, let us propose the following class of stochastic variance-reduced estimators $\widetilde{S}_\gamma^k$ of $S_\gamma^k$.

**Definition 2.1.** A stochastic estimator $\widetilde{S}_\gamma^k$ is said to be a *stochastic unbiased variance-reduced estimator* of $S_\gamma^k$ in (FRQ) if there exist constants $\rho \in (0, 1]$, $C \geq 0$ and $\hat{C} \geq 0$, and a nonnegative sequence $\{\Delta_k\}$ such that:

$$\begin{cases} \mathbb{E}_k[\widetilde{S}_\gamma^k - S_\gamma^k] = 0, \\ \mathbb{E}[\|\widetilde{S}_\gamma^k - S_\gamma^k\|^2] \leq \Delta_k, \\ \Delta_k \leq (1 - \rho)\Delta_{k-1} + C \cdot U_k + \hat{C} \cdot U_{k-1}, \end{cases} \quad (3)$$

where $U_k := \frac{1}{n}\sum_{i=1}^n \mathbb{E}[\|G_i x^k - G_i x^{k-1}\|^2]$.

Here, $\Delta_{-1} = 0$, $x^{-2} = x^{-1} = x^0$, and $\mathbb{E}_k[\,\cdot\,]$ and $\mathbb{E}[\,\cdot\,]$ are the conditional and total expectations defined earlier, respectively. The condition $\rho > 0$ is important to achieve a variance reduction as long as $x^k$ is close to $x^{k-1}$ and $x^{k-1}$ is close to $x^{k-2}$. Otherwise, $\widetilde{S}_\gamma^k$ may not be a variance-reduced estimator of $S_\gamma^k$. Since $S_\gamma^k$ is evaluated at both $x^{k-1}$ and $x^k$, our bound for the estimator $\widetilde{S}_\gamma^k$ depends on three consecutive points $x^{k-2}$, $x^{k-1}$, and $x^k$, which is different from previous works, including (Alacaoglu et al., 2021; Beznosikov et al., 2023; Davis, 2022; Driggs et al., 2020).

Now, we will construct two variance-reduced estimators satisfying Definition 2.1 by exploiting SVRG (Johnson & Zhang, 2013) and SAGA (Defazio et al., 2014).

(a) **Loopless-SVRG estimator for $S_\gamma^k$.** Consider a mini-batch $\mathcal{B}_k \subseteq [n] := \{1, 2, \cdots, n\}$ with a fixed batch size $b := |\mathcal{B}_k|$. Denote $G_{\mathcal{B}_k} z := \frac{1}{b}\sum_{i \in \mathcal{B}_k} G_i z$ for $z \in \mathrm{dom}(G)$. We define the following estimator for $S_\gamma^k$:

$$\begin{aligned} \widetilde{S}_\gamma^k := {}&(1 - \gamma)(Gw^k - G_{\mathcal{B}_k} w^k) \\ &+ G_{\mathcal{B}_k} x^k - \gamma G_{\mathcal{B}_k} x^{k-1}, \end{aligned} \quad \text{(L-SVRG)}$$

where the snapshot point $w^k$ is selected randomly as follows:

$$w^{k+1} := \begin{cases} x^k & \text{with probability } \mathbf{p} \\ w^k & \text{with probability } 1 - \mathbf{p}. \end{cases} \quad (4)$$

The probability $\mathbf{p} \in (0, 1)$ will appropriately be chosen later by nonuniformly flipping a coin. This estimator is known as a loopless variant (Kovalev et al., 2020) of the SVRG estimator (Johnson & Zhang, 2013). However, it is different from existing ones used in root-finding algorithms, including (Davis, 2022) because we define it for $S_\gamma^k$, not for $Gx^k$. In addition, the first term is also damped by a factor $1 - \gamma$ to guarantee the unbiasedness of $\widetilde{S}_\gamma^k$ to $S_\gamma^k$.

The following lemma shows that $\widetilde{S}_\gamma^k$ satisfies Definition 2.1.

**Lemma 2.2.** *Let $S_\gamma^k$ be given by (FRQ) and $\widetilde{S}_\gamma^k$ be generated by the SVRG estimator (L-SVRG) and*

$$\Delta_k := \frac{1}{nb}\sum_{i=1}^n \mathbb{E}[\|G_i x^k - \gamma G_i x^{k-1} - (1-\gamma)G_i w^k\|^2].$$

*Then, $\widetilde{S}_\gamma^k$ satisfies Definition 2.1 with this $\{\Delta_k\}$, $\rho := \frac{\mathbf{p}}{2}$, $C := \frac{4 - 6\mathbf{p} + 3\mathbf{p}^2}{b\mathbf{p}}$, and $\hat{C} := \frac{2\gamma^2(2 - 3\mathbf{p} + \mathbf{p}^2)}{b\mathbf{p}}$.*

(b) **SAGA estimator for $S_\gamma^k$.** Given $S_\gamma^k$ as in (FRQ) and a mini-batch estimator $G_{\mathcal{B}_k}$ as in (L-SVRG), we construct the following SAGA estimator for $S_\gamma^k$:

$$\begin{aligned} \widetilde{S}_\gamma^k := {}&[G_{\mathcal{B}_k} x^k - \gamma G_{\mathcal{B}_k} x^{k-1} - (1-\gamma)\hat{G}_{\mathcal{B}_k}^k] \\ &+ \frac{1-\gamma}{n}\sum_{i=1}^n \hat{G}_i^k, \end{aligned} \quad \text{(SAGA)}$$

where $\mathcal{B}_k$ is a mini-batch of size $b$, and $\hat{G}_i^k$ is updated as

$$\hat{G}_i^{k+1} := \begin{cases} G_i x^k & \text{if } i \in \mathcal{B}_k, \\ \hat{G}_i^k & \text{otherwise.} \end{cases} \quad (5)$$

To form $\widetilde{S}_\gamma^k$, we need to store $n$ components $\hat{G}_i^k$ computed so far for $i \in [n]$ in a table $\mathcal{T}_k := [\hat{G}_1^k, \hat{G}_2^k, \cdots, \hat{G}_n^k]$ initialized at $\hat{G}_i^0 := G_i x^0$ for all $i \in [n]$. Clearly, the SAGA estimator requires significant memory to store $\mathcal{T}_k$ if $n$ and $p$ are both large. We have the following result.

**Lemma 2.3.** *Let $S_\gamma^k$ be defined by (FRQ) and $\widetilde{S}_\gamma^k$ be generated by the SAGA estimator (SAGA), and*

$$\Delta_k := \frac{1}{nb}\sum_{i=1}^n \mathbb{E}[\|G_i x^k - \gamma G_i x^{k-1} - (1-\gamma)\hat{G}_i^k\|^2].$$

*Then, $\widetilde{S}_\gamma^k$ satisfies Definition 2.1 with this $\{\Delta_k\}$, $\rho := \frac{b}{2n} \in (0, 1]$, $C := \frac{[2(n-b)(2n+b)+b^2]}{nb}$, and $\hat{C} := \frac{2(n-b)(2n+b)\gamma^2}{nb}$.*

We only provide two instances: (L-SVRG) and (SAGA) covered by Definition 2.1. However, we believe that similar estimators for $S_\gamma^k$ relied on, e.g., JacSketch (Gower et al., 2021) or SEGA (Hanzely et al., 2018), among others can fulfill our Definition 2.1.

# 3. A Variance-Reduced Forward-Reflected Method for [Non]linear Equations

We first utilize the class of stochastic estimators in Definition 2.1 to develop a variance-reduced forward-reflected (VFR) method for solving (NE) under Ass. 1.3 and 1.4.

## 3.1. The VFR method and its convergence guarantee

(a) **VFR Method.** Our method is described as follows. *Starting from $x^0 \in \mathrm{dom}(G)$, at each iteration $k \geq 0$, we construct an estimator $\widetilde{S}_\gamma^k$ satisfying Definition 2.1 with parameters $\rho \in (0, 1]$, $C \geq 0$, and $\hat{C} \geq 0$, and then update*

$$x^{k+1} := x^k - \eta\widetilde{S}_\gamma^k, \quad \text{(VFR)}$$

*where $\eta > 0$ and $\gamma > 0$ are determined below, $x^{-1} = x^{-2} := x^0$, and $\widetilde{S}_\gamma^0 := (1 - \gamma)Gx^0$.*

At least two estimators $\widetilde{S}_\gamma^k$: the *Loopless-SVRG estimator* in (L-SVRG) and the *SAGA estimator* in (SAGA), can be used in our method (VFR). In terms of *per-iteration complexity*, each iteration $k$ of VFR, the loopless SVRG variant requires three mini-batch evaluations $G_{\mathcal{B}_k} w^k$, $G_{\mathcal{B}_k} x^k$, and $G_{\mathcal{B}_k} x^{k-1}$ of $G$, and occasionally computes one full evaluation $Gw^k$

of $G$ with the probability $\mathbf{p}$. It needs one more mini-batch evaluation $G_{\mathcal{B}_k} x^{k-1}$ compared to SVRG-type methods in optimization. Similarly, the SAGA estimator also requires two mini-batch evaluations $G_{\mathcal{B}_k} x^k$ and $G_{\mathcal{B}_k} x^{k-1}$, which is one more mini-batch $G_{\mathcal{B}_k} x^{k-1}$ compared to SAGA-type methods in optimization, see, e.g., (Reddi et al., 2016b). The SAGA estimator can avoid the occasional full-batch evaluation $Gw^k$ from L-SVRG, but as a compensation, we need to store a table $\mathcal{T}_k := [\hat{G}_1^k, \hat{G}_2^k, \cdots, \hat{G}_n^k]$, which requires significant memory in the large-scale regime.

(b) **Convergence guarantee.** Fixed $\gamma \in \left(\frac{1}{2}, 1\right)$, with $\rho$, $C$, and $\hat{C}$ as in Definition 2.1 we define

$$M := \frac{\gamma(1+5\gamma)}{3(2\gamma-1)} + \frac{1+6\gamma}{3(2\gamma-1)} \cdot \frac{C+\hat{C}}{\rho} \quad \text{and} \quad \delta := \frac{2\gamma-1}{8\sqrt{M}}. \quad (6)$$

Then, the following theorem states the convergence of (VFR), whose proof is given in Supp. Doc. C.

**Theorem 3.1.** *Let us fix $\gamma \in \left(\frac{1}{2}, 1\right)$, and define $M$ and $\delta$ as in (6). Suppose that Assumptions 1.1, 1.3, and 1.4 hold for (NE) with some $\kappa \geq 0$ such that $L\kappa \leq \delta$. Let $\{x^k\}$ be generated by (VFR) using a learning rate $\eta > 0$ such that $\frac{8\kappa}{2\gamma-1} \leq \eta \leq \frac{1}{L\sqrt{M}}$. Then, the following bounds hold:*

$$\begin{aligned}
\frac{1}{K+1} \sum_{k=0}^{K} \mathbb{E}\left[\|Gx^k\|^2\right] &\leq \frac{\Theta_1 \|x^0 - x^\star\|^2}{K+1}, \\
\frac{1}{K+1} \sum_{k=1}^{K} \mathbb{E}\left[\|x^k - x^{k-1}\|^2\right] &\leq \frac{\Theta_2 \|x^0 - x^\star\|^2}{K+1},
\end{aligned} \quad (7)$$

*where $\Theta_1 := \frac{2(1+L^2\eta^2)}{\gamma(1-\gamma)\eta^2}$ and $\Theta_2 := \frac{8(1+L^2\eta^2)}{3(2\gamma-1)(1-ML^2\eta^2)}$*

Theorem 3.1 only proves a $\mathcal{O}(1/K)$ convergence rate of both $\frac{1}{K+1}\sum_{k=0}^{K}\mathbb{E}\left[\|Gx^k\|^2\right]$ and $\frac{1}{K+1}\sum_{k=1}^{K}\mathbb{E}\left[\|x^k - x^{k-1}\|^2\right]$, but does not characterize the oracle complexity of (VFR). If we choose $\gamma := \frac{3}{4}$, then from (6), we have $M = \frac{57}{24} + \frac{11(C+\hat{C})}{3\rho}$ and $\delta = \frac{1}{16\sqrt{M}}$, which can simplify the bounds in Theorem 3.1. In addition, it allows $\kappa > 0$ such that $L\kappa \leq \delta = \mathcal{O}(\sqrt{\rho})$, which means that $\kappa$ can be positive, but depends on $\sqrt{\rho}$. This condition allows us to cover a class of nonmonotone operators $G$, where a weak-Minty solution exists as stated in Assumption 1.4.

**3.2. Complexity Bounds of VFR with SVRG and SAGA**

Let us first apply Theorem 3.1 to the mini-batch SVRG estimator (L-SVRG) in Section 2. For simplicity, we choose $\gamma := \frac{3}{4}$ and $\eta := \frac{1}{L\sqrt{M}}$, but any $\gamma \in \left(\frac{1}{2}, 1\right)$ still works.

**Corollary 3.2.** *Suppose that Assumptions 1.1, 1.3, and 1.4 hold for (NE) with $\kappa \geq 0$ as in Theorem 3.1. Let $\{x^k\}$ be generated by (VFR) using (L-SVRG), $\gamma := \frac{3}{4}$, and $\eta := \frac{1}{L\sqrt{M}} \geq \frac{0.1440\sqrt{b}\mathbf{p}}{L}$, provided that $b\mathbf{p}^2 \leq 1$. Then*

$$\frac{1}{K+1} \sum_{k=0}^{K} \mathbb{E}\left[\|Gx^k\|^2\right] \leq \frac{526L^2\|x^0 - x^\star\|^2}{b\mathbf{p}^2(K+1)}. \quad (8)$$

*For $\epsilon > 0$, if we choose $\mathbf{p} := n^{-1/3}$ and $b := \lfloor n^{2/3} \rfloor$, then (VFR) requires $\mathcal{T}_{G_i} := n + \lfloor \frac{4\Gamma L^2 R_0^2 n^{2/3}}{\epsilon^2} \rfloor$ evaluations of $G_i$ to attain $\frac{1}{K+1}\sum_{k=0}^{K}\mathbb{E}\left[\|Gx^k\|^2\right] \leq \epsilon^2$, where $\Gamma := 731$.*

Corollary 3.2 states that the oracle complexity of (VFR) is $\mathcal{O}\left(n + n^{2/3}\epsilon^{-2}\right)$, matching (up to a constant) the one of SVRG in nonconvex optimization, see, e.g., (Allen-Zhu & Hazan, 2016; Reddi et al., 2016a). It improves by a factor $\mathcal{O}\left(n^{1/3}\right)$ compared to deterministic counterparts. This complexity is known to be the best for SVRG so far without any additional enhancement (e.g., nested techniques (Zhou et al., 2018)) even for a special case of (NE): $Gx = \nabla f(x)$ in nonconvex optimization.

Note that $\eta$ can be computed explicitly when $b$ and $\mathbf{p}$ are given. For example, if $n = 10000$ and we choose $\mathbf{p} = n^{-1/3} = 0.0464$ and $b = \lfloor n^{2/3} \rfloor = 464$, then $\eta = \frac{0.1456}{L}$. If $\mathbf{p} = 0.1$, then $\eta = \frac{0.3038}{L}$. Note that, in general, we can choose appropriate $p := \mathcal{O}(n^{-1/3})$ and $b := \mathcal{O}(n^{2/3})$.

Alternatively, we can apply Theorem 3.1 to (SAGA).

**Corollary 3.3.** *Suppose that Assumptions 1.1, 1.3, and 1.4 hold for (NE) with $\kappa \geq 0$ as in Theorem 3.1. Let $\{x^k\}$ be generated by (VFR) using (SAGA), $\gamma := \frac{3}{4}$, and $\eta := \frac{1}{L\sqrt{M}} \geq \frac{0.1494b^{3/2}}{nL}$, provided that $1 \leq b \leq n^{2/3}$. Then*

$$\frac{1}{K+1} \sum_{k=0}^{K} \mathbb{E}\left[\|Gx^k\|^2\right] \leq \frac{489L^2\|x^0 - x^\star\|^2}{b\mathbf{p}^2(K+1)}. \quad (9)$$

*Moreover, for a given $\epsilon > 0$, if we choose $b := \lfloor n^{2/3} \rfloor$, then (VFR) requires $\mathcal{T}_{G_i} := n + \lfloor \frac{3\Gamma L^2 R_0^2 n^{2/3}}{\epsilon^2} \rfloor$ evaluations of $G_i$ to achieve $\frac{1}{K+1}\sum_{k=0}^{K}\mathbb{E}\left[\|Gx^k\|^2\right] \leq \epsilon^2$, where $\Gamma := 2816$.*

Similar to Corollary 3.2, the learning rate $\eta$ in Corollary 3.3 can explicitly be computed if we know $n$ and $b$. For instance, if $n = 10000$, and we choose $b = \lfloor n^{2/3} \rfloor$, then $\eta = \frac{0.1603}{L}$.

If $\kappa = 0$, i.e. $G$ reduces to a star-monotone operator, then we can choose $\gamma \in \left(\frac{1}{2}, 1\right)$ and $\eta$ as:

- For SVRG, we have $\eta \in \left(0, \frac{1}{L\sqrt{M}}\right]$. If we choose $\mathbf{p} = \mathcal{O}\left(n^{-1/3}\right)$ and $b = \mathcal{O}\left(n^{2/3}\right)$, then $\eta = \mathcal{O}\left(\frac{1}{L}\right)$;
- For SAGA, we have $\eta \in \left(0, \frac{1}{L\sqrt{M}}\right]$. If we choose $b = \mathcal{O}\left(n^{2/3}\right)$, then $\eta = \mathcal{O}\left(\frac{1}{L}\right)$.

Hitherto, the constant factor $\Gamma$ in both corollaries is still relatively large, but it can be further improved by refining our technical proofs (e.g., carefully using Young's inequality).

# 4. A New Variance-Reduced FRBS Method for Nonmonotone Generalized Equations

In this section, we develop a new stochastic variance-reduced forward-reflected-backward splitting (FRBS) method to solve (NI) under Assumptions 1.2, 1.3, and 1.4.

### 4.1. The algorithm and its convergence

(a) **The variance-reduced FRBS method (VFRBS).** Our scheme for solving (NI) is as follows. *Starting from $x^0 \in$ dom$(\Psi)$, at each iteration $k \geq 0$, we generate an estimator $\widetilde{S}_\gamma^k$ satisfying Definition 2.1 and update*

$$x^{k+1} := x^k - \eta \widetilde{S}_\gamma^k - \eta\big(\gamma v^{k+1} - (2\gamma-1)v^k\big), \quad \text{(VFRBS)}$$

*where $\eta > 0$ and $\gamma > 0$ are determined later, $v^k \in Tx^k$, $x^{-1} = x^{-2} := x^0$, and $\widetilde{S}_\gamma^0 := (1-\gamma)Gx^0$.*

(b) **Implementable version.** Since $v^{k+1} \in Tx^{k+1}$ appears on the RHS of (VFRBS), using the resolvent $J_{\gamma\eta T} := (\mathbb{I} + \gamma\eta T)^{-1}$ of $T$, we can rewrite (VFRBS) equivalently to

$$\begin{cases} y^{k+1} := x^k - \eta \widetilde{S}_\gamma^k + \frac{(2\gamma-1)}{\gamma}(y^k - x^k), \\ x^{k+1} := J_{\gamma\eta T}\big(y^{k+1}\big). \end{cases} \quad (10)$$

Here, $y^0 \in$ dom$(\Psi)$ is given, and $x^0 = x^{-1} := J_{\gamma\eta T}(y^0)$. This is an implementable variant of (VFRBS) using the resolvent $J_{\gamma\eta T}$. Clearly, if $\gamma = \frac{1}{2}$, then (10) reduces to

$$x^{k+1} := J_{(\eta/2)T}\big(x^k - \eta \widetilde{S}_{1/2}^k\big),$$

which can be viewed as a stochastic forward-reflected-backward splitting scheme. However, our $\gamma \in \left(\frac{1}{2}, 1\right)$, making (10) different from existing methods, even in the deterministic case.

Compared to (Alacaoglu & Malitsky, 2022), (10) requires only one $J_{\gamma\eta T}$ as in (Alacaoglu et al., 2023), while (Alacaoglu & Malitsky, 2022) needs more than ones. Moreover, our estimator $\widetilde{S}_\gamma^k$ is also different from (Alacaoglu & Malitsky, 2022). Compared to (Beznosikov et al., 2023) and (Alacaoglu et al., 2023), the term $\gamma^{-1}(2\gamma - 1)(y^k - x^k)$ makes it different from SGDA in (Beznosikov et al., 2023) and the golden-ratio method in (Alacaoglu et al., 2023), and also other existing deterministic methods.

(c) **Approximate solution certification.** To certify an approximate solution of (NI), we note that its exact solution $x^\star \in$ zer$(\Psi)$ satisfies $\|Gx^\star + v^\star\|^2 = 0$ for some $v^\star \in Tx^\star$. Therefore, if $(x^k, v^k)$ satisfies $\mathbb{E}\big[\|Gx^k + v^k\|^2\big] \leq \epsilon^2$ for some $v^k \in Tx^k$, then we can say that $x^k$ is an $\epsilon$-solution of (NI). Alternatively, we can define the following forward-backward splitting (FBS) residual of (NI):

$$\mathcal{G}_\eta x := \eta^{-1}(x - J_\eta(x - \eta Gx)),$$

for any given $\eta > 0$. It is well-known that $x^\star \in$ zer$(\Psi)$ iff $\mathcal{G}_\eta x^\star = 0$. Hence, if $\mathbb{E}\big[\|\mathcal{G}_\eta x^k\|^2\big] \leq \epsilon^2$, then $x^k$ is also called an $\epsilon$-solution of (NI). One can easily prove that $\|\mathcal{G}_\eta x^k\| \leq \|Gx^k + v^k\|$ for any $v^k \in Tx^k$. Clearly, the former metric implies the latter one. Therefore, it is sufficient to only certify $\mathbb{E}\big[\|Gx^k + v^k\|^2\big] \leq \epsilon^2$, which implies $\mathbb{E}\big[\|\mathcal{G}_\eta x^k\|^2\big] \leq \epsilon^2$.

(d) **Convergence analysis.** For simplicity of our presentation, for a given $\gamma \in \left(\frac{1}{2}, 1\right)$, with $\rho$, $C$, and $\hat{C}$ in Definition 2.1, we define the following two parameters:

$$M := 4\gamma^2 + \frac{4\gamma}{1-\gamma} \cdot \frac{C+\hat{C}}{\rho} \quad \text{and} \quad \delta := \frac{\gamma(2\gamma-1)}{(3\gamma-1)\sqrt{M}}. \quad (11)$$

Then, Theorem 4.1 below states the convergence of (VFRBS), whose proof can be found in Supp. Doc. D.

**Theorem 4.1.** *Let us fix $\gamma \in \left(\frac{1}{2}, 1\right)$, and define $M$ and $\delta$ as in (11). Suppose that Assumptions 1.1, 1.2, 1.3, and 1.4 hold for (NI) for some $\kappa \geq 0$ such that $L\kappa < \delta$. Let $\{x^k\}$ be generated by (VFRBS) using a fixed learning rate $\eta$ such that $\frac{(3\gamma-1)\kappa}{\gamma(2\gamma-1)} < \eta \leq \frac{1}{L\sqrt{M}}$. Then, we have*

$$\begin{aligned} \frac{1}{K+1}\sum_{k=0}^K \mathbb{E}\big[\|Gx^k + v^k\|^2\big] &\leq \frac{\hat{\Theta}_1\hat{R}_0^2}{\eta^2(K+1)}, \\ \frac{1}{K+1}\sum_{k=0}^K \mathbb{E}\big[\|x^k - x^{k-1}\|^2\big] &\leq \frac{\hat{\Theta}_2\hat{R}_0^2}{K+1}, \end{aligned} \quad (12)$$

*where $\hat{R}_0^2$, $\hat{\Theta}_1$, and $\hat{\Theta}_2$ are respectively given by*

$$\begin{aligned} \hat{R}_0^2 &:= \|x^0 - x^\star\|^2 + \gamma^2\eta^2\|Gx^0 + v^0\|^2, \\ \hat{\Theta}_1 &:= \frac{(3\gamma-1)\eta}{(1-\gamma)[\gamma(2\gamma-1)\eta-(3\gamma-1)\kappa]}, \\ \hat{\Theta}_2 &:= \frac{4(3\gamma-1)\hat{R}_0^2}{(1-\gamma)(1-ML^2\eta^2)}. \end{aligned}$$

The bounds in Theorem 4.1 are similar to Theorem 3.1, but their proof relies on a new Lyapunov function. Note that the condition on $L\kappa$ still depends on $\rho$ as $L\kappa \leq \delta = \mathcal{O}\big(\sqrt{\rho}\big)$.

### 4.2. Complexity of VFRBS with SVRG and SAGA

Similar to Section 3, we can apply Theorem 4.1 for the mini-batch SVRG estimator in Section 2.

**Corollary 4.2.** *Suppose that Assumptions 1.1, 1.2 1.3, and 1.4 hold for (NI) with $\kappa \geq 0$ as in Theorem 4.1. Let $\{x^k\}$ be generated by (VFRBS) using the SVRG estimator (L-SVRG), $\gamma \in \left(\frac{1}{2}, 1\right)$, and $\eta := \frac{1}{L\sqrt{M}} \geq \frac{\sigma\sqrt{b}\mathbf{p}}{L}$ with $\sigma := \frac{\sqrt{1-\gamma}}{2\sqrt{8+\gamma+7\gamma^2}}$, provided that $b\mathbf{p}^2 \leq 1$. Then, we have*

$$\frac{1}{K+1}\sum_{k=0}^K \mathbb{E}\big[\|Gx^k + v^k\|^2\big] \leq \frac{\hat{\Theta}_1 L^2 \hat{R}_0^2}{\sigma^2 b\mathbf{p}^2(K+1)}, \quad (13)$$

*where $\hat{R}_0^2 := \|x^0 - x^\star\|^2 + \gamma^2\eta^2\|Gx^0 + v^0\|^2$.*

*For given $\epsilon > 0$, if we choose $\mathbf{p} := n^{-1/3}$ and $b := \lfloor n^{2/3}\rfloor$, then (VFRBS) requires $\mathcal{T}_{G_i} := n + \lfloor \frac{4\Gamma L^2 \hat{R}_0^2 n^{2/3}}{\epsilon^2}\rfloor$ evaluations of $G_i$ and $\mathcal{T}_T = \lfloor \frac{\Gamma L^2 \hat{R}_0^2}{\epsilon^2}\rfloor$ evaluations of $J_{\gamma\eta T}$ to achieve $\frac{1}{K+1}\sum_{k=0}^K \mathbb{E}\big[\|Gx^k + v^k\|^2\big] \leq \epsilon^2$, where $\Gamma := \frac{\hat{\Theta}_1}{\sigma^2}$.*

Alternatively, we can apply Theorem 4.1 to the mini-batch SAGA estimator (SAGA) in Section 2.

**Corollary 4.3.** *Suppose that Assumptions 1.1, 1.2, 1.3, and 1.4 hold for (NI) with $\kappa \geq 0$ as in Theorem 4.1. Let $\{x^k\}$ be generated by (VFRBS) using the SAGA estimator (SAGA), $\gamma \in \left(\frac{1}{2}, 1\right)$, and $\eta := \frac{1}{L\sqrt{M}} \geq \frac{\sigma b^{3/2}}{nL}$ with $\sigma := \frac{\sqrt{1-\gamma}}{2\sqrt{\gamma(10+\gamma+7\gamma^2)}}$, provided that $1 \leq b \leq n^{2/3}$. Then*

$$\frac{1}{K+1}\sum_{k=0}^{K}\mathbb{E}\big[\|Gx^k + v^k\|^2\big] \leq \frac{n^2\hat{\Theta}_1 L^2 \hat{R}_0^2}{\sigma^2 b^3(K+1)}, \quad (14)$$

*where $\hat{R}_0^2 := \|x^0 - x^\star\|^2 + \gamma^2\eta^2\|Gx^0 + v^0\|^2$.*

*For a given $\epsilon > 0$, if we choose $b := \lfloor n^{2/3}\rfloor$, then (VFRBS) requires $\mathcal{T}_{G_i} := n + \left\lfloor\frac{3\Gamma L^2\hat{R}_0^2 n^{2/3}}{\varepsilon^2}\right\rfloor$ evaluations of $G_i$ and $\mathcal{T}_T = \left\lfloor\frac{\Gamma L^2\hat{R}_0^2}{\epsilon^2}\right\rfloor$ evaluations of $J_{\gamma\eta T}$ to achieve $\frac{1}{K+1}\sum_{k=0}^{K}\mathbb{E}\big[\|Gx^k + v^k\|^2\big] \leq \epsilon^2$, where $\Gamma := \frac{\hat{\Theta}_1}{\sigma^2}$.*

Similar to Subsection 3.2, when $\gamma$, $n$, $b$, and $\mathbf{p}$ are given, we can compute concrete values of the theoretical learning rate $\eta$ in both corollaries. They are larger than the corresponding lower bounds given in these corollaries.

## 5. Numerical Experiments

We provide two examples to illustrate (VFR) and (VFRBS) and compare them with other methods.

### 5.1. Nonconvex-nonconcave minimax optimization

We consider the following nonconvex-nonconcave minimax optimization problem as a special case of (SP):

$$\min_{u\in\mathbb{R}^{p_1}}\max_{v\in\mathbb{R}^{p_2}}\big\{\mathcal{L}(u,v) := \varphi(u) + \mathcal{H}(u,v) - \psi(v)\big\}, \quad (15)$$

where $\mathcal{H}(u,v) := \frac{1}{n}\sum_{i=1}^{n}\mathcal{H}_i(u,v) = \frac{1}{n}\sum_{i=1}^{n}[u^T A_i u + u^T L_i v - v^T B_i v + b_i^\top u - c_i^\top v]$ such that $A_i \in \mathbb{R}^{p_1\times p_1}$ and $B_i \in \mathbb{R}^{p_2\times p_2}$ are symmetric matrices, $L_i \in \mathbb{R}^{p_1\times p_2}$, $b_i \in \mathbb{R}^{p_1}$, and $c_i \in \mathbb{R}^{p_2}$; $\varphi$ and $\psi$ are two proper, closed, and convex functions. The optimality of (15) becomes (NI) (see Supp. Doc. E). In our experiments, we choose $A_i$ and $B_i$ to be not positive semidefinite such that Assumption 1.4 holds. Thus, (15) is nonconvex-nonconcave.

We generate $A_i = Q_i D_i Q_i^T$ for a given orthonormal matrix $Q_i$ and a diagonal matrix $D_i$, where its entries $D_i^j$ are generated from standard normal distribution and clipped by $\max\{D_i^j, -0.1\}$. The matrix $B_i$ is also generated by the same way, while $L_i$, $b_i$, and $c_i$ are generated from standard normal distribution. Hence, $\mathbf{G}$ in (NI) is not symmetric and also not positive semidefinite.

**The unconstrained case.** We implement three variants of (VFR): `VFR-svrg` (double-loop SVRG), `LVFR-svrg` (loopless SVRG), `VFR-saga` (using SAGA estimator) in Python to solve (15) when both $\varphi$ and $\psi$ are vanished, i.e. its optimality is a special case of (NE). We also compare our methods with the deterministic optimistic gradient method (`OG`) in (Daskalakis et al., 2018), the variance-reduced

FRBS scheme (`VFRBS`) in (Alacaoglu et al., 2023), and the variance-reduced extragradient algorithm (`VEG`) in (Alacaoglu & Malitsky, 2022). We select the parameters as suggested by our theory, while choosing appropriate parameters for `OG`, `VFRBS`, and `VEG`. The details of this experiment, including generating data and specific choice of parameters, are given in Supp. Doc. E.

The relative residual norm $\|Gx^k\|/\|Gx^0\|$ against the number of epochs averaged on 10 problem instances is revealed in Figure 1 for two datasets $(p,n) = (100, 5000)$ and $(p,n) = (200, 10000)$.

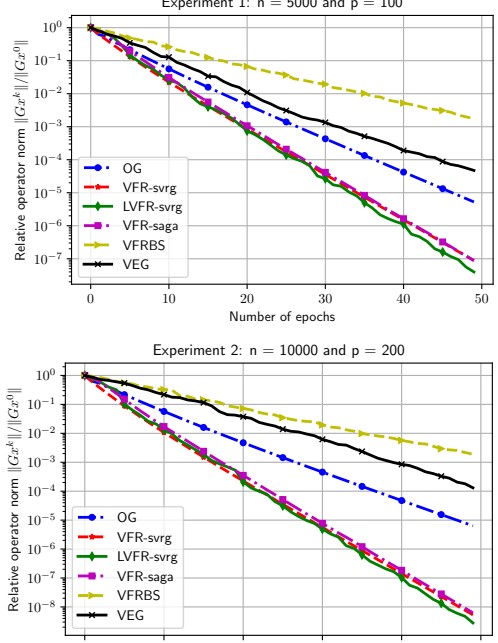

Figure 1: Comparison of 6 algorithms to solve the unconstrained (15) on 2 experiments (The average of 10 runs).

Clearly, with these experiments, three SVRG variants of our method (VFRBS) work well and significantly outperform other competitors. The `LVFR-svrg` variant of (VFRBS) seems to work best, while `VFRBS` and `VEG` still cannot beat the deterministic algorithm `OG` in this example.

**The constrained case.** We now adding two simplex constraints $u \in \Delta_{p_1}$ and $v \in \Delta_{p_2}$ to (15), where $\Delta_p := \{u \in \mathbb{R}_+^p : \sum_{i=1}^{p} u_i = 1\}$ is the standard simplex in $\mathbb{R}^p$. These constraints are common in bilinear games. To handle these constraints, we set $\varphi(u) := \delta_{\Delta_{p_1}}(u)$ and $\psi(v) := \delta_{\Delta_{p_2}}(v)$ as the indicators of $\Delta_{p_1}$ and $\Delta_{p_2}$, respectively.

Again, we run 6 algorithms for solving this constrained case of (15) using the same parameters as **the unconstrained case**. We report the relative norm of the FBS residual $\|\mathcal{G}_\eta x^k\|/\|\mathcal{G}_\eta x^0\|$ against the number of epochs. The results are revealed in Figure 2 for two datasets $(p,n) = (100, 5000)$ and $(p,n) = (200, 10000)$.

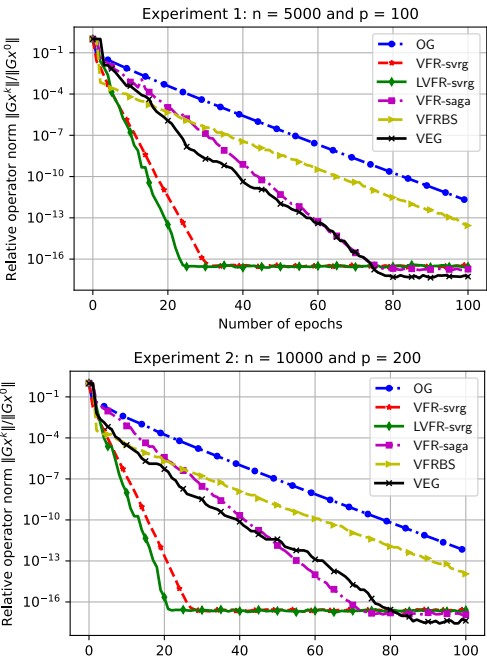

Figure 2: The performance of 6 algorithms to solve the constrained (15) on 2 experiments (The average of 10 runs).

Clearly, with these experiments, both SVRG variants of our method (VFRBS) work well and significantly outperform other competitors. The SVRG variant (VFR-svrg) of (VFRBS) seems to work best, while our VFR-saga has a similar performance as VEG. Again, we also see that VFRBS tends to have a similar performance as OG.

### 5.2. Logistic regression with ambiguous features

We consider the following minimax optimization problem arising from a regularized logistic regression with ambiguous features (see Supp. Doc. E for more details):

$$\min_{w \in \mathbb{R}^d} \max_{z \in \mathbb{R}^m} \Big\{ \mathcal{L}(w, z) := \frac{1}{N} \sum_{i=1}^N \sum_{j=1}^m z_j \ell(\langle X_{ij}, w \rangle, y_i) \tag{16}$$
$$+ \tau R(w) - \delta_{\Delta_m}(z) \Big\},$$

where $\ell(\tau, s) := \log(1 + \exp(\tau)) - s\tau$ is the standard logistic loss, $R(w) := \|w\|_1$ is an $\ell_1$-norm regularizer, $\tau > 0$ is a regularization parameter, and $\delta_{\Delta_m}$ is the indicator of $\Delta_m$ to handle the constraint $z \in \Delta_m$. Then, the optimality condition of (16) can be cast into (NI), where $x := [w, z]$.

We implement three variants of (VFRBS) to solve (16): VFR-svrg, LVFR-svrg, and VFR-saga. We also compare our methods with OG, VFRBS, and VEG as in Subsection 5.1. We cary out a mannual tuning procedure to select appropriate learning rates for all methods. We test these algorithms on two real datasets: a9a (134 features and 3561 samples) and w8a (311 features and 45546 samples) downloaded from LIBSVM (Chang & Lin, 2011). We first normalize the feature vector $\hat{X}_i$ and add a column of all ones to address the bias term. To generate ambiguous features,

we take the nominal feature vector $\hat{X}_i$ and add a random noise generated from a normal distribution of zero mean and variance of $\sigma^2 = 0.5$. In our test, we choose $\tau := 10^{-3}$ and $m := 10$. The relative FBS residual norm $\|\mathcal{G}_\eta x^k\|/\|\mathcal{G}_\eta x^0\|$ against the epochs is plotted in Figure 3 for both datasets.

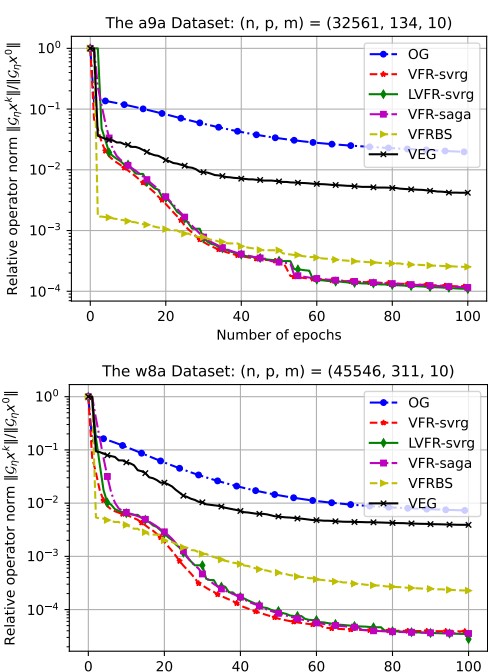

Figure 3: Comparison of 6 algorithms to solve (16) on two real datasets: a8a and w8a.

As we can observe from Figure 3 that three variants VFR-svrg, LVFR-svrg, and VFR-saga have similar performance and are better than their competitors. Among three competitors, VFRBS still works well, and is much better than OG and VEG. The deterministic method, OG, is the worst one in terms of oracle complexity. In this test, VEG has a larger learning rate than ours and VFRBS.

### 6. Conclusions

We develop two new variance-reduced algorithms based on the forward-reflected-backward splitting method to tackle both root-finding problems (NE) and (NI). These methods encompass both SVRG and SAGA estimators as special cases. By carefully selecting the parameters, our algorithms achieve the state-of-the-art oracle complexity for attaining an $\epsilon$-solution, matching the state-of-the-art complexity bounds observed in nonconvex optimization methods using SVRG and SAGA. While the first scheme resembles a stochastic variant of the optimistic gradient method, the second one is entirely novel and distinct from existing approaches, even their deterministic counterparts. We have validated our methods through numerical examples, and the results demonstrate promising performance compared to existing techniques under careful parameter selections.

## Impact Statement

This paper proposes new algorithms with rigorous convergence guarantees and complexity estimates for solving a broad class of large-scale problems. These problems cover many fundamental challenges and applications in optimization, machine learning, and related fields as special cases. We believe that our new algorithms have the potential to make a significant impact in machine learning and related areas. Additionally, there are various potential societal consequences of our work, though none that we feel require specific emphasis at this time.

## Acknowledgements

This work was partly supported by the National Science Foundation (NSF): NSF-RTG grant No. NSF DMS-2134107 and the Office of Naval Research (ONR), grant No. N00014-23-1-2588.

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

# Supplementary Document:
# Variance-Reduced Forward-Reflected-Backward Splitting Methods for Nonmonotone Generalized Equations

Due to space limit, some parts of our algorithmic construction and theory are not described in detail and motivated in the main text. This supplementary document aims at providing more details of the algorithmic construction, motivation, related work, technical proofs, and additional experiments related to our methods.

## A. Further Discussion of Related Work and Assumptions

Let us further expand our discussion of related work in the main text. Then, we show that our assumptions, Assumptions 1.3 and 1.4, indeed cover nonmonotone problems.

### A.1. Further Discussion of Related Work

As we already discussed in the introduction of the main text, both standard stochastic approximation and variance-reduction methods have been broadly studied for (NE) and (NI), including (Juditsky et al., 2011; Kotsalis et al., 2022; Pethick et al., 2023). In this section, we further discuss some other related work to (NE) and (NI), their special cases, and equivalent forms.

(a) **Beyond monotonicity.** Classical methods such as extragradient, prox-mirror, and projective schemes often relax the monotonicity to star-monotonicity, and other forms such as pseudo-monotonicity and quasi-monotonicity (Konnov, 2001; Noor, 2003; Noor & Al-Said, 1999; Tu, 2018). These assumptions are certainly weaker than the monotonicity and can cover some wider classes of problems, including some nonmonotone subclasses. Another extension of monotonicity is the weak-Minty solution condition in Assumption 1.4, which was proposed in early work, perhaps in the most recent one such as (Diakonikolas et al., 2021), as an extension of the star-monotonicity and star-weak-monotonicity assumptions. Other following-up works include (Böhm, 2022; Gorbunov et al., 2022b; Luo & Tran-Dinh, 2022). A comprehensive survey for extragradient-type methods using the weak-Minty solution condition can be found in (Tran-Dinh, 2023; Tran-Dinh & Nguyen-Trung, 2025a;b). The monotonicity has also been extended to a weak monotonicity, or related, prox-regularity (Rockafellar & Wets, 1997) (in particular, weak-convexity). Other types of hypo-monotonicity or co-monotonicity concepts can be found, e.g., in (Bauschke et al., 2020). These concepts have been exploited to develop algorithms for solving (NE) and (NI) and their special cases. For stochastic methods, extensions beyond monotonicity have been also extensively explored. For instance, some further structures beyond monotonicity such as weak solution were exploited for MVIs in (Song et al., 2020), a pseudo-monotonicity was used in (Boţ et al., 2021; Kannan & Shanbhag, 2019) for stochastic VIPs, a two-sided Polyak-Łojasiewicz condition was extended to VIP in (Yang et al., 2020) to tackle a class on nonconvex-nononcave minimax problems, an expected co-coercivity was used (Loizou et al., 2021), and a strongly star-monotone was further exploited in (Gorbunov et al., 2022a). While these structures are occasionally used in different works, the relation between them is still largely elusive. In addition, their relation to concrete applications is still not well studied.

(b) **Further discussion on stochastic methods.** Under the monotonicity, several authors have exploited the stochastic approximation approach (Robbins & Monro, 1951) to develop stochastic variants for solving (NE) and (NI) and their special cases. For example, a stochastic Mirror-Prox was proposed in (Juditsky et al., 2011), which has convergence on a gap function, but requires a bounded domain assumption. This approach was later extended to the extragradient method under additional assumptions in (Mishchenko et al., 2020). In (Hsieh et al., 2019), the authors discussed several methods for solving MVIs, a special case of (NI), including stochastic methods. They experimented on numerical examples and showed that the norm of the operator can asymptotically converge for unconstrained MVIs with a double learning rate. In the last few years, there were many works focusing on developing stochastic methods for solving (NE) and (NI), and their special cases using different techniques such as single-call stochastic schemes in (Hsieh et al., 2019), non-accelerated and accelerated variance reduction with Halpern-type iterations in (Cai et al., 2024; 2022), co-coercive structures in (Beznosikov et al., 2023), and bilinear game models in (Li et al., 2022).

(c) **The challenge of using biased estimators in algorithms for solving** (NI) **and related problems.** In optimization, especially in nonconvex optimization, stochastic methods using biased estimators such as SARAH, Nested SVRG, Hybrid-SGD, and STORM can achieve better, even "optimal" oracle complexity compared to unbiased ones such as standard SVRG and SAGA, see, e.g., (Cutkosky & Orabona, 2019; Driggs et al., 2022; Pham et al., 2020; Tran-Dinh et al., 2022). However,

it becomes challenging in root-finding algorithms for solving (NI) and related problems such as minimax optimization and VIPs. One main reason for this is that the convergence analysis of these optimization methods relies on the objective function as a key metric to prove convergence guarantee and to estimate oracle complexity. However, in (NI) and (NE), we do not have such an object, making it difficult to process the biased terms, including product terms such as $\langle e^k, x^{k+1} - x^\star \rangle$ and $\langle e^k, x^{k+1} - x^k \rangle$, where $e^k$ is a bias rendered from the underlying stochastic estimator. This is currently one of the main obstacles to move from using unbiased to biased estimators in root-finding algorithms, including our methods in this paper.

(d) **Comparison to (Cai et al., 2024).** Among many existing works, perhaps, (Cai et al., 2024) is one of the most recent works that develops variance-reduction methods for solving (NI) and achieves the state-of-the-art oracle complexity. However, (Cai et al., 2024) explores a different approach than ours, which relies on some recent development of the Halpern fixed-point iteration and a biased SARAH estimator. Let us clarify the differences of this work and our paper here. Algorithm 1 in (Cai et al., 2024) is a single-loop and achieves a better oracle complexity. However, it requires a much stronger assumption, Assumption 3, which is a co-coercive condition. Note that this assumption excludes the well-known bilinear matrix game, or the synthetic WGAN model (37) below. Section 4 of (Cai et al., 2024) studies both the monotone and the co-hypomonotone cases of (NI). The main idea is to reformulate (NI) into a resolvent equation $J_{\eta(G+T)}x = 0$ and then apply a deterministic variant of Algorithm 1 to this equation, where $J_{\eta(G+T)}$ is co-coercive. However, exactly evaluating $J_{\eta(G+T)}$ is impractical, one needs to approximate it by an appropriate algorithm. For instance, (Cai et al., 2024) suggests to use the variance-reduced FRBS method in (Alacaoglu et al., 2023) to approximate this resolvent, leading to a double loop algorithm. Note that this method also relies on a unbiased estimator, namely SVRG. This approach is not a direct variance-reduced method (i.e., the inner loop can be any algorithm) as ours or Algorithm 1 of (Cai et al., 2024). Moreover, practically implementing as well as rigorously analyzing an inexact double loop algorithm, when the inner loop is also a stochastic method, is often very challenging and technical as it is difficult to conduct a stopping criterion of the inner loop, and to select appropriate parameters. Nevertheless, our algorithms developed in this paper are simple to implement and applicable to both (NE) and (NI) whose weak-Minty solution exists. These problems are broader than the ones in (Cai et al., 2024). We also believe that our oracle complexity in this paper can be further improved by exploiting enhancement techniques such as nested trick or multiple loops as done in (Cai et al., 2024; Zhou et al., 2018).

(e) **Randomized coordinate and cyclic coordinate methods for (NE) and (NI).** Together with stochastic algorithms for solving (NE) and (NI) and their special cases, randomized coordinate methods have also been proposed to solve these problems, including (Combettes & Eckstein, 2018; Combettes & Pesquet, 2015; Peng et al., 2016). Recent works on randomized coordinate and cyclic coordinate methods can be found, e.g., in (Chakrabarti et al., 2024; Cui & Shanbhag, 2021; Hamedani et al., 2018; Song & Diakonikolas, 2023; Tran-Dinh & Luo, 2025; Yousefian et al., 2018). These methods are not directly related to our work, but they can be considered as a dual form of stochastic methods in certain settings such as convex-concave minimax problems. Studying relations between randomized coordinate methods and stochastic algorithms for (NE) and (NI) appears to be an interesting research topic.

### A.2. A Nonmonotone Example (NI)

As an example of (NI), we can consider the following linear operators

$$Gx := \mathbf{G}x + \mathbf{g} \quad \text{and} \quad Tx := \mathbf{T}x,$$

where $\mathbf{G}$ and $\mathbf{T}$ are given square matrices in $\mathbb{R}^{p \times p}$ and $\mathbf{g} \in \mathbb{R}^p$ is a given vector. Clearly, for any $\mathbf{G}$, $G$ is $L$-Lipschitz continuous with $L := \|\mathbf{G}\|$ (the operator norm of $\mathbf{G}$). Our goal is to choose $\mathbf{G}$ and $\mathbf{T}$ such that $\Psi x := Gx + Tx$ is nonmonotone and satisfies Assumption 1.4.

- Clearly, we can choose $\mathbf{G}$ and $\mathbf{T}$ such that $\frac{1}{2}(\mathbf{G} + \mathbf{G}^\top)$ is positive semidefinite and $\frac{1}{2}(\mathbf{G} + \mathbf{G}^\top + \mathbf{T} + \mathbf{T}^\top)$ is not positive semidefinite. This shows that $\Psi$ is nonmonotone.
- Now, Assumption 1.4 holds if $\Psi$ is $\kappa$-co-hypomonotone, i.e. there exists $\kappa > 0$ such that $\langle u - v, x - y \rangle \geq -\kappa \|u - v\|^2$, for any $(x, u), (y, v) \in \text{gra}(\Psi)$. In the linear case, this condition is equivalent to

$$\mathbf{S} := \frac{1}{2}(\mathbf{G} + \mathbf{G}^\top + \mathbf{T} + \mathbf{T}^\top) + \kappa(\mathbf{G} + \mathbf{T})^\top(\mathbf{G} + \mathbf{T}) \quad \text{is positive semidefinite.}$$

  However, since $\frac{1}{2}(\mathbf{G} + \mathbf{G}^\top)$ is positive semidefinite, this condition holds if $\frac{1}{2}(\mathbf{T} + \mathbf{T}^\top) + \kappa(\mathbf{G} + \mathbf{T})^\top(\mathbf{G} + \mathbf{T})$ is positive semidefinite. In particular, $\Psi$ satisfies Assumption 1.4.

For example, given any $\epsilon > 0$, we choose

$$\mathbf{G} := \begin{bmatrix} 0 & 1 \\ 1 & 0 \end{bmatrix}, \quad \mathbf{T} := \begin{bmatrix} -\epsilon & 0 \\ 0 & 0 \end{bmatrix}, \quad \text{and} \quad \kappa := \epsilon > 0.$$

Then, it is clear that $\mathbf{G}$ is symmetric and positive semidefinite, but $\mathbf{G} + \mathbf{T}$ is symmetric and not positive semidefinite. Thus, $\Psi x = Gx + Tx$ is nonmonotone. Moreover, $Gx = \mathbf{G}x + \mathbf{g}$ is $L$-Lipschitz continuous with $L = 1$.

Next, we check Assumption 1.4. Clearly, $\mathbf{G}$ is symmetric and positive semidefinite. Moreover, since $\mathbf{G}$ and $\mathbf{T}$ are symmetric, we have

$$
\mathbf{M} := \frac{1}{2}(\mathbf{T} + \mathbf{T}^\top) + \kappa(\mathbf{G} + \mathbf{T})^\top(\mathbf{G} + \mathbf{T}) = \begin{bmatrix} -\epsilon & 0 \\ 0 & 0 \end{bmatrix} + \kappa \begin{bmatrix} -\epsilon & 1 \\ 1 & 0 \end{bmatrix}^\top \begin{bmatrix} -\epsilon & 1 \\ 1 & 0 \end{bmatrix} = \begin{bmatrix} -\epsilon + \kappa(1 + \epsilon^2) & -\kappa\epsilon \\ -\kappa\epsilon & \kappa \end{bmatrix}.
$$

If we choose $\kappa = \epsilon$, then one can easily check that $\mathbf{M}$ is positive semidefinite. Hence, we can conclude that $\Psi$ is $\kappa$-co-hypomonotone with $\kappa = \epsilon > 0$. In particular, $\Psi$ satisfies Assumption 1.4. In addition, we can choose $\epsilon$ sufficiently small such that $L\kappa$ is sufficiently small, which fulfills the condition $L\kappa \leq \delta$ in Theorem 4.1.

## B. The Proof of Technical Results in Section 2

This supplementary section provides the full proof of Lemma 2.2 and Lemma 2.3.

**Further discussion of the FR quantity.** Let us recall our quantity $S_\gamma^k$ defined by (FRQ) as follows:

$$
S_\gamma^k := Gx^k - \gamma Gx^{k-1}. \tag{FRO}
$$

As we mentioned earlier, $\gamma$ plays a crucial role in our methods as $\gamma \in \left(\frac{1}{2}, 1\right)$. If $\gamma = \frac{1}{2}$, then we can write $S_{1/2}^k = \frac{1}{2}Gx^k + \frac{1}{2}(Gx^k - Gx^{k-1}) = \frac{1}{2}[2Gx^k - Gx^{k-1}]$ used in both the forward-reflected-backward splitting (FRBS) method (Malitsky & Tam, 2020) and the optimistic gradient method (Daskalakis et al., 2018).

Note that if we write $Gx^k - Gx^{k-1} = \hat{J}_G(x^k)(x^k - x^{k-1})$ by the Mean-Value Theorem, where $\hat{J}_G(x^k) := \int_0^1 \nabla G(x^{k-1} + \tau(x^k - x^{k-1}))d\tau$, then $S_\gamma^k = (1 - \gamma)G(x^k) + \gamma\hat{J}_G(x^k)(x^k - x^{k-1})$. Clearly, if $\gamma$ is small, then $S_\gamma^k$ can be considered as an approximation of $Gx^k$ augmented by a second-order correction term $\gamma\hat{J}_G(x^k)(x^k - x^{k-1})$ (called Hessian-driven damping term or second-order dissipative term) widely used in dynamical systems for convex optimization, see, e.g., (Adly & Attouch, 2021; Attouch & Cabot, 2020). These two viewpoints motivate the use of our new operator $S_\gamma^k$, not only in our (VFR) and (VFRBS), but in other methods such as accelerated algorithms. Thus, the results in Section 2 are of independent interest, and can potentially be used to develop other methods.

**Other possible stochastic estimators for $S_\gamma^k$.** One natural idea to construct an unbiased estimator for $S^k$ is to use an increasing mini-batch stochastic estimator as $\widetilde{S}_\gamma^k := \frac{1}{b_k}\sum_{i \in \mathcal{B}_k}[G_ix^k - \gamma G_ix^{k-1}]$, where $\mathcal{B}_k$ is an increasing mini-batch in $[n]$, with $b_k := |\mathcal{B}_k| \geq \frac{b_{k-1}}{1 - \rho_k} \geq b_{k-1}$, see, e.g., (Iusem et al., 2017). While this idea may work well for the general expectation case $Gx = \mathbb{E}_\xi\big[\mathbf{G}(x, \xi)\big]$, it may not be an ideal choice for the finite-sum operator (1) as $b_k \leq n$, which requires to stop increasing after finite iterations (i.e. $\mathcal{O}\left(\frac{\ln(n)}{-\ln(1-\rho)}\right)$ iterations). Other stochastic approximations may also fall into our class in Definition 2.1 such as JacSketch (Gower et al., 2021), SEGA (Hanzely et al., 2018), and quantized and compressed estimators ( see, e.g., (Horváth et al., 2023)).

### B.1. Proof of Lemma 2.2: Loopless-SVRG Estimator

Let us further expand Lemma 2.2 in detail as follows and then provide its full proof.

**Lemma B.1.** *Let $S_\gamma^k := Gx^k - \gamma Gx^{k-1}$ be defined by (FRQ) and $\widetilde{S}_\gamma^k$ be generated by (L-SVRG). We define*

$$
\Delta_k := \frac{1}{nb}\sum_{i=1}^n \mathbb{E}\big[\|G_ix^k - \gamma G_ix^{k-1} - (1-\gamma)G_iw^k\|^2\big]. \tag{17}
$$

*Then, we have*

$$
\begin{aligned}
\mathbb{E}_k\big[\widetilde{S}_\gamma^k\big] &= S_\gamma^k \equiv Gx^k - \gamma Gx^{k-1}, \\
\mathbb{E}\big[\|\widetilde{S}_\gamma^k - S_\gamma^k\|^2\big] &\leq \Delta_k - \frac{1}{b}\mathbb{E}\big[\|Gx^k - \gamma Gx^{k-1} - (1-\gamma)Gw^k\|^2\big] \leq \Delta_k, \\
\Delta_k &\leq \left(1 - \frac{\mathbf{p}}{2}\right)\Delta_{k-1} + \frac{(4 - 6\mathbf{p} + 3\mathbf{p}^2)}{nb\mathbf{p}}\sum_{i=1}^n \mathbb{E}\big[\|G_ix^k - G_ix^{k-1}\|^2\big] \\
&\quad + \frac{2\gamma^2(2 - 3\mathbf{p} + \mathbf{p}^2)}{nb\mathbf{p}}\sum_{i=1}^n \mathbb{E}\big[\|G_ix^{k-1} - G_ix^{k-2}\|^2\big].
\end{aligned} \tag{18}
$$

*Consequently, the SVRG estimator $\widetilde{S}_\gamma^k$ constructed by (L-SVRG) satisfies Definition 2.1 with $\Delta_k$ in (17), $\rho := \frac{\mathbf{p}}{2} \in (0, 1]$, $C := \frac{4 - 6\mathbf{p} + 3\mathbf{p}^2}{b\mathbf{p}}$, and $\hat{C} := \frac{4\gamma^2(2 - 3\mathbf{p} + \mathbf{p}^2)}{b\mathbf{p}}$.*

*Proof.* It is well-known, see, e.g., (Johnson & Zhang, 2013), that $\widetilde{S}_\gamma^k$ is an unbiased estimator of $S^k$ conditioned on $\mathcal{F}_k$, we have $\mathbb{E}_k\big[\widetilde{S}_\gamma^k\big] = S_\gamma^k$.

Next, let $X_i := G_i x^k - \gamma G_i x^{k-1} - (1-\gamma)G_i w^k$ for any $i \in [n]$. Then, we have $\mathbb{E}_k\big[X_i\big] = Gx^k - \gamma Gx^{k-1} - (1-\gamma)Gw^k$ for any $i \in [n]$. Since $\mathcal{B}_k$ is in $\mathcal{F}_k$, using the property of expectation, we can derive

$$
\begin{aligned}
\mathbb{E}_k\big[\|\widetilde{S}_\gamma^k - S_\gamma^k\|^2\big] &\overset{\text{(L-SVRG)}}{=} \mathbb{E}_k\big[\|\tfrac{1}{b}\sum_{i\in\mathcal{B}_k} X_i - [Gx^k + \gamma Gx^{k-1} - (1-\gamma)Gw^k]\|^2\big]\\
&= \mathbb{E}_k\big[\|\tfrac{1}{b}\sum_{i\in\mathcal{B}_k}\big[X_i - \mathbb{E}_k\big[X_i\big]\big]\|^2\big]\\
&\overset{①}{=} \tfrac{1}{b^2}\mathbb{E}_k\big[\sum_{i\in\mathcal{B}_k}\|X_i - \mathbb{E}_k\big[X_i\big]\|^2\big]\\
&\overset{②}{=} \tfrac{1}{b^2}\mathbb{E}_k\big[\sum_{i\in\mathcal{B}_k}\|G_i x^k - \gamma G_i x^{k-1} - (1-\gamma)G_i w^k\|^2\big] - \tfrac{1}{b}\big[\mathbb{E}_k\big[X_i\big]\big]^2\\
&= \tfrac{1}{nb}\sum_{i=1}^n\|G_i x^k - \gamma G_i x^{k-1} - (1-\gamma)G_i w^k\|^2 - \tfrac{1}{b}\big[\mathbb{E}_k\big[X_i\big]\big]^2.
\end{aligned}
$$

Here, ① holds due to the i.i.d. property of $\mathcal{B}_k$, and ② holds since $\mathbb{E}_k\big[\|X_i - \mathbb{E}_k\big[X_i\big]\|^2\big] = \mathbb{E}_k\big[\|X_i\|^2\big] - \big(\mathbb{E}_k\big[X_i\big]\big)^2$. This estimate implies the second line of (18) by taking the total expectation $\mathbb{E}\big[\cdot\big]$ both sides and the definition of $\Delta_k$ from (17).

Now, from (4) and (17), we can show that

$$
\begin{aligned}
\Delta_k &\overset{(17)}{:=} \tfrac{1}{nb}\sum_{i=1}^n\mathbb{E}\big[\|G_i x^k - \gamma G_i x^{k-1} - (1-\gamma)G_i w^k\|^2\big]\\
&\overset{(4)}{=} \tfrac{(1-\mathbf{p})}{nb}\sum_{i=1}^n\mathbb{E}\big[\|G_i x^k - \gamma G_i x^{k-1} - (1-\gamma)G_i w^{k-1}\|^2\big]\\
&\quad + \tfrac{\mathbf{p}}{nb}\sum_{i=1}^n\mathbb{E}\big[\|G_i x^k - \gamma G_i x^{k-1} - (1-\gamma)G_i x^{k-1}\|^2\big]\\
&\overset{①}{\leq} \tfrac{(1+c)(1-\mathbf{p})}{nb}\sum_{i=1}^n\mathbb{E}\big[\|G_i x^{k-1} - \gamma G_i x^{k-2} - (1-\gamma)G_i w^{k-1}\|^2\big]\\
&\quad + \tfrac{(1+c)(1-\mathbf{p})}{cnb}\sum_{i=1}^n\mathbb{E}\big[\|G_i x^k - \gamma G_i x^{k-1} - [G_i x^{k-1} - \gamma G_i x^{k-2}]\|^2\big]\\
&\quad + \tfrac{\mathbf{p}}{nb}\sum_{i=1}^n\mathbb{E}\big[\|G_i x^k - G_i x^{k-1}\|^2\big]\\
&\overset{②}{\leq} \tfrac{(1+c)(1-\mathbf{p})}{nb}\sum_{i=1}^n\mathbb{E}\big[\|G_i x^{k-1} - \gamma G_i x^{k-2} - (1-\gamma)G_i w^{k-1}\|^2\big]\\
&\quad + \tfrac{2(1+c)(1-\mathbf{p})\gamma^2}{nbc}\sum_{i=1}^n\mathbb{E}\big[\|G_i x^{k-1} - G_i x^{k-2}\|^2\big]\\
&\quad + \tfrac{1}{nb}\big[\mathbf{p} + \tfrac{2(1+c)(1-\mathbf{p})}{c}\big]\sum_{i=1}^n\mathbb{E}\big[\|G_i x^k - G_i x^{k-1}\|^2\big]\\
&= (1+c)(1-\mathbf{p})\Delta_{k-1} + \tfrac{2(1+c)(1-\mathbf{p})\gamma^2}{nbc}\sum_{i=1}^n\mathbb{E}\big[\|G_i x^{k-1} - G_i x^{k-2}\|^2\big]\\
&\quad + \tfrac{1}{nb}\big[\mathbf{p} + \tfrac{2(1+c)(1-\mathbf{p})}{c}\big]\sum_{i=1}^n\mathbb{E}\big[\|G_i x^k - G_i x^{k-1}\|^2\big].
\end{aligned}
$$

Here, in both inequalities ① and ②, we have used Young's inequality twice. If we choose $c := \frac{\mathbf{p}}{2(1-\mathbf{p})}$, then $(1+c)(1-\mathbf{p}) = 1 - \frac{\mathbf{p}}{2}$, $\frac{(1+c)(1-\mathbf{p})}{c} = (1-\mathbf{p})\big(1 + \frac{2(1-\mathbf{p})}{\mathbf{p}}\big) = \frac{(2-\mathbf{p})(1-\mathbf{p})}{\mathbf{p}} = \frac{2-3\mathbf{p}+\mathbf{p}^2}{\mathbf{p}}$, and $\frac{2(1+c)(1-\mathbf{p})}{c} + \mathbf{p} = \frac{4-6\mathbf{p}+3\mathbf{p}^2}{\mathbf{p}}$. Hence, we obtain

$$
\begin{aligned}
\Delta_k &\leq \big(1 - \tfrac{\mathbf{p}}{2}\big)\Delta_{k-1} + \tfrac{(4-6\mathbf{p}+3\mathbf{p}^2)}{nb\mathbf{p}}\sum_{i=1}^n\mathbb{E}\big[\|G_i x^k - G_i x^{k-1}\|^2\big]\\
&\quad + \tfrac{2\gamma^2(2-3\mathbf{p}+\mathbf{p}^2)}{nb\mathbf{p}}\sum_{i=1}^n\mathbb{E}\big[\|G_i x^{k-1} - G_i x^{k-2}\|^2\big].
\end{aligned}
$$

This is exactly the last inequality of (18). $\qquad\square$

### B.2. Proof of Lemma 2.3: SAGA estimator

Similarly, we also further expand Lemma 2.3 in detail as follows and then provide its full proof.

**Lemma B.2.** *Let $S_\gamma^k := Gx^k - \gamma Gx^{k-1}$ be defined by* (FRQ) *and $\widetilde{S}_\gamma^k$ be generated by the SAGA estimator* (SAGA), *and $e^k := \widetilde{S}_\gamma^k - S_\gamma^k$. We consider the following quantity:*

$$
\Delta_k := \tfrac{1}{nb}\sum_{i=1}^n\mathbb{E}\big[\|G_i x^k - \gamma G_i x^{k-1} - (1-\gamma)\hat{G}_i^k\|^2\big]. \tag{19}
$$

*Then, we have*

$$
\begin{aligned}
\mathbb{E}_k\big[\widetilde{S}_\gamma^k\big] &= S_\gamma^k \equiv Gx^k - \gamma Gx^{k-1}, \\
\mathbb{E}\big[\|\widetilde{S}_\gamma^k - S_\gamma^k\|^2\big] &\leq \Delta_k - \tfrac{1}{b}\mathbb{E}\big[\|Gx^k - \gamma Gx^{k-1} - \tfrac{(1-\gamma)}{n}\sum_{i=1}^n \hat{G}_i^k\|^2\big] \leq \Delta_k, \\
\Delta_k &\leq \big(1 - \tfrac{b}{2n}\big)\Delta_{k-1} + \tfrac{[2(n-b)(2n+b)+b^2]}{n^2 b^2}\sum_{i=1}^n \mathbb{E}\big[\|G_i x^k - G_i x^{k-1}\|^2\big] \\
&\quad + \tfrac{2(n-b)(2n+b)\gamma^2}{n^2 b^2}\sum_{i=1}^n \mathbb{E}\big[\|G_i x^{k-1} - G_i x^{k-2}\|^2\big].
\end{aligned}
\tag{20}
$$

*Consequently, the SAGA estimator $\widetilde{S}_\gamma^k$ constructed by (SAGA) satisfies Definition 2.1 with $\Delta_k$ in (19), $\rho := \tfrac{b}{2n} \in (0,1]$, $C := \tfrac{[2(n-b)(2n+b)+b^2]}{nb^2}$, and $\hat{C} := \tfrac{2(n-b)(2n+b)\gamma^2}{nb^2}$.*

*Proof.* It is well-known, see, e.g., (Defazio et al., 2014), that $\widetilde{S}_\gamma^k$ defined by (SAGA) is an unbiased estimator of $S^k$. Indeed, we have $\mathbb{E}_k\big[\hat{G}_{\mathcal{B}_k}^k\big] = \tfrac{1}{n}\sum_{i=1}^n \hat{G}_i^k$, $\mathbb{E}_k\big[G_{\mathcal{B}_k} x^k\big] = Gx^k$, and $\mathbb{E}_k\big[G_{\mathcal{B}_k} x^{k-1}\big] = Gx^{k-1}$. Using these relations and the definition of $\widetilde{S}^k$, we have

$$
\begin{aligned}
\mathbb{E}_k\big[\widetilde{S}^k\big] &= \mathbb{E}_k\big[\tfrac{(1-\gamma)}{n}\sum_{i=1}^n \hat{G}_i^k\big] - (1-\gamma)\mathbb{E}_k\big[\hat{G}_{\mathcal{B}_k}^k\big] + \mathbb{E}_k\big[G_{\mathcal{B}_k} x^k\big] - \gamma\mathbb{E}_k\big[G_{\mathcal{B}_k} x^{k-1}\big] \\
&= \tfrac{(1-\gamma)}{n}\sum_{i=1}^n \hat{G}_i^k - \tfrac{(1-\gamma)}{n}\sum_{i=1}^n \hat{G}_i^k + Gx^k - \gamma Gx^{k-1} \\
&= Gx^k - \gamma Gx^{k-1} \\
&= S^k.
\end{aligned}
$$

Hence, $\widetilde{S}^k$ is an unbiased estimator of $S^k$.

Next, let $X_i := G_i x^k - \gamma G_i x^{k-1} - (1-\gamma)\hat{G}_i^k$ for any $i \in [n]$. Then, we have $\mathbb{E}_k\big[X_i\big] = Gx^k - \gamma Gx^{k-1} - \tfrac{(1-\gamma)}{n}\sum_{i=1}^n \hat{G}_i^k$ for any $i \in [n]$. Therefore, we can derive

$$
\begin{aligned}
\mathbb{E}_k\big[\|\widetilde{S}_\gamma^k - S_\gamma^k\|^2\big] &= \mathbb{E}_k\big[\|\tfrac{1}{b}\sum_{i\in\mathcal{B}_k} X_i - [Gx^k + \gamma Gx^{k-1} - \tfrac{(1-\gamma)}{n}\sum_{i=1}^n \hat{G}_i^k]\|^2\big] \\
&= \mathbb{E}_k\big[\|\tfrac{1}{b}\sum_{i\in\mathcal{B}_k} X_i - \mathbb{E}_k[X_i]\|^2\big] \\
&= \tfrac{1}{b^2}\mathbb{E}_k\big[\sum_{i\in\mathcal{B}_k} \|X_i - \mathbb{E}_k[X_i]\|^2\big] \\
&= \tfrac{1}{b^2}\mathbb{E}_k\big[\sum_{i\in\mathcal{B}_k} \|G_i x^k - \gamma G_i x^{k-1} - (1-\gamma)\hat{G}_i^k\|^2\big] - \tfrac{1}{b}\big[\mathbb{E}_k[X_i]\big]^2 \\
&= \tfrac{1}{nb}\sum_{i=1}^n \|G_i x^k - \gamma G_i x^{k-1} - (1-\gamma)\hat{G}_i^k\|^2 - \tfrac{1}{b}\big[\mathbb{E}_k[X_i]\big]^2.
\end{aligned}
$$

This implies the second line of (20) by taking the total expectation $\mathbb{E}\big[\cdot\big]$ both sides.

Now, from (5) and (19) and the rule (5), for any $c > 0$, by Young's inequality, we can show that

$$
\begin{aligned}
\Delta_k &\overset{(19)}{:=} \tfrac{1}{nb}\sum_{i=1}^n \mathbb{E}\big[\|G_i x^k - \gamma G_i x^{k-1} - (1-\gamma)\hat{G}_i^k\|^2\big] \\
&\overset{(5)}{=} \big(1 - \tfrac{b}{n}\big)\tfrac{1}{nb}\sum_{i=1}^n \mathbb{E}\big[\|G_i x^k - \gamma G_i x^{k-1} - (1-\gamma)\hat{G}_i^{k-1}\|^2\big] \\
&\quad + \tfrac{b}{n}\cdot\tfrac{1}{nb}\sum_{i=1}^n \mathbb{E}\big[\|G_i x^k - \gamma G_i x^{k-1} - (1-\gamma)G_i x^{k-1}\|^2\big] \\
&\leq \tfrac{(1+c)}{nb}\big(1 - \tfrac{b}{n}\big)\sum_{i=1}^n \mathbb{E}\big[\|G_i x^{k-1} - \gamma G_i x^{k-2} - (1-\gamma)\hat{G}_i^{k-1}\|^2\big] \\
&\quad + \tfrac{(1+c)}{cnb}\big(1 - \tfrac{b}{n}\big)\sum_{i=1}^n \mathbb{E}\big[\|G_i x^k - \gamma G_i x^{k-1} - (G_i x^{k-1} - \gamma G_i x^{k-2})\|^2\big] \\
&\quad + \tfrac{1}{n^2}\sum_{i=1}^n \mathbb{E}\big[\|G_i x^k - G_i x^{k-1}\|^2\big] \\
&\leq (1+c)\big(1 - \tfrac{b}{n}\big)\Delta_{k-1} + \big[\tfrac{1}{n^2} + \big(1 - \tfrac{b}{n}\big)\tfrac{2(1+c)}{cnb}\big]\sum_{i=1}^n \mathbb{E}\big[\|G_i x^k - G_i x^{k-1}\|^2\big] \\
&\quad + \tfrac{2(1+c)\gamma^2}{cnb}\big(1 - \tfrac{b}{n}\big)\sum_{i=1}^n \mathbb{E}\big[\|G_i x^{k-1} - G_i x^{k-2}\|^2\big].
\end{aligned}
$$

If we choose $c := \tfrac{b}{2n} \in (0,1)$, then $(1 - \tfrac{b}{n})(1 + c) = 1 - \tfrac{b}{2n} - \tfrac{b^2}{2n^2} \leq 1 - \tfrac{b}{2n}$. Hence, we can further upper bound the last inequality as

$$
\begin{aligned}
\Delta_k &\leq \big(1 - \tfrac{b}{2n}\big)\Delta_{k-1} + \tfrac{[2(n-b)(2n+b)+b^2]}{n^2 b^2}\sum_{i=1}^n \mathbb{E}\big[\|G_i x^k - G_i x^{k-1}\|^2\big] \\
&\quad + \tfrac{2(n-b)(2n+b)\gamma^2}{n^2 b^2}\sum_{i=1}^n \mathbb{E}\big[\|G_i x^{k-1} - G_i x^{k-2}\|^2\big].
\end{aligned}
$$

This is exactly the last inequality of (20). $\qquad\square$

## C. Convergence Analysis of VFR for (NE): Technical Proofs

To analyze our (VFR) scheme, we introduce the following two functions:

$$
\begin{aligned}
\mathcal{L}_k &:= \|x^k + \gamma\eta Gx^{k-1} - x^\star\|^2 + \mu\|x^k - x^{k-1}\|^2, \\
\mathcal{E}_k &:= \mathcal{L}_k + \frac{\eta^2(1+\mu)(1-\rho)}{\rho}\Delta_{k-1} + \frac{L^2\eta^2\hat{C}(1+\mu)}{\rho}\|x^{k-1} - x^{k-2}\|^2,
\end{aligned}
\tag{21}
$$

where $\mu$ is a given positive parameter, $\rho$, $C$, $\hat{C}$, and $\Delta_k$ are given in Definition 2.1, and $x^{-2} = x^{-1} = x^0$. Clearly, we have $\mathcal{L}_k \geq 0$ and $\mathcal{E}_k \geq 0$ for all $k \geq 0$ a.s.

One key step to analyze the convergence of (VFR) is to prove a descent property of $\mathcal{E}_k$ defined by (21). The following lemma provides such a key estimate to prove the convergence of (VFR).

**Lemma C.1.** *Suppose that Assumptions 1.3 and 1.4 hold for (NE). Let $\{x^k\}$ be generated by (VFR) and $\mathcal{E}_k$ be defined by (21) for any $\gamma \in [0,1]$. Then, with $M := \frac{\gamma(1+\mu-\gamma)}{\mu} + \frac{(1+\mu)(C+\hat{C})}{\mu\rho}$, we have*

$$
\begin{aligned}
\mathbb{E}\big[\mathcal{E}_k\big] - \mathbb{E}\big[\mathcal{E}_{k+1}\big] &\geq \mu\left(1 - M \cdot L^2\eta^2\right)\mathbb{E}\big[\|x^k - x^{k-1}\|^2\big] \\
&\quad + \eta(1-\gamma)\big[\eta(2\gamma - 1 - \mu) - 2\kappa\big]\mathbb{E}\big[\|Gx^k\|^2\big] \\
&\quad + \eta^2\gamma(1-\gamma)(1+\mu)\mathbb{E}\big[\|Gx^{k-1}\|^2\big].
\end{aligned}
\tag{22}
$$

*Proof.* First, using $x^{k+1} := x^k - \eta\widetilde{S}_\gamma^k$ from (VFR), we can expand

$$
\begin{aligned}
\|x^{k+1} + \gamma\eta Gx^k - x^\star\|^2 &\overset{\text{(VFR)}}{=} \|x^k - x^\star + \gamma\eta Gx^k - \eta\widetilde{S}_\gamma^k\|^2 \\
&= \|x^k - x^\star\|^2 + 2\gamma\eta\langle Gx^k, x^k - x^\star\rangle + \gamma^2\eta^2\|Gx^k\|^2 \\
&\quad - 2\eta\langle\widetilde{S}_\gamma^k, x^k - x^\star\rangle - 2\gamma\eta^2\langle Gx^k, \widetilde{S}_\gamma^k\rangle + \eta^2\|\widetilde{S}_\gamma^k\|^2.
\end{aligned}
$$

Second, it is obvious to show that

$$
\|x^k + \gamma\eta Gx^{k-1} - x^\star\|^2 = \|x^k - x^\star\|^2 + 2\gamma\eta\langle Gx^{k-1}, x^k - x^\star\rangle + \gamma^2\eta^2\|Gx^{k-1}\|^2.
$$

Third, using again $x^{k+1} := x^k - \eta\widetilde{S}_\gamma^k$ from (VFR), we can show that

$$
\|x^{k+1} - x^k\|^2 = \eta^2\|\widetilde{S}_\gamma^k\|^2.
$$

Combining three expressions above, and using $\mathcal{L}_k$ from (21), we can establish that

$$
\begin{aligned}
\mathcal{L}_k - \mathcal{L}_{k+1} &= \|x^k + \gamma\eta Gx^{k-1} - x^\star\|^2 - \|x^{k+1} + \gamma\eta Gx^k - x^\star\|^2 \\
&\quad + \mu\|x^k - x^{k-1}\|^2 - \mu\|x^{k+1} - x^k\|^2 \\
&= 2\gamma\eta\langle Gx^{k-1}, x^k - x^\star\rangle - 2\gamma\eta\langle Gx^k, x^k - x^\star\rangle + \gamma^2\eta^2\|Gx^{k-1}\|^2 \\
&\quad - \gamma^2\eta^2\|Gx^k\|^2 + 2\eta\langle\widetilde{S}_\gamma^k, x^k - x^\star\rangle + 2\gamma\eta^2\langle Gx^k, \widetilde{S}_\gamma^k\rangle \\
&\quad + \mu\|x^k - x^{k-1}\|^2 - \eta^2(1+\mu)\|\widetilde{S}_\gamma^k\|^2.
\end{aligned}
\tag{23}
$$

Next, since $\mathbb{E}_k\big[\widetilde{S}_\gamma^k\big] = S_\gamma^k \equiv Gx^k - \gamma Gx^{k-1}$ as shown in the first line of (3) of Definition 2.1. Moreover, since $\widetilde{S}_\gamma^k$ is conditionally independent of $x^k - x^\star$ and $Gx^k$ w.r.t. the $\sigma$-field $\mathcal{F}_k$, we have

$$
\begin{aligned}
\mathbb{E}_k\big[\langle\widetilde{S}_\gamma^k, x^k - x^\star\rangle\big] &= \langle Gx^k, x^k - x^\star\rangle - \gamma\langle Gx^{k-1}, x^k - x^\star\rangle, \\
2\mathbb{E}_k\big[\langle\widetilde{S}_\gamma^k, Gx^k\rangle\big] &= 2\|Gx^k\|^2 - 2\gamma\langle Gx^{k-1}, Gx^k\rangle \\
&= (2-\gamma)\|Gx^k\|^2 - \gamma\|Gx^{k-1}\|^2 + \gamma\|Gx^k - Gx^{k-1}\|^2.
\end{aligned}
$$

Taking the conditional expectation $\mathbb{E}_k[\,\cdot\,]$ both sides of (23) and using the last two expressions, we can show that

$$
\begin{aligned}
\mathcal{L}_k - \mathbb{E}_k\big[\mathcal{L}_{k+1}\big] &= 2\gamma\eta\langle Gx^{k-1}, x^k - x^\star\rangle - 2\gamma\eta\langle Gx^k, x^k - x^\star\rangle + \gamma^2\eta^2\|Gx^{k-1}\|^2 \\
&\quad - \gamma^2\eta^2\|Gx^k\|^2 + 2\eta\mathbb{E}_k\big[\langle \widetilde{S}_\gamma^k, x^k - x^\star\rangle\big] + 2\gamma\eta^2\mathbb{E}_k\big[\langle Gx^k, \widetilde{S}_\gamma^k\rangle\big] \\
&\quad - \eta^2(1+\mu)\mathbb{E}_k\big[\|\widetilde{S}_\gamma^k\|^2\big] + \mu\|x^k - x^{k-1}\|^2 \\
&= 2\eta(1-\gamma)\langle Gx^k, x^k - x^\star\rangle + 2\gamma(1-\gamma)\eta^2\|Gx^k\|^2 \\
&\quad + \gamma^2\eta^2\|Gx^k - Gx^{k-1}\|^2 - \eta^2(1+\mu)\mathbb{E}_k\big[\|\widetilde{S}_\gamma^k\|^2\big] + \mu\|x^k - x^{k-1}\|^2.
\end{aligned}
$$

Since $\widetilde{S}_\gamma^k$ is an unbiased estimator of $S_\gamma^k$, if we denote $e^k := \widetilde{S}_\gamma^k - S_\gamma^k$, then we have $\mathbb{E}_k\big[e^k\big] = 0$. Hence, we can show that $\mathbb{E}_k\big[\|\widetilde{S}_\gamma^k\|^2\big] = \mathbb{E}_k\big[\|S_\gamma^k + e^k\|^2\big] = \|S_\gamma^k\|^2 + 2\mathbb{E}_k\big[\langle e^k, S_\gamma^k\rangle\big] + \mathbb{E}_k\big[\|e^k\|^2\big] = \mathbb{E}_k\big[\|e^k\|^2\big] + \|S_\gamma^k\|^2$. Using this relation and $S_\gamma^k = Gx^k - \gamma Gx^{k-1}$, we can show that

$$
\begin{aligned}
\mathbb{E}_k\big[\|\widetilde{S}_\gamma^k\|^2\big] &= \|S_\gamma^k\|^2 + \mathbb{E}_k\big[\|e^k\|^2\big] = \|Gx^k - \gamma Gx^{k-1}\|^2 + \mathbb{E}_k\big[\|e^k\|^2\big] \\
&= \|Gx^k\|^2 - 2\gamma\langle Gx^k, Gx^{k-1}\rangle + \gamma^2\|Gx^{k-1}\|^2 + \mathbb{E}_k\big[\|e^k\|^2\big] \\
&= (1-\gamma)\|Gx^k\|^2 - \gamma(1-\gamma)\|Gx^{k-1}\|^2 + \gamma\|Gx^k - Gx^{k-1}\|^2 + \mathbb{E}_k\big[\|e^k\|^2\big].
\end{aligned}
$$

Substituting this expression into the last estimate, we can show that

$$
\begin{aligned}
\mathcal{L}_k - \mathbb{E}_k\big[\mathcal{L}_{k+1}\big] &= 2\eta(1-\gamma)\langle Gx^k, x^k - x^\star\rangle + \eta^2(1-\gamma)\big(2\gamma - 1 - \mu\big)\|Gx^k\|^2 \\
&\quad + \eta^2\gamma(1-\gamma)(1+\mu)\|Gx^{k-1}\|^2 - \gamma\eta^2(1+\mu-\gamma)\|Gx^k - Gx^{k-1}\|^2 \\
&\quad - \eta^2(1+\mu)\mathbb{E}_k\big[\|e^k\|^2\big] + \mu\|x^k - x^{k-1}\|^2.
\end{aligned}
$$

Taking the total expectation $\mathbb{E}[\,\cdot\,]$ both sides of this expression, we get

$$
\begin{aligned}
\mathbb{E}\big[\mathcal{L}_k\big] - \mathbb{E}\big[\mathcal{L}_{k+1}\big] &= 2(1-\gamma)\eta\mathbb{E}\big[\langle Gx^k, x^k - x^\star\rangle\big] + \eta^2\gamma(1-\gamma)(1+\mu)\mathbb{E}\big[\|Gx^{k-1}\|^2\big] \\
&\quad + \eta^2(1-\gamma)\big(2\gamma - 1 - \mu\big)\mathbb{E}\big[\|Gx^k\|^2\big] + \mu\mathbb{E}\big[\|x^k - x^{k-1}\|^2\big] \\
&\quad - \gamma\eta^2(1+\mu-\gamma)\mathbb{E}\big[\|Gx^k - Gx^{k-1}\|^2\big] - \eta^2(1+\mu)\mathbb{E}\big[\|e^k\|^2\big].
\end{aligned}
$$

By Young's inequality in ① and (2) of Assumption 1.3, we have

$$
\begin{aligned}
\|Gx^k - Gx^{k-1}\|^2 &= \|\tfrac{1}{n}\sum_{i=1}^n [G_i x^k - G_i x^{i-1}]\|^2 \overset{①}{\le} \tfrac{1}{n}\sum_{i=1}^n \|G_i x^k - G_i x^{k-1}\|^2 \\
&\overset{(2)}{\le} L^2\|x^k - x^{k-1}\|^2.
\end{aligned}
\tag{24}
$$

Utilizing this inequality, $\langle Gx^k, x^k - x^\star\rangle \ge -\kappa\|Gx^k\|^2$ from Assumption 1.4 with $T = 0$, and $\mathbb{E}\big[\|e^k\|^2\big] \le \Delta_k$ from (3), we can bound the last expression as

$$
\begin{aligned}
\mathbb{E}\big[\mathcal{L}_k\big] - \mathbb{E}\big[\mathcal{L}_{k+1}\big] &\ge \big[\mu - L^2\eta^2\gamma(1+\mu-\gamma)\big]\mathbb{E}\big[\|x^k - x^{k-1}\|^2\big] \\
&\quad + (1+\mu)\gamma(1-\gamma)\eta^2\mathbb{E}\big[\|Gx^{k-1}\|^2\big] \\
&\quad + \eta(1-\gamma)\big[\eta\big(2\gamma - 1 - \mu\big) - 2\kappa\big]\mathbb{E}\big[\|Gx^k\|^2\big] - \eta^2(1+\mu)\Delta_k.
\end{aligned}
\tag{25}
$$

By the third line of (3) in Definition 2.1 and again (2), we have

$$
\Delta_k \le (1-\rho)\Delta_{k-1} + CL^2\mathbb{E}\big[\|x^k - x^{k-1}\|^2\big] + \hat{C}L^2\mathbb{E}\big[\|x^{k-1} - x^{k-2}\|^2\big].
$$

Rearranging this inequality, we get

$$
\begin{aligned}
\Delta_k \le \big(\tfrac{1-\rho}{\rho}\big)\big(\Delta_{k-1} - \Delta_k\big) &+ \tfrac{\hat{C}L^2}{\rho}\big[\mathbb{E}\big[\|x^{k-1} - x^{k-2}\|^2\big] - \mathbb{E}\big[\|x^k - x^{k-1}\|^2\big]\big] \\
&+ \tfrac{(C+\hat{C})L^2}{\rho}\mathbb{E}\big[\|x^k - x^{k-1}\|^2\big].
\end{aligned}
$$

Substituting this inequality into (25), we can show that

$$
\begin{aligned}
\mathbb{E}\big[\mathcal{L}_k\big] - \mathbb{E}\big[\mathcal{L}_{k+1}\big] \geq\ & \Big[\mu - L^2\eta^2\gamma(1+\mu-\gamma) - \tfrac{L^2\eta^2(1+\mu)(C+\hat{C})}{\rho}\Big]\mathbb{E}\big[\|x^k - x^{k-1}\|^2\big] \\
& + \eta(1-\gamma)\big[\eta\big(2\gamma - 1 - \mu\big) - 2\kappa\big]\mathbb{E}\big[\|Gx^k\|^2\big] \\
& + (1+\mu)\gamma(1-\gamma)\eta^2\mathbb{E}\big[\|Gx^{k-1}\|^2\big] \\
& - \tfrac{L^2\eta^2\hat{C}(1+\mu)}{\rho}\big[\mathbb{E}\big[\|x^{k-1} - x^{k-2}\|^2\big] - \mathbb{E}\big[\|x^k - x^{k-1}\|^2\big]\big] \\
& - \tfrac{\eta^2(1+\mu)(1-\rho)}{\rho}\big(\Delta_{k-1} - \Delta_k\big).
\end{aligned}
$$

Rearranging this inequality and using $\mathcal{E}_k$ from (21), we obtain (22). $\hfill\square$

Now, we are ready to prove our first main result, Theorem 3.1 in the main text.

***Proof of Theorem 3.1.*** Let us denote by $M := \frac{\gamma(1+\mu-\gamma)}{\mu} + \frac{(1+\mu)(C+\hat{C})}{\rho\mu}$. Then, to keep the right-hand side of (22) positive, we need to choose the parameters such that $L^2\eta^2 \leq \frac{1}{M}$ and $\eta \geq \frac{2\kappa}{2\gamma-1-\mu}$. These two conditions lead to $\frac{4L^2\kappa^2}{(2\gamma-1-\mu)^2} \leq L^2\eta^2 \leq \frac{1}{M}$.

Now, for a given $\gamma \in \big(\frac{1}{2}, 1\big)$, let us choose $\mu := \frac{3(2\gamma-1)}{4} > 0$. Then, the last condition holds if $L\kappa \leq \delta := \frac{2\gamma-1}{8\sqrt{M}}$ as stated in Theorem 3.1. In this case, we have $M = \frac{\gamma(1+5\gamma)}{3(2\gamma-1)} + \frac{1+6\gamma}{3(2\gamma-1)} \cdot \frac{C+\hat{C}}{\rho}$ as stated in (6). Hence, we can choose $\frac{8\kappa}{2\gamma-1} \leq \eta \leq \frac{1}{L\sqrt{M}}$ as claimed in Theorem 3.1.

Next, utilizing $\mu + 1 = \frac{1+6\gamma}{2} \geq 1$ and $\mu = \frac{3(2\gamma-1)}{4}$, (22) reduces to

$$
\mathbb{E}\big[\mathcal{E}_k\big] - \mathbb{E}\big[\mathcal{E}_{k+1}\big] \geq \tfrac{3(2\gamma-1)}{4}\big(1 - M \cdot L^2\eta^2\big)\mathbb{E}\big[\|x^k - x^{k-1}\|^2\big] + \gamma(1-\gamma)\eta^2\mathbb{E}\big[\|Gx^{k-1}\|^2\big].
$$

Averaging this inequality from $k := 0$ to $k := K$, we obtain

$$
\begin{cases}
\frac{1}{K+1}\sum_{k=0}^{K}\mathbb{E}\big[\|Gx^{k-1}\|^2\big] & \leq \frac{\mathbb{E}[\mathcal{E}_0]}{\gamma(1-\gamma)\eta^2(K+1)}, \\[2mm]
\frac{(1-ML^2\eta^2)}{K+1}\sum_{k=0}^{K}\mathbb{E}\big[\|x^k - x^{k-1}\|^2\big] & \leq \frac{4\mathbb{E}[\mathcal{E}_0]}{3(2\gamma-1)(K+1)}.
\end{cases}
$$

Finally, since $x^{-1} = x^{-2} = x^0$, and $\widetilde{S}_\gamma^0$ is chosen as $\widetilde{S}_\gamma^0 := (1-\gamma)Gx^0$, we have $\Delta_{-1} = \Delta_0 = 0$. Using this fact, $Gx^\star = 0$, the Lipschitz continuity of $G$, $\rho \in [0, 1]$, and $\gamma < 1$, we can show that

$$
\begin{aligned}
\mathbb{E}\big[\mathcal{E}_0\big] &= \mathbb{E}\big[\|x^0 + \eta\gamma G(x^0) - x^\star\|^2\big] + \tfrac{\eta^2(1+\mu)(1-\rho)}{\rho}\Delta_0 \\
&\leq 2\mathbb{E}\big[\|x^0 - x^\star\|^2\big] + 2\eta^2\gamma^2\mathbb{E}\big[\|Gx^0 - Gx^\star\|^2\big] + \tfrac{(1+6\gamma)\eta^2}{4\rho}\Delta_0 \\
&\leq 2(1 + L^2\eta^2\gamma^2)\mathbb{E}\big[\|x^0 - x^\star\|^2\big] + \tfrac{(1+6\gamma)\eta^2}{4\rho}\Delta_0 \\
&\leq 2\big(1 + L^2\eta^2\big)\|x^0 - x^\star\|^2.
\end{aligned}
$$

Substituting this upper bound into the above estimates, we get the second bound of (7). For the first bound, we replace $k-1$ by $k$, and $K$ by $K+1$, using $\|Gx^0\|^2 \leq 2\|Gx^0\|^2$, and then multiplying both sides of the result by $\frac{K+2}{K+1}$ to obtain the first line of (7). $\hfill\square$

Next, we restate Corollary 3.2 for the case $\gamma \in \big(\frac{1}{2}, 1\big)$ instead of $\gamma = \frac{3}{4}$ as in the main text. Then, we derive the proof of Corollary 3.2 from this result by fixing $\gamma = \frac{3}{4}$.

**Corollary C.2.** *Suppose that Assumptions 1.1, 1.3, and 1.4 hold for* (NE) *with $\kappa \geq 0$ as in Theorem 3.1. Let $\{x^k\}$ be generated by* (VFR) *using the SVRG estimator* (L-SVRG), *$\gamma \in \big(\frac{1}{2}, 1\big)$, and*

$$
\eta := \tfrac{1}{L\sqrt{M}}, \quad \text{where} \quad \Lambda := \tfrac{4(1+\gamma^2)(2-3\mathbf{p})+2(3+2\gamma^2)\mathbf{p}^2}{b\mathbf{p}^2} \quad \text{and} \quad M := \tfrac{\gamma(1+5\gamma)}{3(2\gamma-1)} + \tfrac{1+6\gamma}{3(2\gamma-1)} \cdot \Lambda. \tag{26}
$$

*Then, we have $\eta \geq \frac{\sigma\sqrt{b}\mathbf{p}}{L}$ for $\sigma := \frac{\sqrt{3(2\gamma-1)}}{\sqrt{8+49\gamma+13\gamma^2+48\gamma^3}}$, and the following bound holds:*

$$\frac{1}{K+1}\sum_{k=0}^{K}\mathbb{E}\big[\|Gx^k\|^2\big] \leq \frac{2(1+L^2\eta^2)R_0^2}{\gamma(1-\gamma)\eta^2(K+1)}, \quad \text{where} \quad R_0 := \|x^0 - x^\star\|. \tag{27}$$

*For a given tolerance $\epsilon > 0$, if we choose $\mathbf{p} := n^{-1/3}$ and $b := \lfloor n^{2/3} \rfloor$, then (VFR) requires $\mathcal{T}_{G_i} := n + \left\lfloor \frac{4\Gamma L^2 R_0^2 n^{2/3}}{\epsilon^2} \right\rfloor$ evaluations of $G_i$ to achieve $\frac{1}{K+1}\sum_{k=0}^{K}\mathbb{E}\big[\|Gx^k\|^2\big] \leq \epsilon^2$, where $\Gamma := \frac{2(5\gamma^2+7\gamma-3)(8+49\gamma+13\gamma^2+48\gamma^3)}{3\gamma^2(2\gamma-1)(1-\gamma)(1+5\gamma)}$.*

*Proof.* For the SVRG estimator (L-SVRG), by Lemma 2.2, we have $\rho := \frac{\mathbf{p}}{2} \in (0, 1]$, $C := \frac{4-6\mathbf{p}+3\mathbf{p}^2}{b\mathbf{p}}$, and $\hat{C} := \frac{2\gamma^2(2-3\mathbf{p}+\mathbf{p}^2)}{b\mathbf{p}}$. Therefore, we can compute $\Lambda := \frac{C+\hat{C}}{\rho} = \frac{4(1+\gamma^2)(2-3\mathbf{p})+2(3+2\gamma^2)\mathbf{p}^2}{b\mathbf{p}^2} \leq \frac{8(1+\gamma^2)}{b\mathbf{p}^2}$, and $M$ in Theorem 3.1 as $M := \frac{\gamma(1+5\gamma)}{3(2\gamma-1)} + \frac{1+6\gamma}{3(2\gamma-1)} \cdot \Lambda \leq \frac{\gamma(1+5\gamma)}{3(2\gamma-1)} + \frac{8(1+6\gamma)(1+\gamma^2)}{3(2\gamma-1)b\mathbf{p}^2}$ as stated in (26). The estimate (27) is exactly the first line of (7).

Now, suppose that $b\mathbf{p}^2 \leq 1$. Then, by (26), we have $M \leq \frac{8+49\gamma+13\gamma^2+48\gamma^3}{3(2\gamma-1)b\mathbf{p}^2}$. Therefore, if we choose $\eta := \frac{1}{L\sqrt{M}}$, then $\eta$ satisfies the conditions of Theorem 3.1, provided that $L\rho \leq \delta$. Moreover, we have $\eta \geq \frac{\sqrt{3(2\gamma-1)}\sqrt{b}\mathbf{p}}{\sqrt{8+49\gamma+13\gamma^2+48\gamma^3}} = \frac{\sigma\sqrt{b}\mathbf{p}}{L}$, where $\sigma := \frac{\sqrt{3(2\gamma-1)}}{\sqrt{8+49\gamma+13\gamma^2+48\gamma^3}}$.

From (27), to guarantee $\frac{1}{K+1}\sum_{k=0}^{K}\mathbb{E}\big[\|Gx^k\|^2\big] \leq \epsilon^2$, we need to impose $\frac{2(1+L^2\eta^2)R_0^2}{\gamma(1-\gamma)\eta^2(K+1)} \leq \epsilon^2$, where $R_0 := \|x^0 - x^\star\|$. However, since $1 + L^2\eta^2 = 1 + \frac{1}{M} \leq \frac{5\gamma^2+7\gamma-3}{\gamma(1+5\gamma)}$ and $\eta \geq \frac{\sigma\sqrt{b}\mathbf{p}}{L}$, the last condition holds if we choose $K := \left\lceil \Gamma \cdot \frac{L^2 R_0^2}{b\mathbf{p}^2\epsilon^2} \right\rceil$, where $\Gamma := \frac{2(5\gamma^2+7\gamma-3)}{\sigma^2\gamma^2(1-\gamma)(1+5\gamma)} = \frac{2(5\gamma^2+7\gamma-3)(8+49\gamma+13\gamma^2+48\gamma^3)}{3\gamma^2(2\gamma-1)(1-\gamma)(1+5\gamma)}$.

Finally, note that, at each iteration $k$, (VFR) requires 3 mini-batches of size $b$, and occasionally compute the full batch $Gw^k$, leading to the cost of $n\mathbf{p} + 3b$. The total complexity is

$$\mathcal{T}_c := K(n\mathbf{p}+3b) = \frac{\Gamma L^2 R_0^2(n\mathbf{p}+3b)}{b\mathbf{p}^2\epsilon^2} = \frac{\Gamma L^2 R_0^2}{\epsilon^2}\left(\frac{n}{b\mathbf{p}} + \frac{3}{\mathbf{p}^2}\right).$$

If we choose $b := \lfloor n^{2/3} \rfloor$ and $\mathbf{p} := n^{-1/3}$, then $b\mathbf{p}^2 = 1$ and $\mathcal{T}_c = \frac{4\Gamma n^{2/3}L^2 R_0^2}{\epsilon^2}$. For the SVRG estimator (L-SVRG), one needs to compute $Gw^0$, which requires $n$ evaluations of $G_i$. Hence, the total complexity of the algorithm is $\mathcal{T}_{G_i} := n + \left\lfloor \frac{4\Gamma n^{2/3}L^2 R_0^2}{\epsilon^2} \right\rfloor$ as stated. $\square$

*Proof of Corollary 3.2.* Since we fix $\gamma := \frac{3}{4}$, we can easily compute $\sigma := \frac{\sqrt{3(2\gamma-1)}}{\sqrt{8+49\gamma+13\gamma^2+48\gamma^3}} \approx 0.144025 \geq 0.1440$ and $\Gamma := \frac{2(5\gamma^2+7\gamma-3)(8+49\gamma+13\gamma^2+48\gamma^3)}{3\gamma^2(2\gamma-1)(1-\gamma)(1+5\gamma)} \approx 730.736842 \leq 731$. Therefore, we obtain $\eta \geq \frac{0.1440\sqrt{b}\mathbf{p}}{L}$ and $\mathcal{T}_{G_i} := n + \left\lfloor \frac{4\Gamma n^{2/3}L^2 R_0^2}{\epsilon^2} \right\rfloor$, where $\Gamma := 731$. Moreover, (27) reduces to $\frac{1}{K+1}\sum_{k=0}^{K}\mathbb{E}\big[\|Gx^k\|^2\big] \leq \frac{32(1+0.1440^2)L^2 R_0^2}{3\cdot 0.1440^2 b\mathbf{p}^2(K+1)} \leq \frac{526\cdot L^2 R_0^2}{b\mathbf{p}^2(K+1)}$. $\square$

Finally, we also restate Corollary 3.3 for the case $\gamma \in \left(\frac{1}{2}, 1\right)$ and then derive the oracle complexity of Corollary 3.3 from this result by fixing $\gamma := \frac{3}{4}$.

**Corollary C.3.** *Suppose that Assumptions 1.1, 1.3, and 1.4 hold for (NE) with $\kappa \geq 0$ as in Theorem 3.1. Let $\{x^k\}$ be generated by (VFR) using the SAGA estimator (SAGA), $\gamma \in \left(\frac{1}{2}, 1\right)$, and*

$$\eta := \frac{1}{L\sqrt{M}}, \quad \text{where} \quad \Lambda := \frac{2}{b} + \frac{4(1+\gamma^2)(n-b)(2n+b)}{b^3} \quad \text{and} \quad M := \frac{\gamma(1+5\gamma)}{3(2\gamma-1)} + \frac{1+6\gamma}{3(2\gamma-1)} \cdot \Lambda. \tag{28}$$

*Then, we have $\eta \geq \frac{\sigma b^{3/2}}{nL}$ for $\sigma := \frac{\sqrt{3(2\gamma-1)}}{\sqrt{10+61\gamma+13\gamma^2+48\gamma^3}}$, and the following bound holds:*

$$\frac{1}{K+1}\sum_{k=0}^{K}\mathbb{E}\big[\|Gx^k\|^2\big] \leq \frac{2(1+L^2\eta^2)R_0^2}{\gamma(1-\gamma)\eta^2(K+1)}, \quad \text{where} \quad R_0 := \|x^0 - x^\star\|. \tag{29}$$

*Moreover, for a given tolerance $\epsilon > 0$, if we choose $b := \lfloor n^{2/3} \rfloor$, then (VFR) requires $\mathcal{T}_{G_i} := n + \left\lfloor \frac{3\Gamma L^2 R_0^2 n^{2/3}}{\epsilon^2} \right\rfloor$ evaluations of $G_i$ to achieve $\frac{1}{K+1}\sum_{k=0}^{K}\mathbb{E}\big[\|Gx^k\|^2\big] \leq \epsilon^2$, where $\Gamma := \frac{2(7\gamma+5\gamma^2-3)(10+61\gamma+13\gamma^2+48\gamma^3)}{3\gamma^2(1-\gamma)(2\gamma-1)(1+5\gamma)}$.*

*Proof.* Since we use the SAGA estimator (SAGA), we have $\rho := \frac{b}{2n} \in (0, 1]$, $C := \frac{[2(n-b)(2n+b)+b^2]}{nb^2}$, and $\hat{C} := \frac{2(n-b)(2n+b)\gamma^2}{nb^2}$. In this case, since $b \geq 1$, we can easily show that $\Lambda := \frac{C+\hat{C}}{\rho} = \frac{2}{b} + \frac{4(1+\gamma^2)(n-b)(2n+b)}{b^3} \leq 2 + \frac{8(1+\gamma^2)n^2}{b^3}$. Hence, $M$ in Theorem 3.1 reduces to

$$M := \frac{\gamma(1+5\gamma)}{3(2\gamma-1)} + \frac{1+6\gamma}{3(2\gamma-1)} \cdot \Lambda \leq \frac{2+13\gamma+5\gamma^2}{3(2\gamma-1)} + \frac{8(1+\gamma^2)(1+6\gamma)n^2}{3(2\gamma-1)b^3}.$$

Suppose that $1 \leq b \leq n^{2/3}$. Then, one can prove that $M \leq \left[\frac{2+13\gamma+5\gamma^2}{3(2\gamma-1)} + \frac{8(1+\gamma^2)(1+6\gamma)}{3(2\gamma-1)}\right] \frac{n^2}{b^3} = \frac{(10+61\gamma+13\gamma^2+48\gamma^3)n^2}{3(2\gamma-1)b^3} = \frac{n^2}{\sigma^2 b^3}$, where $\sigma := \frac{\sqrt{3(2\gamma-1)}}{\sqrt{10+61\gamma+13\gamma^2+48\gamma^3}}$. Hence, if we choose $\eta := \frac{1}{L\sqrt{M}}$, then we get $\eta \geq \frac{\sigma b^{3/2}}{nL}$ as stated. Moreover, we obtain (27) from the first line of (7) as before.

Now, for $\eta := \frac{1}{L\sqrt{M}} \geq \frac{\sigma b^{3/2}}{nL}$, from (27), to guarantee $\frac{1}{K+1}\sum_{k=0}^{K} \mathbb{E}\left[\|Gx^k\|^2\right] \leq \epsilon^2$, we need to impose $\frac{2(1+L^2\eta^2)R_0^2}{\gamma(1-\gamma)\eta^2(K+1)} \leq \epsilon^2$, where $R_0 := \|x^0 - x^\star\|$. Since $1 + L^2\eta^2 = 1 + \frac{1}{M} \leq \frac{7\gamma+5\gamma^2-3}{\gamma(1+5\gamma)}$ and $\eta \geq \frac{\sigma b^{3/2}}{nL}$, the last condition holds if we choose $K := \left\lfloor \Gamma \cdot \frac{L^2 R_0^2 n^2}{b^3 \epsilon^2} \right\rfloor$, where $\Gamma := \frac{2(7\gamma+5\gamma^2-3)}{\sigma^2\gamma^2(1-\gamma)(1+5\gamma)} = \frac{2(7\gamma+5\gamma^2-3)(10+61\gamma+13\gamma^2+48\gamma^3)}{3\gamma^2(1-\gamma)(2\gamma-1)(1+5\gamma)}$.

Finally, at each iteration $k$, (VFR) requires 3 mini-batches of size $b$, leading to the cost of $3b$ per iteration. Hence, the total complexity is

$$\mathcal{T}_c := 3bK = \left\lfloor \frac{3\Gamma L^2 R_0^2 n^2}{b^2 \epsilon^2} \right\rfloor.$$

If we choose $b := \lfloor n^{2/3} \rfloor$, then $\mathcal{T}_c = \left\lfloor \frac{3\Gamma L^2 R_0^2 n^{2/3}}{\epsilon^2} \right\rfloor$. For the SAGA estimator (SAGA), one needs to compute $Gw^0$, which requires $n$ evaluations of $G_i$. Hence, the total complexity of the algorithm is $\mathcal{T}_{G_i} := n + \left\lfloor \frac{3\Gamma L^2 R_0^2 n^{2/3}}{\epsilon^2} \right\rfloor$. $\qquad\square$

***Proof of Corollary 3.3.*** Since we choose $\gamma := \frac{3}{4}$, we have $\sigma := \frac{\sqrt{3(2\gamma-1)}}{\sqrt{10+61\gamma+13\gamma^2+48\gamma^3}} = 0.14948 \geq 0.1494$ and $\Gamma := \frac{2(7\gamma+5\gamma^2-3)(10+61\gamma+13\gamma^2+48\gamma^3)}{3\gamma^2(1-\gamma)(2\gamma-1)(1+5\gamma)} = 2815.8 \leq 2816$. Applying the results of Corollary C.3, we obtain our conclusions in Corollary 3.3. Moreover, (29) reduces to $\frac{1}{K+1}\sum_{k=0}^{K} \mathbb{E}\left[\|Gx^k\|^2\right] \leq \frac{32(1+0.494^2)L^2 R_0^2}{3 \cdot 0.1494^2 b\mathbf{p}^2(K+1)} \leq \frac{489 \cdot L^2 R_0^2}{b\mathbf{p}^2(K+1)}$ as stated. $\qquad\square$

## D. Convergence Analysis of VFRBS for (NI): Technical Proofs

One key step to analyze the convergence of (VFRBS) is to construct an appropriate potential function. For this purpose, we introduce the following function:

$$\mathcal{L}_k := \|x^k + \gamma\eta(Gx^{k-1} + v^k) - x^\star\|^2 + \mu\|x^k - x^{k-1} + \gamma\eta(Gx^{k-1} + v^k)\|^2, \tag{30}$$

where $\mu > 0$ is a given parameter and $v^k \in Tx^k$ is given. This function is then combined with $\mathcal{E}_k$ from (21) to establish the convergence of (VFRBS).

Let us first state and prove Lemma D.1, which provides a key estimate for our convergence analysis of (VFRBS) in Theorem 4.1.

**Lemma D.1.** *Suppose that Assumption 1.3 holds for* (NI). *Let $\{x^k\}$ be generated by* (VFRBS), *$\mathcal{L}_k$ be defined by* (30), *and $\mathcal{E}_k$ be defined by* (21). *Then, we have*

$$\begin{aligned}
\mathcal{L}_k - \mathbb{E}_k\left[\mathcal{L}_{k+1}\right] \geq{}& 2(1-\gamma)\eta\langle Gx^k + v^k, x^k - x^\star\rangle + (1+\mu)(1-\gamma)(2\gamma-1)\eta^2\|Gx^k + v^k\|^2 \\
&+ \gamma[1-\gamma-\mu(3\gamma-1)]\eta^2\|Gx^{k-1} + v^k\|^2 - (1+\mu)\eta^2\mathbb{E}_k\left[\|e^k\|^2\right] \\
&+ \tfrac{1}{2}\left[\mu - 2(1+\mu)\gamma(1-\gamma)L^2\eta^2\right]\|x^k - x^{k-1}\|^2.
\end{aligned} \tag{31}$$

*If, additionally, Assumption 1.4 holds for* (NI), *then we have*

$$\begin{aligned}
\mathbb{E}\left[\mathcal{E}_k\right] - \mathbb{E}\left[\mathcal{E}_{k+1}\right] \geq{}& \tfrac{1}{2}\left[\mu - 2(1+\mu)\gamma(1-\gamma)L^2\eta^2 - \frac{2L^2\eta^2(1+\mu)(C+\hat{C})}{\rho}\right]\mathbb{E}\left[\|x^k - x^{k-1}\|^2\right] \\
&+ \gamma[1-\gamma-\mu(3\gamma-1)]\eta^2\mathbb{E}\left[\|Gx^{k-1} + v^k\|^2\right] \\
&+ (1-\gamma)\eta\left[(1+\mu)(2\gamma-1)\eta - 2\kappa\right]\mathbb{E}\left[\|Gx^k + v^k\|^2\right].
\end{aligned} \tag{32}$$

*Proof.* Let us introduce two notations $w^k := Gx^k + v^k$ and $\hat{w}^k := Gx^{k-1} + v^k$, where $v^k \in Tx^k$. We also recall $S_\gamma^k := Gx^k - \gamma Gx^{k-1}$ and $e^k := \widetilde{S}_\gamma^k - S_\gamma^k$ from (FRQ). Then, it is obvious that $\widetilde{S}_\gamma^k = S_\gamma^k + e^k = Gx^k - \gamma Gx^{k-1} + e^k$.

Now, using $\widetilde{S}_\gamma^k = Gx^k - \gamma Gx^{k-1} + e^k$, it follows from (VFRBS) that

$$
\begin{aligned}
x^{k+1} &= x^k - \eta \widetilde{S}_\gamma^k - \gamma \eta v^{k+1} - (2\gamma - 1)\eta v^k \\
&= x^k - \gamma \eta (Gx^k + v^{k+1}) - (1-\gamma)\eta(Gx^k + v^k) + \eta\gamma(Gx^{k-1} + v^k) - \eta e^k \\
&= x^k - \gamma\eta \hat{w}^{k+1} - (1-\gamma)\eta w^k + \gamma\eta \hat{w}^k - \eta e^k.
\end{aligned}
\tag{33}
$$

Then, using (33) and $\hat{w}^{k+1} = Gx^k + v^{k+1}$, we can show that

$$
\begin{aligned}
\mathcal{T}_{[1]} &:= \|x^{k+1} + \gamma\eta(Gx^k + v^{k+1}) - x^\star\|^2 = \|x^{k+1} - x^\star + \gamma\eta \hat{w}^{k+1}\|^2 \\
&\overset{(33)}{=} \|x^k - \gamma\eta\hat{w}^{k+1} - (1-\gamma)\eta w^k + \gamma\eta\hat{w}^k - \eta e^k - x^\star + \gamma\eta\hat{w}^{k+1}\|^2 \\
&= \|x^k - x^\star\|^2 - 2(1-\gamma)\eta\langle w^k, x^k - x^\star\rangle + 2\gamma\eta\langle\hat{w}^k, x^k - x^\star\rangle + \eta^2\|e^k\|^2 \\
&\quad + (1-\gamma)^2\eta^2\|w^k\|^2 - 2\gamma(1-\gamma)\eta^2\langle w^k, \hat{w}^k\rangle + \gamma^2\eta^2\|\hat{w}^k\|^2 \\
&\quad - 2\eta\langle e^k, x^k - x^\star\rangle + 2(1-\gamma)\eta^2\langle e^k, w^k\rangle - 2\gamma\eta^2\langle e^k, \hat{w}^k\rangle.
\end{aligned}
$$

Alternatively, using $\hat{w}^k = Gx^{k-1} + v^k$, we also have

$$
\begin{aligned}
\mathcal{T}_{[2]} &:= \|x^k + \gamma\eta(Gx^{k-1} + v^k) - x^\star\|^2 = \|x^k - x^\star + \gamma\eta\hat{w}^k\|^2 \\
&= \|x^k - x^\star\|^2 + 2\gamma\eta\langle\hat{w}^k, x^k - x^\star\rangle + \gamma^2\eta^2\|\hat{w}^k\|^2.
\end{aligned}
$$

Subtracting $\mathcal{T}_{[1]}$ from $\mathcal{T}_{[2]}$, we can show that

$$
\begin{aligned}
\mathcal{T}_{[3]} &:= \|x^k + \gamma\eta(Gx^{k-1} + v^k) - x^\star\|^2 - \|x^{k+1} + \gamma\eta(Gx^k + v^{k+1}) - x^\star\|^2 \\
&= 2(1-\gamma)\eta\langle w^k, x^k - x^\star\rangle - (1-\gamma)^2\eta^2\|w^k\|^2 + 2\gamma(1-\gamma)\eta^2\langle w^k, \hat{w}^k\rangle \\
&\quad + 2\eta\langle e^k, x^k - x^\star\rangle - 2(1-\gamma)\eta^2\langle e^k, w^k\rangle + 2\gamma\eta^2\langle e^k, \hat{w}^k\rangle - \eta^2\|e^k\|^2 \\
&= 2(1-\gamma)\eta\langle w^k, x^k - x^\star\rangle + (1-\gamma)(2\gamma-1)\eta^2\|w^k\|^2 \\
&\quad + \gamma(1-\gamma)\eta^2\|\hat{w}^k\|^2 - \gamma(1-\gamma)\eta^2\|w^k - \hat{w}^k\|^2 \\
&\quad + 2\eta\langle e^k, x^k - x^\star\rangle - 2(1-\gamma)\eta^2\langle e^k, w^k\rangle + 2\gamma\eta^2\langle e^k, \hat{w}^k\rangle - \eta^2\|e^k\|^2.
\end{aligned}
\tag{34}
$$

Next, using again $\hat{w}^{k+1} = Gx^k + v^{k+1}$ and (33), we have

$$
\begin{aligned}
\mathcal{T}_{[4]} &:= \|x^{k+1} - x^k + \gamma\eta(Gx^k + v^{k+1})\|^2 = \|x^{k+1} - x^k + \gamma\eta\hat{w}^{k+1}\|^2 \\
&\overset{(33)}{=} \eta^2\|(1-\gamma)w^k - \gamma\hat{w}^k + e^k\|^2 \\
&= (1-\gamma)^2\eta^2\|w^k\|^2 - 2\gamma(1-\gamma)\eta^2\langle w^k, \hat{w}^k\rangle + \gamma^2\eta^2\|\hat{w}^k\|^2 \\
&\quad + \eta^2\|e^k\|^2 + 2(1-\gamma)\eta^2\langle e^k, w^k\rangle - 2\gamma\eta^2\langle e^k, \hat{w}^k\rangle \\
&= -(1-\gamma)(2\gamma-1)\eta^2\|w^k\|^2 + \gamma(2\gamma-1)\eta^2\|\hat{w}^k\|^2 + \gamma(1-\gamma)\eta^2\|w^k - \hat{w}^k\|^2 \\
&\quad + \eta^2\|e^k\|^2 + 2(1-\gamma)\eta^2\langle e^k, w^k\rangle - 2\gamma\eta^2\langle e^k, \hat{w}^k\rangle.
\end{aligned}
$$

Moreover, by the Cauchy-Schwarz inequality in ① and Young's inequality in ②, we can prove that

$$
\begin{aligned}
\|x^k - x^{k-1} + \gamma\eta\hat{w}^k\|^2 &= \|x^k - x^{k-1}\|^2 + 2\gamma\eta\langle\hat{w}^k, x^k - x^{k-1}\rangle + \gamma^2\eta^2\|\hat{w}^k\|^2 \\
&\overset{①}{\geq} \|x^k - x^{k-1}\|^2 - 2\gamma\eta\|\hat{w}^k\|\|x^k - x^{k-1}\| + \gamma^2\eta^2\|\hat{w}^k\|^2 \\
&\overset{②}{\geq} \tfrac{1}{2}\|x^k - x^{k-1}\|^2 - \gamma^2\eta^2\|\hat{w}^k\|^2.
\end{aligned}
$$

Combining the last two expressions, we can show that

$$
\begin{aligned}
\mathcal{T}_{[5]} &:= \|x^k - x^{k-1} + \gamma\eta(Gx^{k-1} + v^k)\|^2 - \|x^{k+1} - x^k + \gamma\eta(Gx^k + v^{k+1})\|^2 \\
&= \|x^k - x^{k-1} + \gamma\eta\hat{w}^k\|^2 - \|x^{k+1} - x^k + \gamma\eta\hat{w}^{k+1}\|^2 \\
&\geq \tfrac{1}{2}\|x^k - x^{k-1}\|^2 + (1-\gamma)(2\gamma-1)\eta^2\|w^k\|^2 - \gamma(3\gamma-1)\eta^2\|\hat{w}^k\|^2 \\
&\quad - \gamma(1-\gamma)\eta^2\|w^k - \hat{w}^k\|^2 - \eta^2\|e^k\|^2 - 2(1-\gamma)\eta^2\langle e^k, w^k\rangle + 2\gamma\eta^2\langle e^k, \hat{w}^k\rangle.
\end{aligned}
$$

Multiplying $\mathcal{T}_{[5]}$ by $\mu > 0$, and adding the result to (34), and using $\mathcal{L}_k$ from (30), we have

$$
\begin{aligned}
\mathcal{L}_k - \mathcal{L}_{k+1} &= \|x^k + \gamma\eta(Gx^{k-1} + v^k) - x^\star\|^2 - \|x^{k+1} + \gamma\eta(Gx^k + v^{k+1}) - x^\star\|^2 \\
&\quad + \mu\|x^k - x^{k-1} + \gamma\eta(Gx^{k-1} + v^k)\|^2 - \mu\|x^{k+1} - x^k + \gamma\eta(Gx^k + v^{k+1})\|^2 \\
&\geq 2(1-\gamma)\eta\langle w^k, x^k - x^\star\rangle + \tfrac{\mu}{2}\|x^k - x^{k-1}\|^2 + (1+\mu)(1-\gamma)(2\gamma-1)\eta^2\|w^k\|^2 \\
&\quad + \gamma[(1-\gamma) - \mu(3\gamma-1)]\eta^2\|\hat{w}^k\|^2 - (1+\mu)\gamma(1-\gamma)\eta^2\|w^k - \hat{w}^k\|^2 \\
&\quad + 2\eta\langle e^k, x^k - x^\star\rangle - 2(1+\mu)(1-\gamma)\eta^2\langle e^k, w^k\rangle \\
&\quad + 2(1+\mu)\gamma\eta^2\langle e^k, \hat{w}^k\rangle - (1+\mu)\eta^2\|e^k\|^2.
\end{aligned}
$$

Taking the conditional expectation $\mathbb{E}_k[\,\cdot\,]$ both sides of this expression, and noting that

$$
\begin{aligned}
\mathbb{E}_k\big[\langle e^k, x^k - x^\star\rangle\big] &= \langle \mathbb{E}_k[e^k], x^k - x^\star\rangle = 0, \\
\mathbb{E}_k\big[\langle e^k, w^k\rangle\big] &= \langle \mathbb{E}_k[e^k], w^k\rangle = 0, \\
\mathbb{E}_k\big[\langle e^k, \hat{w}^k\rangle\big] &= \langle \mathbb{E}_k[e^k], \hat{w}^k\rangle = 0,
\end{aligned}
$$

we obtain

$$
\begin{aligned}
\mathcal{L}_k - \mathbb{E}_k[\mathcal{L}_{k+1}] &\geq 2(1-\gamma)\eta\langle w^k, x^k - x^\star\rangle + \tfrac{\mu}{2}\|x^k - x^{k-1}\|^2 + (1+\mu)(1-\gamma)(2\gamma-1)\eta^2\|w^k\|^2 \\
&\quad + \gamma[(1-\gamma) - \mu(3\gamma-1)]\eta^2\|\hat{w}^k\|^2 - (1+\mu)\gamma(1-\gamma)\eta^2\|w^k - \hat{w}^k\|^2 \\
&\quad - (1+\mu)\eta^2\mathbb{E}_k\big[\|e^k\|^2\big].
\end{aligned}
$$

Finally, by the $L$-Lipschitz continuity of $G$ from (2) of Assumption 1.3, we have $\|w^k - \hat{w}^k\|^2 = \|Gx^k - Gx^{k-1}\|^2 \leq L^2\|x^k - x^{k-1}\|^2$ as shown in (24). Using this inequality into the last estimate, we can show that

$$
\begin{aligned}
\mathcal{L}_k - \mathbb{E}_k[\mathcal{L}_{k+1}] &\geq 2(1-\gamma)\eta\langle w^k, x^k - x^\star\rangle + (1+\mu)(1-\gamma)(2\gamma-1)\eta^2\|w^k\|^2 \\
&\quad + \gamma[1-\gamma-\mu(3\gamma-1)]\eta^2\|\hat{w}^k\|^2 - (1+\mu)\eta^2\mathbb{E}_k\big[\|e^k\|^2\big] \\
&\quad + \tfrac{1}{2}\big[\mu - 2(1+\mu)\gamma(1-\gamma)L^2\eta^2\big]\|x^k - x^{k-1}\|^2,
\end{aligned}
$$

which proves (31) by recalling $w^k := Gx^k + v^k$ and $\hat{w}^k := Gx^{k-1} + v^k$.

Taking the full expectation of (31) and using $\langle Gx^k + v^k, x^k - x^\star\rangle \geq -\kappa\|Gx^k + v^k\|^2$ from Assumption 1.4 and $\mathbb{E}_k\big[\|e^k\|^2\big] \leq \Delta_k$ from (3), we can bound it as

$$
\begin{aligned}
\mathbb{E}[\mathcal{L}_k] - \mathbb{E}[\mathcal{L}_{k+1}] &\geq \tfrac{1}{2}\big[\mu - 2(1+\mu)\gamma(1-\gamma)L^2\eta^2\big]\mathbb{E}\big[\|x^k - x^{k-1}\|^2\big] - (1+\mu)\eta^2\Delta_k \\
&\quad + \gamma[1-\gamma-\mu(3\gamma-1)]\eta^2\mathbb{E}\big[\|Gx^{k-1} + v^k\|^2\big] \\
&\quad + (1-\gamma)\eta\big[(1+\mu)(2\gamma-1)\eta - 2\kappa\big]\mathbb{E}\big[\|Gx^k + v^k\|^2\big].
\end{aligned}
\tag{35}
$$

By the third line of (3) in Definition 2.1 and utilizing again (2), we have

$$
\Delta_k \leq (1-\rho)\Delta_{k-1} + CL^2\mathbb{E}\big[\|x^k - x^{k-1}\|^2\big] + \hat{C}L^2\mathbb{E}\big[\|x^{k-1} - x^{k-2}\|^2\big].
$$

Rearranging this inequality, we get

$$\Delta_k \leq \left(\tfrac{1-\rho}{\rho}\right)\left(\Delta_{k-1} - \Delta_k\right) + \tfrac{\hat{C}L^2}{\rho}\left[\mathbb{E}\left[\|x^{k-1} - x^{k-2}\|^2\right] - \mathbb{E}\left[\|x^k - x^{k-1}\|^2\right]\right]$$
$$+ \tfrac{(C+\hat{C})L^2}{\rho}\mathbb{E}\left[\|x^k - x^{k-1}\|^2\right].$$

Substituting this inequality into (35), we can show that

$$\mathbb{E}\left[\mathcal{L}_k\right] - \mathbb{E}\left[\mathcal{L}_{k+1}\right] \geq \tfrac{1}{2}\left[\mu - 2(1+\mu)\gamma(1-\gamma)L^2\eta^2 - \tfrac{2L^2\eta^2(1+\mu)(C+\hat{C})}{\rho}\right]\mathbb{E}\left[\|x^k - x^{k-1}\|^2\right]$$
$$+ \gamma[1 - \gamma - \mu(3\gamma - 1)]\eta^2\mathbb{E}\left[\|Gx^{k-1} + v^k\|^2\right]$$
$$+ (1-\gamma)\eta\left[(1+\mu)(2\gamma - 1)\eta - 2\kappa\right]\mathbb{E}\left[\|Gx^k + v^k\|^2\right]$$
$$- \tfrac{L^2\eta^2\hat{C}(1+\mu)}{\rho}\left[\mathbb{E}\left[\|x^{k-1} - x^{k-2}\|^2\right] - \mathbb{E}\left[\|x^k - x^{k-1}\|^2\right]\right]$$
$$- \tfrac{\eta^2(1+\mu)(1-\rho)}{\rho}\left(\Delta_{k-1} - \Delta_k\right).$$

Rearranging this inequality and using $\mathcal{E}_k$ from (21), we obtain (32). □

Now, we are ready to prove our second main result, Theorem 4.1 in the main text.

***Proof of Theorem 4.1.*** Since we fix $\gamma \in \left(\tfrac{1}{2}, 1\right)$ and $\mu := \tfrac{1-\gamma}{3\gamma-1}$, we have $\mu > 0$ and $1 + \mu = \tfrac{2\gamma}{3\gamma-1}$. Let us denote by $M := 4\gamma^2 + \tfrac{4\gamma}{1-\gamma} \cdot \tfrac{C+\hat{C}}{\rho}$ as in Theorem 4.1. Then, (32) reduces to

$$\mathbb{E}\left[\mathcal{E}_k\right] - \mathbb{E}\left[\mathcal{E}_{k+1}\right] \geq \tfrac{(1-\gamma)(1-M\cdot L^2\eta^2)}{2(3\gamma-1)}\mathbb{E}\left[\|x^k - x^{k-1}\|^2\right] \qquad (36)$$
$$+ 2(1-\gamma)\eta\left[\tfrac{\gamma(2\gamma-1)\eta}{3\gamma-1} - \kappa\right]\mathbb{E}\left[\|Gx^k + v^k\|^2\right].$$

Let us choose $\eta > 0$ such that $\tfrac{\gamma(2\gamma-1)\eta}{3\gamma-1} - \kappa > 0$ and $1 - M \cdot L^2\eta^2 \geq 0$. These two conditions lead to $\tfrac{(3\gamma-1)\kappa}{\gamma(2\gamma-1)} < \eta \leq \tfrac{1}{L\sqrt{M}}$ as stated in Theorem 4.1. However, this condition holds if $L^2\kappa^2 < \tfrac{\gamma^2(2\gamma-1)^2}{M(3\gamma-1)^2}$. This condition is equivalent to $L\kappa \leq \delta$ as our condition in Theorem 4.1, where $\delta := \tfrac{\gamma(2\gamma-1)}{(3\gamma-1)\sqrt{M}}$.

Averaging (36) from $k = 0$ to $K$ and noting that $\mathbb{E}\left[\mathcal{E}_k\right] \geq 0$ for all $k \geq 0$, we get

$$\tfrac{1}{K+1}\sum_{k=0}^{K}\mathbb{E}\left[\|Gx^k + v^k\|^2\right] \leq \tfrac{(3\gamma-1)\cdot\mathbb{E}[\mathcal{E}_0]}{2(1-\gamma)[\gamma(2\gamma-1)\eta-(3\gamma-1)\kappa]\eta(K+1)},$$
$$\tfrac{(1-ML^2\eta^2)}{K+1}\sum_{k=0}^{K}\mathbb{E}\left[\|x^k - x^{k-1}\|^2\right] \leq \tfrac{2(3\gamma-1)\cdot\mathbb{E}[\mathcal{E}_0]}{(1-\gamma)(K+1)}.$$

Finally, since $x^{-1} = x^{-2} = x^0$, we have $\Delta_{-1} = \Delta_0$. However, since $\widetilde{S}_\gamma^0 = (1-\gamma)Gx^0 = S_\gamma^0$, we get $\Delta_0 = \|\widetilde{S}_\gamma^0 - S_\gamma^0\|^2 = 0$. Using these relations, $\rho \in [0, 1]$ and $\gamma < 1$, we can show that

$$\mathbb{E}\left[\mathcal{E}_0\right] = \mathbb{E}\left[\|x^0 + \gamma\eta(Gx^0 + v^0) - x^\star\|^2\right] + \tfrac{\eta^2(1+\mu)(1-\rho)}{\rho}\Delta_0$$
$$\leq 2\mathbb{E}\left[\|x^0 - x^\star\|^2\right] + 2\gamma^2\eta^2\mathbb{E}\left[\|Gx^0 + v^0\|^2\right] + \tfrac{2\gamma\eta^2}{(3\gamma-1)\rho}\Delta_0$$
$$= 2\mathbb{E}\left[\|x^0 - x^\star\|^2\right] + 2\gamma^2\eta^2\mathbb{E}\left[\|Gx^0 + v^0\|^2\right].$$

Substituting this upper bound into the above two estimates, we get two lines of (12). □

Finally, we prove Corollaries 4.2 and 4.3 in the main text. Unlike Corollaries 3.2 and 3.3 where we fix $\gamma := \tfrac{3}{4}$, here we state these corollaries for any value of $\gamma \in \left(\tfrac{1}{2}, 1\right)$.

***Proof of Corollary 4.2.*** For the SVRG estimator (L-SVRG), we have $\rho := \tfrac{\mathbf{p}}{2} \in (0, 1]$, $C := \tfrac{4-6\mathbf{p}+3\mathbf{p}^2}{b\mathbf{p}}$, $\hat{C} := \tfrac{2\gamma^2(2-3\mathbf{p}+\mathbf{p}^2)}{b\mathbf{p}}$, and $\Delta_0 = 0$ due to (17) and $x^0 = x^{-1} = w^0$. In this case, we have $\Lambda := \tfrac{C+\hat{C}}{\rho} = \tfrac{4(1+\gamma^2)(2-3\mathbf{p})+2(3+2\gamma^2)\mathbf{p}^2}{b\mathbf{p}^2} \leq \tfrac{8(1+\gamma^2)}{b\mathbf{p}^2}$, and thus $M$ in Theorem 3.1 reduces to $M := 4\gamma^2 + \tfrac{4\gamma}{1-\gamma}\Lambda \leq 4\gamma^2 + \tfrac{32(1+\gamma^2)}{b\mathbf{p}^2}$.

Suppose that $b\mathbf{p}^2 \leq 1$. Since $\Lambda \leq \frac{8(1+\gamma^2)}{b\mathbf{p}^2}$ and $M = 4\gamma^2 + \frac{4\gamma}{1-\gamma}\Lambda \leq 4\gamma^2 + \frac{32\gamma(1+\gamma^2)}{(1-\gamma)b\mathbf{p}^2} \leq \frac{4\gamma(8+\gamma+7\gamma^2)}{(1-\gamma)b\mathbf{p}^2}$. If we choose $\eta := \frac{1}{L\sqrt{M}}$, then we have $\eta \geq \frac{\sqrt{1-\gamma}\sqrt{b}\mathbf{p}}{2L\sqrt{\gamma(8+\gamma+7\gamma^2)}} = \frac{\sigma\sqrt{b}\mathbf{p}}{L}$ with $\sigma := \frac{\sqrt{1-\gamma}}{2\sqrt{8+\gamma+7\gamma^2}}$, then it satisfies $\frac{(3\gamma-1)\kappa}{\gamma(2\gamma-1)} < \eta \leq \frac{1}{L\sqrt{M}}$ in Theorem 4.1, provided that $L\kappa \leq \delta$. Note that using $\eta \geq \frac{\sigma\sqrt{b}\mathbf{p}}{L}$ in (12) of Theorem 4.1 we obtain the bound (13).

Now, from the first line of (12), to guarantee $\frac{1}{K+1}\sum_{k=0}^{K}\mathbb{E}\left[\|Gx^k + v^k\|^2\right] \leq \epsilon^2$, we need to impose $\frac{\Theta\hat{R}_0^2}{\eta^2(K+1)} \leq \epsilon^2$, where $\hat{R}_0^2 := \|x^0 - x^\star\|^2 + \gamma^2\eta^2\|Gx^0 + v^0\|^2$. Since $\eta \geq \frac{\sigma\sqrt{b}\mathbf{p}}{L}$, the last condition holds if we choose $K := \left\lfloor \Gamma \cdot \frac{L^2\hat{R}_0^2}{b\mathbf{p}^2\epsilon^2} \right\rfloor$, where $\Gamma := \frac{\Theta}{\sigma^2}$.

Finally, at each iteration $k$, (VFRBS) requires 3 mini-batches of size $b$, and occasionally compute the full $Gw^k$, leading to the cost of $n\mathbf{p} + 3b$ per iteration. Thus the total complexity is

$$\mathcal{T}_c := K(n\mathbf{p} + 3b) = \frac{\Gamma L^2\hat{R}_0^2(n\mathbf{p}+3b)}{b\mathbf{p}^2\epsilon^2} = \frac{\Gamma L^2\hat{R}_0^2}{\epsilon^2}\left(\frac{n}{b\mathbf{p}} + \frac{3}{\mathbf{p}^2}\right).$$

If we choose $b := \lfloor n^{2/3} \rfloor$ and $\mathbf{p} := n^{-1/3}$, then $b\mathbf{p}^2 = 1$ and $\mathcal{T}_c = \frac{4\Gamma n^{2/3}L^2\hat{R}_0^2}{\epsilon^2}$. For the SVRG estimator (L-SVRG), one needs to compute $Gw^0$, which requires $n$ evaluations of $G_i$. Hence, the total evaluations of $G_i$ is $\mathcal{T}_{G_i} = n + \left\lfloor \frac{4\Gamma n^{2/3}L^2\hat{R}_0^2}{\epsilon^2} \right\rfloor$. Moreover, at each iteration, we need one evaluation of $J_{\gamma\eta T}$. Therefore, the total evaluations of $J_{\gamma\eta T}$ is $\mathcal{T}_T := K = \left\lfloor \Gamma \cdot \frac{L^2\hat{R}_0^2}{b\mathbf{p}^2\epsilon^2} \right\rfloor = \left\lfloor \Gamma \cdot \frac{L^2\hat{R}_0^2}{\epsilon^2} \right\rfloor$. $\square$

***Proof of Corollary 4.3.*** Since we use the SAGA estimator (SAGA), we have $\rho := \frac{b}{2n} \in (0,1]$, $C := \frac{[2(n-b)(2n+b)+b^2]}{nb^2}$, and $\hat{C} := \frac{2(n-b)(2n+b)\gamma^2}{nb^2}$. In this case, since $b \geq 1$, we get $\Lambda := \frac{C+\hat{C}}{\rho} = \frac{2}{b} + \frac{4(1+\gamma^2)(n-b)(2n+b)}{b^3} \leq 2 + \frac{8(1+\gamma^2)n^2}{b^3}$. Hence, $M$ in Theorem 3.1 reduces to

$$M := 4\gamma^2 + \frac{4\gamma}{1-\gamma} \cdot \Lambda \leq \frac{4\gamma(2+\gamma-\gamma^2)}{1-\gamma} + \frac{32\gamma(1+\gamma^2)n^2}{(1-\gamma)b^3}$$

Suppose that $1 \leq b \leq n^{2/3}$. Then, we can show that $M \leq \left[\frac{4\gamma(2+\gamma-\gamma^2)}{1-\gamma} + \frac{32\gamma(1+\gamma^2)}{1-\gamma}\right]\frac{n^2}{b^3} = \frac{4\gamma(10+\gamma+7\gamma^2)}{(1-\gamma)b^3} = \frac{n^2}{\sigma^2 b^3}$, where $\sigma := \frac{\sqrt{1-\gamma}}{2\sqrt{\gamma(10+\gamma+7\gamma^2)}}$. Hence, if we choose $\eta := \frac{1}{L\sqrt{M}}$, then we get $\eta \geq \frac{\sigma b^{3/2}}{nL}$. Note that using $\eta \geq \frac{\sigma b^{3/2}}{nL}$ in (12) of Theorem 4.1 we obtain the bound (14).

For $\eta := \frac{1}{L\sqrt{M}} \geq \frac{\sigma b^{3/2}}{nL}$, from the first line of (12), to guarantee $\frac{1}{K+1}\sum_{k=0}^{K}\mathbb{E}\left[\|Gx^k + v^k\|^2\right] \leq \epsilon^2$, we need to impose $\frac{\Theta\hat{R}_0^2}{\eta^2(K+1)} \leq \epsilon^2$, where $\hat{R}_0^2 := \|x^0 - x^\star\|^2 + \gamma^2\eta^2\|Gx^0 + v^0\|^2$. Since $\eta \geq \frac{\sigma b^{3/2}}{nL}$, the last condition holds if we choose $K := \left\lfloor \Gamma \cdot \frac{L^2\hat{R}_0^2 n^2}{b^3\epsilon^2} \right\rfloor$, where $\Gamma := \frac{\Theta}{\sigma^2}$.

Finally, at each iteration $k$, (VFRBS) requires 3 mini-batches of size $b$, leading to the cost of $3b$ per iteration. Thus the total complexity is

$$\mathcal{T}_c := 3bK = \left\lfloor \frac{3\Gamma L^2\hat{R}_0^2 n^2}{b^2\epsilon^2} \right\rfloor.$$

If we choose $b := \lfloor n^{2/3} \rfloor$, then $\mathcal{T}_c = \left\lfloor \frac{3\Gamma L^2\hat{R}_0^2 n^{2/3}}{\epsilon^2} \right\rfloor$. For the SAGA estimator (SAGA), one needs to compute $Gw^0$, which requires $n$ evaluations of $G_i$. We conclude that (VFRBS) requires $\mathcal{T}_{G_i} := n + \left\lfloor \frac{3\Gamma L^2\hat{R}_0^2 n^{2/3}}{\epsilon^2} \right\rfloor$ evaluations of $G_i$. Moreover, since each iteration, it requires one evaluation of $J_{\gamma\eta T}$, we need $\mathcal{T}_T := K = \left\lfloor \Gamma \cdot \frac{L^2\hat{R}_0^2}{\epsilon^2} \right\rfloor$ evaluations of $J_{\gamma\eta T}$. $\square$

*Remark* D.2. For the SVRG estimator, if we choose $\gamma = \frac{3}{4}$, then we have $\sigma := 0.0702$. Hence, we have $\eta \geq \frac{0.0702\sqrt{b}\mathbf{p}}{L}$. However, if we choose $\gamma := 0.55$, then $\eta \geq \frac{0.1027\sqrt{b}\mathbf{p}}{L}$. If we choose $b = \lfloor n^{2/3} \rfloor$ and $\mathbf{p} = n^{-1/3}$, then the latter lower bound becomes $\eta \geq \frac{0.1027}{L}$.

For the SAGA estimator, if we choose $\gamma = \frac{3}{4}$, then we have $\sigma := 0.0753$. Hence, we get $\eta \geq \frac{0.0753b^{3/2}}{nL}$. However, if we set $\gamma := 0.55$, then $\eta \geq \frac{0.1271b^{3/2}}{nL}$. If we choose $b = \lfloor n^{2/3} \rfloor$, then the latter lower bound becomes $\eta \geq \frac{0.1271}{L}$.

Note that these lower bounds of $\eta$ can be further improved by refining the related parameters in Lemma D.1, and carefully choosing $\mu$ in the proof of Theorem 4.1.

# E. Details of Experiments and Additional Experiments

Due to space limit, we do not provide the details of experiments in Section 5. In this Supp. Doc., we provide the details of our implementation and experiments. We also add more examples to illustrate our algorithms and compare them with existing methods. All algorithms are implemented in Python, and all the experiments are run on a MacBookPro. 2.8GHz Quad-Core Intel Core I7, 16Gb Memory.

## E.1. Synthetic WGAN Example

We modify the synthetic example in (Daskalakis et al., 2018) built up on WGAN from (Arjovsky et al., 2017) as our first example. Suppose that the generator is a simple additive model $G_\theta(z) = \theta + z$ with the noise input $z$ generated from a normal distribution $\mathcal{N}(0, \mathbb{I})$, and the discriminator is also a linear function $D_\beta(w) = \langle K\beta, w \rangle$ for a given matrix $K$, where $\theta \in \mathbb{R}^{p_1}$ and $\beta \in \mathbb{R}^{p_2}$, and $K \in \mathbb{R}^{p_1 \times p_2}$ is a given matrix. The goal of the generator is to find a true distribution $\theta = \theta^*$, leading to the following loss:

$$\mathcal{L}(\theta, \beta) := \mathbb{E}_{u \sim \mathcal{N}(\theta^*, \mathbb{I})}\big[\langle K\beta, w \rangle\big] - \mathbb{E}_{z \sim \mathcal{N}(0, \mathbb{I})}\big[\langle K\beta, \theta + z \rangle\big].$$

Suppose that we have $n$ samples for both $w$ and $z$ leading to the following bilinear minimax problem:

$$\inf_{\theta \in \mathbb{R}^{p_1}} \sup_{\beta \in \mathbb{R}^{p_2}} \big\{ \mathcal{L}(\theta, \beta) := f(\theta) + \frac{1}{n} \sum_{i=1}^{n} \big[\langle K\beta, w_i - z_i - \theta \rangle\big] - g(\beta) \big\}. \tag{37}$$

Here, we add two convex functions $f(\theta)$ and $g(\beta)$ to possibly handle constraints or regularizers associated with $\theta$ and $\beta$, respectively.

If we define $x := [\theta, \beta] \in \mathbb{R}^{p_1 + p_2}$, $Gx = [\nabla_\theta \mathcal{L}(\theta, \beta), -\nabla_\beta \mathcal{L}(\theta, \beta)] := -[K\beta, \frac{1}{n} \sum_{i=1}^{n} K^\top (w_i - z_i - \theta)]$, and $T := [\partial f(\theta), \partial g(\beta)]$, then the optimality condition of this minimax problem becomes $0 \in Gx + Tx$, which is a special case of (NI) with $Gx$ being linear. The model (37) is different from the one in (Daskalakis et al., 2018) at two points:

- It involves a linear operator $K$, making it more general than (Daskalakis et al., 2018).
- It has two additional terms $f$ and $g$, making it broader to also cover constraints or non-smooth regularizers.

In our experiments below, we consider two cases:

- **Case 1 (Unconstrained setting).** We assume that $\theta \in \mathbb{R}^{p_1}$ and $\beta \in \mathbb{R}^{p_2}$.
- **Case 2 (Constrained setting).** Assume that $\theta$ and $\beta$ stays in an $\ell_\infty$-ball of radius $r > 0$, leading to $f(\theta) := \delta_{[-r,r]^{p_1}}(\theta)$ and $g(\beta) := \delta_{[-r,r]^{p_2}}(\beta)$, the indicator of the $\ell_\infty$-balls.

### E.1.1. THE UNCONSTRAINED CASE

(a) **Algorithms.** We implement three variants of (VFR) to solve (37).

- The first variant is using a double-loop SVRG strategy (called VFR-svrg), where the full operator $Gw^s$ at a snapshot point $w^s$ is computed at the beginning of each epoch $s$. Then, we perform $\lfloor n/b \rfloor$ iterations $k$ to update $x^k$ using (VFR), where $b$ is the mini-batch size. Finally, we set the next snapshot point $w^{s+1} := x^{k+1}$ after finishing the inner loop.
- The second variant is called a loopless one, LVFR-svrg, where we implement exactly the same scheme (VFR) as in this paper and using the Loopless-SVRG estimator.
- The third variant is VFR-saga, where we use the SAGA estimator in (VFR).

We also compare our methods with the deterministic optimistic gradient (OG) in (Daskalakis et al., 2018), the variance-reduced FRBS (VFRBS) in (Alacaoglu et al., 2023), and the variance-reduced extragradient (VEG) in (Alacaoglu & Malitsky, 2022).

(b) **Input data.** For (NE), we generate a vector $\theta^*$ from the standard normal distribution as our true mean in $\mathbb{R}^{p_1}$. Then, we generate i.i.d. samples $w_i$ and $z_i$ from normal distribution $\mathcal{N}(\theta^*, \mathbb{I})$ and $\mathcal{N}(0, \mathbb{I})$, respectively for $i = 1, 2, \cdots, n$ in $\mathbb{R}^{p_1}$ and $\mathbb{R}^{p_2}$, respectively. We perform two expertiments: **Experiment 1** with $n = 5000$ and $p_1 = p_2 = 100$, and **Experiment 2** with $n = 10000$ and $p_1 = p_2 = 200$. For each experiment, we run 10 times up to 100 epochs, corresponding to 10 problem instances, using the same setting, but different input data $(w_i, z_i)$, and then compute the mean of the relative operator norm $\|Gx^k\| / \|Gx^0\|$. This mean is then plotted.

(c) **Parameters.** For the optimistic gradient algorithm (OG), we choose its learning rate $\eta := \frac{1}{L}$, where $L$ is the Lipschitz constant of $G$, though its theoretical learning rate is much smaller. For our methods in (VFR), if $n = 5000$, and we choose $b := \lfloor 0.5n^{2/3} \rfloor = 146$, and the probability $\mathbf{p} := \frac{2}{n^{1/3}} = 0.1170$, then $\eta := \frac{1}{L\sqrt{M}} = \frac{0.1905}{L}$. However, due to the under

estimation of $M$, we instead use a larger learning rate $\eta := \frac{1}{2L}$ for all three variants, and choose a mini-batch of size $b := \lfloor 0.5n^{2/3} \rfloor$, and a probability $\mathbf{p} := \frac{1}{n^{1/3}}$ for the loopless SVRG variant.

For the forward-reflected-backward splitting method with variance reduction (VFRBS) in (Alacaoglu et al., 2023), we choose its learning rate $\eta := \frac{0.95(1-\sqrt{1-\mathbf{p}})}{2L}$ as suggested by (Alacaoglu et al., 2023). However, we still choose the probability $\mathbf{p} = \frac{1}{n^{1/3}}$ and the mini-batch size $b = \lfloor 0.5n^{2/3} \rfloor$ as our methods. These values are much larger the ones suggested in (Alacaoglu et al., 2023), typically $\mathbf{p} = \mathcal{O}(1/n)$.

For the variance reduction extragradient method (VEG) in (Alacaoglu & Malitsky, 2022), we choose its learning rate $\eta := \frac{0.95\sqrt{1-\alpha}}{L}$ for $\alpha := 1 - \mathbf{p}$ from the paper. However, again, we also choose $\mathbf{p} := \frac{1}{n^{1/3}}$ and $b = \lfloor 0.5n^{2/3} \rfloor$ in this method, which is the same as ours, though their theoretical results suggest smaller values of $\mathbf{p}$ (e.g., $\mathbf{p} = \frac{1}{n}$). Note that if $n = 5000$, then the batch size $b := 150$ and the probability $\mathbf{p} := 0.062$, but if $n = 10000$, then $b = 239$ and $\mathbf{p} = 0.0479$.

(d) **Experiments for $K = \mathbb{I}$.** We perform two experiments: **Experiment 1** with $(n, p) = (5000, 200)$ and **Experiment 2** with $(n, p) = (10000, 400)$ as discussed above. We run each experiment with 10 problem instances and compute the mean of the relative residual norm $\|Gx^k\|/\|Gx^0\|$. The results of this test are plotted in Figure 4.

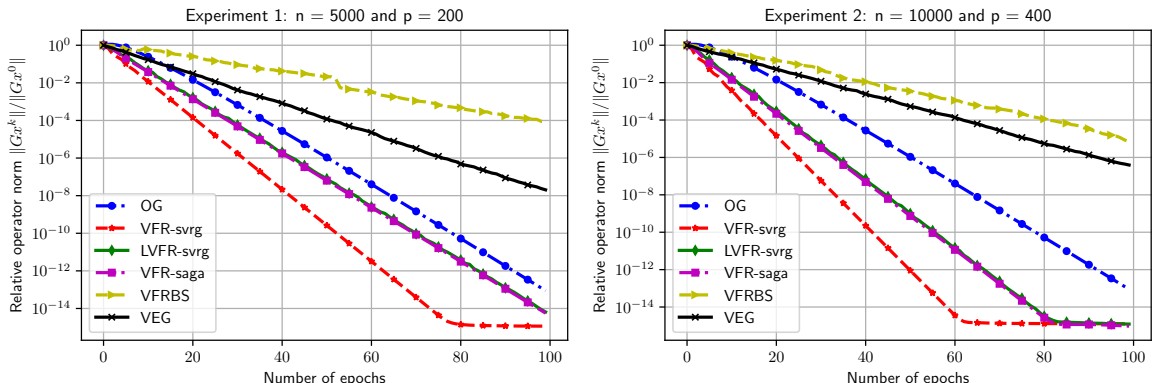

Figure 4: Performance of 6 algorithms to solve (37) on 2 experiments when $K = \mathbb{I}$.

For these particular experiments, our methods highly outperform OG, VFRBS, and VEG. It shows that VFR-svrg is the best overall, while LVFR-saga and VFR-svrg have a similar performance in both experiments. Both the competitors: VFRBS and VEG do not perform well in this test and they are much slower than ours and also OG. This is perhaps due to a small learning rate of VFRBS although we choose the same mini-batch size $b$ and the same probability $\mathbf{p}$ as ours.

(e) **Experiments for $K \neq \mathbb{I}$.** Now, we test these 6 algorithms for the case $K \neq \mathbb{I}$ in our extended model (37), where $K$ is generated randomly from the standard normal distribution. Then, we normalize $K$ as $K/\|K\|$ to get a unit Lipschitz constant $L = 1$.

Again, we use the same configuration as in Figure 4 and also run our experiments on 10 problems and report the mean results. We perform two experiments: **Experiment 1** with $n = 5000$ and $p_1 = p_2 = p = 100$, and **Experiment 2** with $n = 10000$ and $p_1 = p_2 = p = 200$. The results are reported in Figure 5.

We still observe that our algorithms work well and outperform their competitors. However, after 100 epochs, these methods can only reach a $10^{-2}$ accuracy level for an approximate solution.

E.1.2. THE UNCONSTRAINED CASE – VARYING $b$ AND $\mathbf{p}$
We can certainly tune the parameters to make our competitors (VFRBS) and (VEG) work better. However, such parameter configurations are far from satisfying the conditions of their theoretical results. For example, if we set $\mathbf{p} = \frac{20}{\sqrt{n}}$, then both VFRBS and VEG work better. In particular, if $n = 5000$, then we get $\mathbf{p} = \frac{20}{\sqrt{n}} = 0.28$, which is several times larger than its suggested value $\mathbf{p} = \frac{1}{n} = 2 \times 10^{-4}$.

Let us further experiment other choices of parameters (i.e. the mini-batch size $b$ and the probability $\mathbf{p}$ of flipping a coin) to observe the performance of these algorithms.

(a) **Larger $b$.** Figure 6 reveals the performance of these algorithms when we increase the mini-batch size $b$ to a larger value

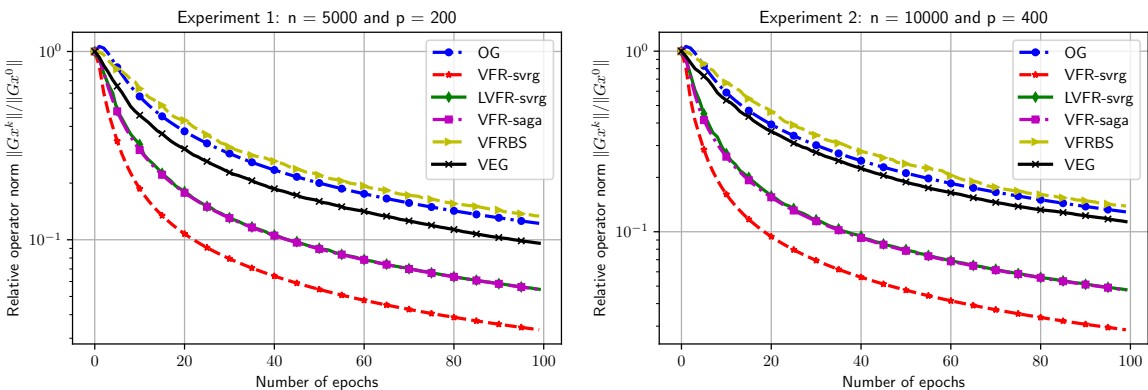

Figure 5: Performance of 6 algorithms to solve (37) on 2 experiments when $K \neq \mathbb{I}$.

$b = \lfloor 0.1n \rfloor$, while keeping the probability $\mathbf{p} = \frac{1}{n^{1/3}}$ unchanged.

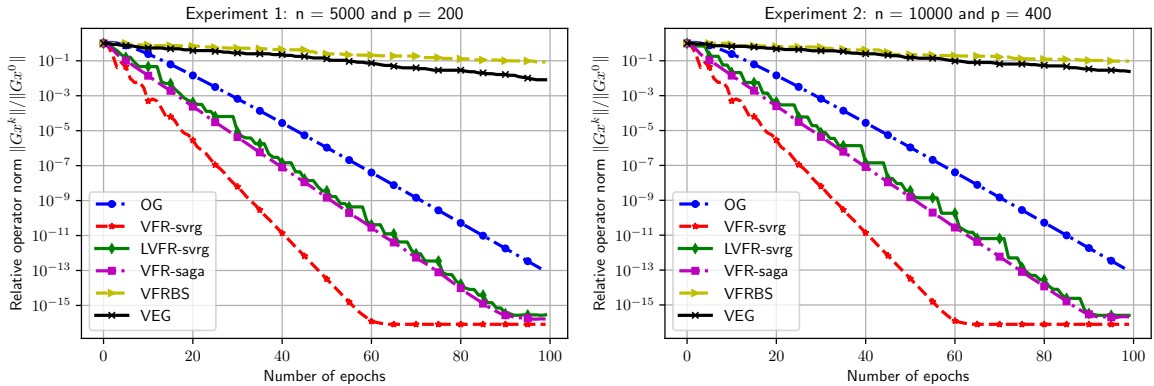

Figure 6: Performance of 6 algorithms for a large $b = \lfloor 0.1n \rfloor$ and a unchanged $\mathbf{p} = \frac{1}{n^{1/3}}$.

Note that for $n = 5000$, we have $b = 500$ and $\mathbf{p} = 0.058$, and for $n = 10000$, we have $b = 1000$ and $\mathbf{p} = 0.046$. With these large mini-batches, our algorithms still outperform other methods, while VFRBS and VEG are significantly slowed down. The double-loop variant of (VFR) with SVRG performs best, while LVFR-svrg and VFR-saga have a similar performance.

(b) **Medium** $b$ **and larger p.** Next, we set $b$ to a medium size of $b = \lfloor 0.05n \rfloor$ (corresponding to $b = 250$ for $n = 5000$ and $b = 500$ for $n = 10000$) and increase $\mathbf{p} = \frac{1}{n^{1/4}}$ (corresponding to $\mathbf{p} = 0.119$ for $n = 5000$ and $\mathbf{p} = 0.1$ for $n = 10000$). Then, the results are shown in Figure 7.

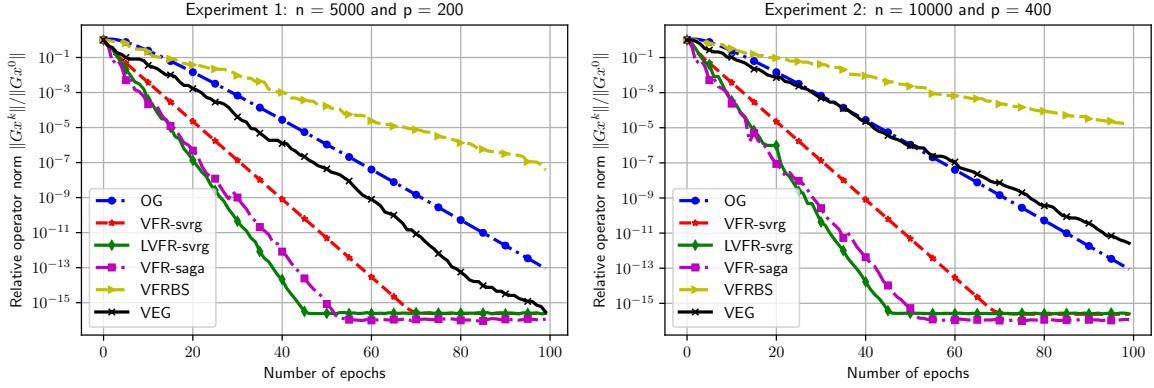

Figure 7: Performance of 6 algorithms for a medium $b = \lfloor 0.05n \rfloor$ and larger $\mathbf{p} = \frac{1}{n^{1/4}}$.

Then, we observe that `LVFR-svrg` and `VFR-saga` superiorly outperform the others. The performance of the double-loop `VFR-svrg` is still similar to the previous tests since it is not affected by **p**. In addition, `VEG` is now comparable with `OG`, but `VFRBS` remains the slowest one.

(c) **Large** $b$ **and small p.** To see the effect of **p** on our competitors: `VFRBS` and `VEG`, as suggested by their theory, we decrease **p** to **p** $= \frac{1}{n^{1/2}}$ (corresponding to **p** $= 0.014$ for $n = 5000$ and **p** $= 0.01$ for $n = 10000$) and still set $b = \lfloor 0.1n \rfloor$, and the results are plotted in Figure 8.

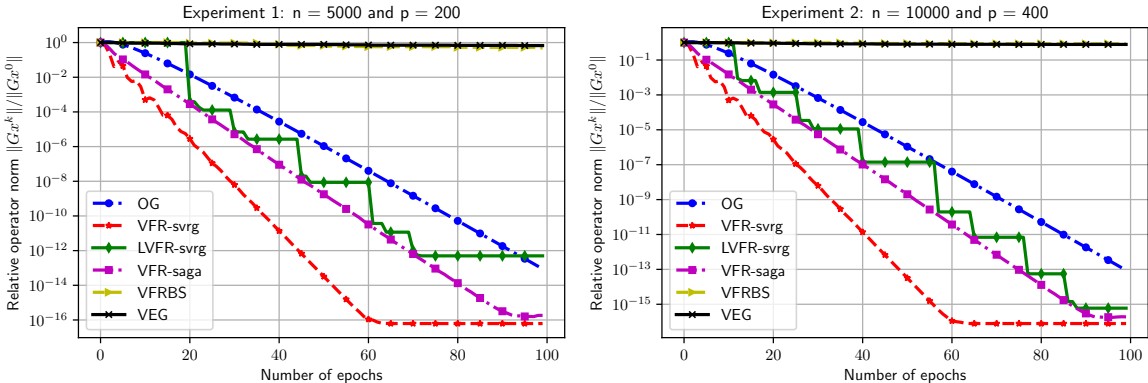

Figure 8: Performance of 6 algorithms for a large $b = \lfloor 0.1n \rfloor$ and a small **p** $= \frac{1}{n^{1/2}}$.

As we can observed from Figure 8, our methods highly outperform `VFRBS` and `VEG`, suggesting that these competitors require a larger probability to select the snap-shot point $w^k$ for full-batch evaluation. This is certainly not suggested in their theoretical results.

### E.1.3. THE CONSTRAINED CASE
Next, we choose $f(\theta) = \delta_{[-r,r]^{p_1}}(\theta)$ and $g(\beta) := \delta_{[-r,r]^{p_2}}(\beta)$ as the indicators of the $\ell_\infty$-balls of radius $r = 5$, respectively. In this case, we implement three variants of (VFRBS): the double-loop (`VFR-svrg`), the loopless (`LVFR-svrg`), and the SAGA (`VFR-saga`) variants to solve (NI) and compare against 3 algorithms as in the unconstrained case. Using the same data generating procedure as in the unconstrained case, we obtain the results as shown in Figure 9 when $K = \mathbb{I}$.

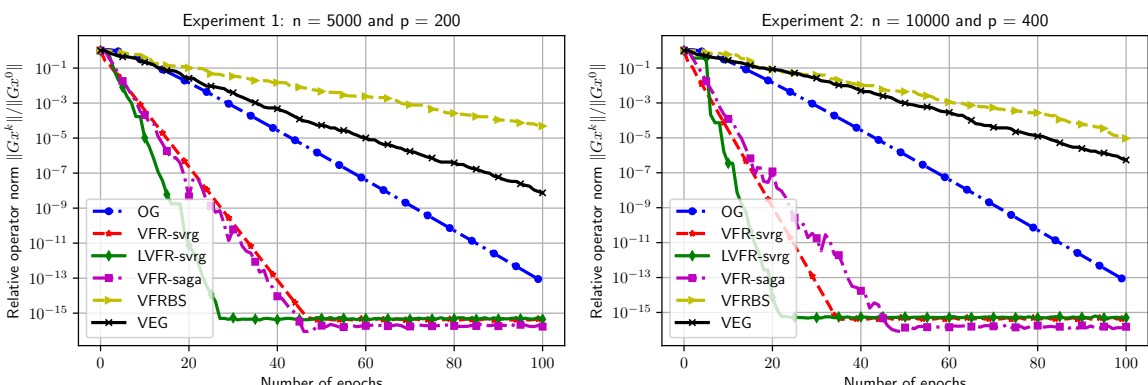

Figure 9: Comparison of 6 algorithms to solve constrained instances of (37) on 2 experiments when $K = \mathbb{I}$ (The average of 10 runs).

We see that the two SVRG variants of our (VFRBS): `VFR-svrg` and `LVFR-svrg`, as well as our `VFR-saga` variant remain working well compared to other methods. They superiorly outperform the three competitors.

Finally, we test our methods and their competitors for the case $K \neq \mathbb{I}$ as we done in Figure 5. Our results are plotted in Figure 10, where we observe a similar behavior as in Figure 5.

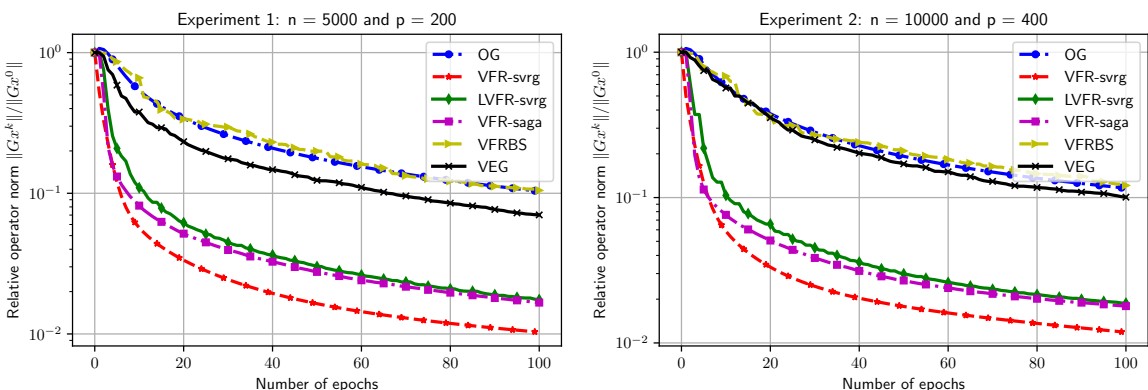

Figure 10: Comparison of 6 algorithms to solve constrained instances of (37) on 2 experiments when $K \neq \mathbb{I}$ (The average of 10 runs).

### E.2. Nonconvex-Nonconcave Quadratic Minimax Problems

We recall the nonconvex-nonconcave quadratic minimax optimization problem (15) in this subsection:

$$\min_{u \in \mathbb{R}^{p_1}} \max_{v \in \mathbb{R}^{p_2}} \left\{ \mathcal{L}(u,v) := \varphi(u) + \frac{1}{n} \sum_{i=1}^{n} \left[ u^T A_i u + u^T L_i v - v^T B_i v + b_i^\top u - c_i^\top v \right] - \psi(v) \right\}, \tag{38}$$

where $A_i \in \mathbb{R}^{p_1 \times p_1}$ and $B_i \in \mathbb{R}^{p_2 \times p_2}$ are symmetric matrices, $L_i \in \mathbb{R}^{p_1 \times p_2}$, $b_i \in \mathbb{R}^{p_1}$, $c_i \in \mathbb{R}^{p_2}$, and $\varphi = \delta_{\Delta_{p_1}}$ and $\psi = \delta_{\Delta_{p_2}}$ are the indicator of standard simplexes in $\mathbb{R}^{p_1}$ and $\mathbb{R}^{p_2}$, respectively.

Let us first define $x := [u, v] \in \mathbb{R}^p$ as the concatenation of the primal and dual variables $u$ and $v$, where $p := p_1 + p_2$. Next, we define

$$G_i x = \mathbf{G}_i x + \mathbf{g}_i := \begin{bmatrix} A_i & L_i \\ -L_i & B_i \end{bmatrix} \begin{bmatrix} u \\ v \end{bmatrix} + \begin{bmatrix} b_i \\ c_i \end{bmatrix} = \begin{bmatrix} A_i u + L_i v + b_i \\ -L_i u + B_i v + c_i \end{bmatrix}, \quad \text{and} \quad T := \begin{bmatrix} \partial \varphi \\ \partial \psi \end{bmatrix}.$$

Then, we denote $\mathbf{G}_i := \begin{bmatrix} A_i & L_i \\ -L_i & B_i \end{bmatrix}$, and $\mathbf{g}_i := \begin{bmatrix} b_i \\ c_i \end{bmatrix}$. Clearly, $G_i(\cdot)$ is an affine mapping from $\mathbb{R}^p$ to $\mathbb{R}^p$, but $\mathbf{G}_i$ is nonsymmetric. Let $Gx := \frac{1}{n} \sum_{i=1}^{n} G_i x = \left( \frac{1}{n} \sum_{i=1}^{n} \mathbf{G}_i \right) x + \frac{1}{n} \sum_{i=1}^{n} \mathbf{g}_i = \mathbf{G}x + \mathbf{g}$, where $\mathbf{G} := \frac{1}{n} \sum_{i=1}^{n} \mathbf{G}_i$ and $\mathbf{g} := \frac{1}{n} \sum_{i=1}^{n} \mathbf{g}_i$. Then, the optimality condition of (38) becomes $0 \in Gx + Tx$, which is exactly in the form (NI). Clearly, if $A_i$ and/or $B_i$ are not positive semidefinite, then (38) possibly covers nonconvex-nonconcave minimax optimization instances.

#### E.2.1. THE UNCONSTRAINED CASE

We consider the case $\varphi = 0$ and $\psi = 0$, leading to an unconstrained setting of (38), i.e. $T = 0$ as considered in (15) of the main text. Hence, the optimality condition of (38) reduces to $Gx = 0$, which is of the form (NE).

(a) **How to generate data?** To run our experiments, we generate synthetic data as follows. First, we fix the dimensions $p_1$ and $p_2$ and the number of components $n$. We generate $A_i = Q_i D_i Q_i^T$ for a given orthonormal matrix $Q_i$ and a diagonal matrix $D_i = \text{diag}(D_i^1, \cdots, D_i^{p_1})$, where its elements are generated from standard normal distribution and clipped its negative entries as $\max\{D_i^j, \varepsilon\}$ for $j = 1, \cdots, p_1$ and $\varepsilon := -0.1$. This choice of $A_i$ guarantees that $A_i$ is symmetric, but possibly not positive semidefinite. The matrix $B_i$ is also generated by the same way. The pay-off matrix $L_i$ is an $p_1 \times p_2$ matrix, which is also generated from the standard normal distribution for all $i \in [n]$. The vectors $b_i$ and $c_i$ are generated from the standard normal distribution for $i \in [n]$. With this data generating procedure, $\mathbf{G}_i$ is not symmetric and possibly not positive semidefinite.

(b) **Algorithms.** We again test 6 algorithms: two variants (double-loop SVRG – `VFR-svrg`) and (loopless SVRG – `LVFR-svrg`) of (VFR), our (VFR) with SAGA estimator (`VFR-saga`), `VFRBS` from (Alacaoglu et al., 2023), `VEG` from (Alacaoglu & Malitsky, 2022), and `OG` (the standard optimistic gradient method), e.g., from (Daskalakis et al., 2018).

(c) **The details of Subsection 5.1 in Section 5.** First, we provide the details of **Subsection** 5.1 in Section 5. The purpose of this example is to verify our theoretical results stated in Corollaries 3.2 and 3.3.

For the SVRG estimator, let us first choose $\gamma := 0.75$, $b := \lfloor n^{2/3} \rfloor$, and $\mathbf{p} := \frac{1}{n^{1/3}}$ as suggested by Corollary 3.2. Then, we can directly compute $\eta := \frac{1}{L\sqrt{M}}$, where $\Lambda := \frac{6.25(2-3\mathbf{p})+4.125\mathbf{p}^2}{b\mathbf{p}^2}$ and $M = 2.375 + \frac{11}{3}\Lambda$. Clearly, if $n = 5000$, then $\eta = \frac{0.146153}{L}$. Alternatively, if $n = 10000$, then $\eta = \frac{0.148934}{L}$. These learning rates are used in our experiments plotted in Figure 1.

Similarly, for the SAGA estimator, we also choose $\gamma := 0.75$ and $b := \lfloor n^{2/3} \rfloor$. In this case, by Corollary 3.3, we can also directly compute $\eta := \frac{1}{L\sqrt{M}}$. If $n = 5000$, then $\eta = \frac{0.146153}{L}$. Alternatively, if $n = 10000$, then $\eta = \frac{0.145693}{L}$. These learning rates are used in `VFR-saga`.

Note that since the theoretical value of $\mathbf{p}$ in `VFRBS` and `VEG` is too small, we instead choose $\mathbf{p} := \frac{1}{n^{1/3}}$ and also $b := \lfloor n^{2/3} \rfloor$ as in our methods. Then, we compute the learning rate $\eta$ of these methods based on the formula given in (Alacaoglu et al., 2023) for `VFRBS` and (Alacaoglu & Malitsky, 2022) for `VEG`, respectively.

(d) **Results for a different set of parameters.** Unlike **Subsection** 5.1 in the main text, we choose the parameters for these algorithms as in Subsection E.1. The 6 algorithms are run on 2 experiments. The first experiment is with $n = 5000$ and $p_1 = p_2 = 50$, while the second one is with $n = 10000$ and $p_1 = p_2 = 100$. These experiments are run 10 times, corresponding to 10 problem instances, and the average results are reported in Figure 11 in terms of the relative operator norm $\|Gx^k\|/\|Gx^0\|$ against the number of epochs.

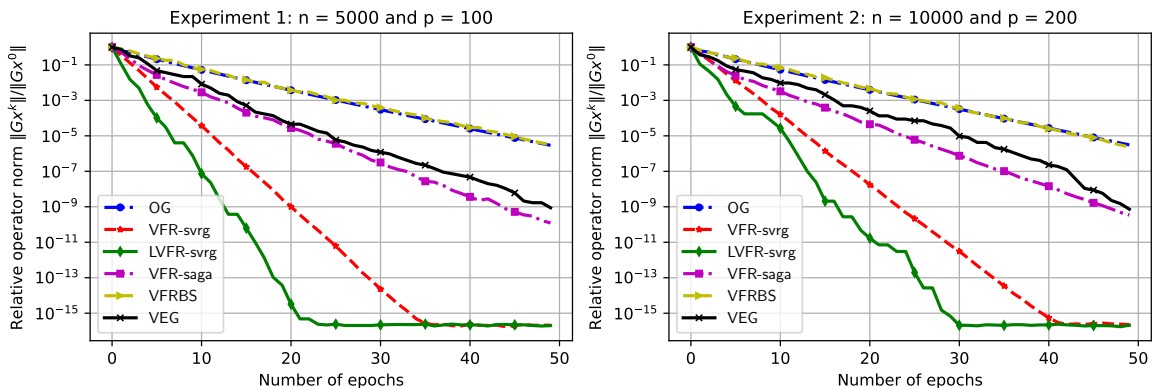

Figure 11: The performance of 6 algorithms to solve the unconstrained case of (38) on 2 experiments (The average of 10 runs).

Clearly, under this configuration, both SVRG variants of our methods work well and significantly outperform other competitors. The loopless SVRG variant (`VFR-svrg`) of (`VFR`) seems to work best, while our `VFR-saga` has a similar performance as `VEG`. We also see that `VFRBS` has a similar performance as `OG`.

To improve the performance of these competitors, especially, `VFRBS` and `VEG`, one can tune their parameters as in Subsection E.1, where the probability $\mathbf{p}$ of updating the snapshot point $w^k$ is increased. However, with such a choice of $\mathbf{p}$, its value is often greater or equal to $0.5$, making these methods to be closed to deterministic variants. Hence, their theoretical complexity bounds are no longer improved over the deterministic counterparts.

E.2.2. THE CONSTRAINED CASE

We conduct two more experiments for the constrained case of (38) as in the main text when $u \in \Delta_{p_1}$ and $v \in \Delta_{p_2}$, where $\Delta_p := \{u \in \mathbb{R}^p_+ : \sum_{i=1}^p u_i = 1\}$ is the standard simplex in $\mathbb{R}^p$.

We run 6 algorithms for solving the constrained case of (38) using the same parameters as Subsection 5.1, but with larger problems. We report the relative norm of the FBS residual $\|\mathcal{G}_\eta x^k\|/\|\mathcal{G}_\eta x^0\|$ against the number of epochs. The results are revealed in Figure 12 for two datasets $(p, n) = (500, 5000)$ and $(p, n) = (300, 10000)$.

With these two additional experiments, both SVRG variants of our method (`VFRBS`) again work well and significantly outperform other competitors. The loopless SVRG variant (`VFR-svrg`) of (`VFRBS`) tends to work best, while our `VFR-saga` has a relatively similar performance as `VEG`. We also see that `VFRBS` has a similar performance trend as `OG`,

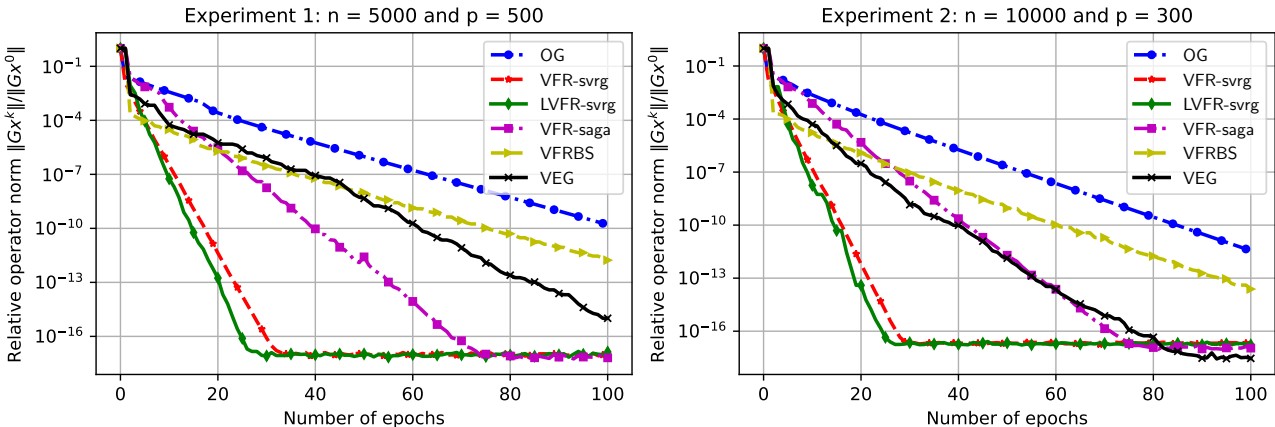

Figure 12: The performance of 6 algorithms to solve the constrained case of (38) on 2 experiments (The average of 10 runs).

but works better.

### E.3. The $\ell_1$-Regularized Logistic Regression with Ambiguous Features

This Supp. Doc. provides the details of **Subsection** 5.2 in Section 5 in the main text.

(a) **Model.** We consider a standard regularized logistic regression model associated with a given dataset $\{(\hat{X}_i, y_i)\}_{i=1}^N$, where $\hat{X}_i$ is an i.i.d. sample of a feature vector and $y_i \in \{0, 1\}$ is the associated label of $\hat{X}_i$. Unfortunately, $\hat{X}_i$ is ambiguous, i.e. it belongs to one of $m$ possible examples $\{X_{ij}\}_{j=1}^m$. Since we do not know $\hat{X}_i$ to evaluate the loss, we consider the worst-case loss $f_i(w) := \max_{1 \le j \le m} \ell(\langle X_{ij}, w \rangle, y_i)$ computed from $m$ examples, where $\ell(\tau, s) := \log(1 + \exp(\tau)) - s\tau$ is the standard logistic loss.

Using the fact that $\max_{1 \le j \le m} \ell_j(\cdot) = \max_{z \in \Delta_m} \sum_{j=1}^m z_j \ell_j(\cdot)$, where $\Delta_m$ is the standard simplex in $\mathbb{R}^m$, we can model this regularized logistic regression into the following minimax problem:

$$\min_{w \in \mathbb{R}^d} \max_{z \in \mathbb{R}^m} \left\{ \mathcal{L}(w, z) := \frac{1}{N} \sum_{i=1}^N \sum_{j=1}^m z_j \ell(\langle X_{ij}, w \rangle, y_i) + \tau R(w) - \delta_{\Delta_m}(z) \right\}, \tag{39}$$

where $\ell(\tau, s) := \log(1 + \exp(\tau)) - s\tau$ is the standard logistic loss, $R(w) := \|w\|_1$ is an $\ell_1$-norm regularizer, $\tau > 0$ is a regularization parameter, and $\delta_{\Delta_m}$ is the indicator of $\Delta_m$ that handles the constraint $z \in \Delta_m$. This problem is exactly the one stated in (16) of the main text.

First, let us denote $x := [w; z] \in \mathbb{R}^p$ as the concatenation of $w$ and $z$ with $p = d + m$, and

$$G_i x := \begin{bmatrix} \sum_{j=1}^m z_j \ell'(\langle X_{ij}, w \rangle, y_i) X_{ij} \\ -\ell(\langle X_{i1}, w \rangle, y_i) \\ \dots \\ -\ell(\langle X_{im}, w \rangle, y_i) \end{bmatrix} \quad \text{and} \quad Tx := \begin{bmatrix} \tau \partial R(w) \\ \partial \delta_{\Delta_m}(z) \end{bmatrix},$$

where $\ell'(\tau, s) = \frac{\exp(\tau)}{1 + \exp(\tau)} - s$. Then, the optimality condition of (39) can be written as (NI): $0 \in Gx + Tx$, where $Gx := \frac{1}{n} \sum_{i=1}^n G_i x$.

(b) **Input data.** We test our algorithms and their competitors on two real datasets: `a9a` (134 features and 3561 samples) and `w8a` (311 features and 45546 samples) downloaded from `LIBSVM` (Chang & Lin, 2011). For a given nominal dataset $\{(\hat{X}_i, y_i)\}_{i=1}^n$, we first normalize the feature vector $\hat{X}_i$ such that its column norm is one, and then add a column of all ones to address the bias term. To generate ambiguous features, we take the nominal feature vector $\hat{X}_i$ and add a random noise generated from a normal distribution of zero mean and variance of $\sigma = 0.5$. In our test, we choose $\tau := 10^{-3}$ and $m := 10$ for all the experiments.

(c) **Algorithms.** As before, we implement 3 variants of our method (VFRBS): `VFR-svrg`, `LVFR-svrg`, and `VFR-saga`

to solve (39). We also compare them with OG, VFRBS, and VEG. We choose $x^0 := 0.5 \cdot \text{ones}(p)$ in all experiments. We run all the algorithms for 100 epochs and report the relative FBS residual norm $\|\mathcal{G}_\eta x^k\|/\|\mathcal{G}_\eta x^0\|$ against the epochs.

(d) **Parameters.** Since it is very difficult to estimate the Lipschitz constant $L$ of $G$, we are unable to set a correct learning rate $\eta$ in the underlying algorithms. We instead compute an estimation $\hat{L} := \|\hat{X}\|$, and then set $\eta := \frac{\omega}{\hat{L}}$, by tuning $\omega$ for each algorithm. More specifically, after tuning, we obtain the following configuration.

- For the three variants of (VFRBS): VFR-svrg, LVFR-svrg, and VFR-saga, we set $\eta = \frac{25}{\hat{L}}$ for a9a and $\eta = \frac{50}{\hat{L}}$ for w8a.
- For OG, we set $\eta = \frac{50}{\hat{L}}$ for a9a and $\eta = \frac{100}{\hat{L}}$ for w8a.
- For VFRBS, we choose $\eta = \frac{47.5(1-\sqrt{1-\mathbf{p}})}{2\hat{L}}$ for a9a and $\eta = \frac{95(1-\sqrt{1-\mathbf{p}})}{2\hat{L}}$ for w8a.
- For VEG, we select $\eta = \frac{47.5\sqrt{1-\alpha}}{\hat{L}}$ for a9a and $\eta = \frac{95\sqrt{1-\alpha}}{\hat{L}}$ for w8a with $\alpha := 1 - \mathbf{p}$.

We still choose the mini-batch size $b$ and the probability $\mathbf{p}$ of updating the snapshot point $w^k$ in SVRG variants as $b = \lfloor 0.5n^{2/3} \rfloor$ and $\mathbf{p} = n^{-1/3}$, respectively for all the algorithms.

We conduct two more experiments using the well-known MNIST dataset ($n = 70000$ and $p = 780$) where we want to classify the even and odd numbers into two different classes, respectively. We use the same parameter selection as in the experiment with the a9a dataset. In the first experiment, we choose $m = 10$ and the variance of noise $\sigma = 0.5$. In the second experiment, we choose $m = 20$ and the variance of noise $\sigma = 0.25$. We only run 5 algorithms and leave out the VFRBS method since we have not managed to find the parameters that make it work stably. The result of this experiment is shown in Figure 13.

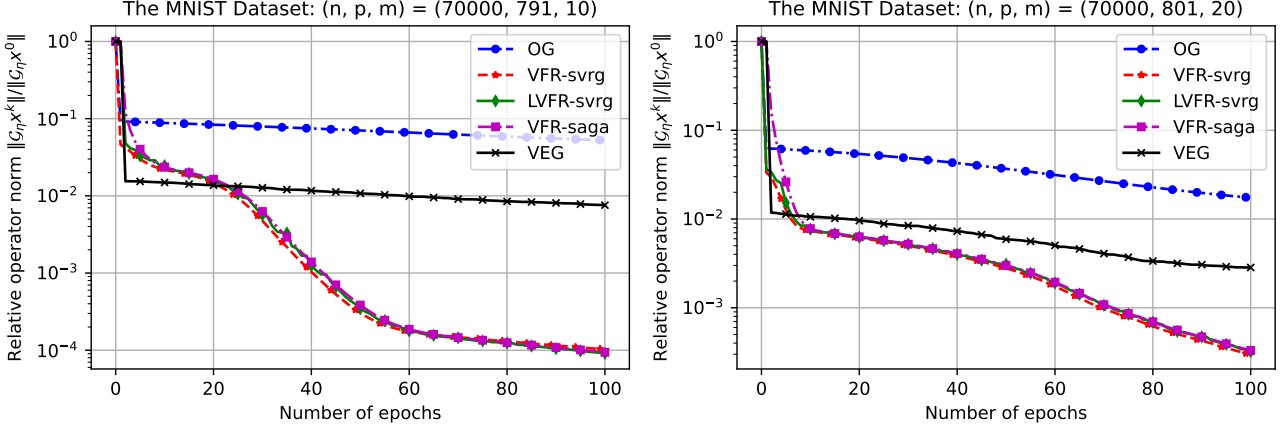

Figure 13: Comparison of 6 algorithms to solve (16) on the real dataset: MNIST.

We can see from Figure 13 again that three variants VFR-svrg, LVFR-svrg, and VFR-saga have similar performance and are much better than their two competitors. Here, VEG is better than OG, but both methods are slower than ours.

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
