# OpenReview forum: "Variance-Reduced Forward-Reflected-Backward Splitting Methods for Nonmonotone Generalized Equations"
_ICML.cc/2025/Conference — ICML 2025 poster_

### Official Review · Reviewer_Q5J9 · 2025-03-04

**Overall Recommendation:** 3

**Summary:**

This paper proposes two stochastic variance-reduction algorithms to solve a class of nonmonotone equations. The key technical tool used in this paper is the intermediate object $S_{\gamma}$.
The authors apply classical variance reduction techniques on $S_{\gamma}$ instead of the operator $G$, and they show their proposed algorithms obtain convergence rates of $\mathcal{O}(n+n^{2/3}\epsilon^{-2})$.

## update after rebuttal
The author's reply makes sense, so I keep my score.

**Claims And Evidence:**

Strength:

* The authors provide a theoretical analysis of their proposed approaches and show their algorithm enjoys a convergence rate of $\mathcal{O}(n+n^{2/3}\epsilon^{-2})$.

* Applying variance reduction on the intermediate object $S_{\gamma}$ is a novel and interesting idea.

Weakness:

* Paper organization is not good enough. The comparison with (Cai et al. 2023) can be presented in the main text instead. The comparison with existing work can be summarized in a table. The author may consider highlighting the co-coercive condition presented in (Cai et al. 2023) in the main text.

* The convergence rate of the proposed methods is inferior to the methods obtained in (Cai et al. 2023). Could the authors recover the $\mathcal{O}(n+n^{1/2}\epsilon^{-2})$ convergence rate if the co-coercive assumption is imposed?

*  In Theorem 3.1, the choice of $\kappa$ seems restrictive.

* In the experiments, the author may consider comparing with the methods proposed in (Cai et al. 2023).

**Essential References Not Discussed:**

I think the most relevant references have been discussed/

**Experimental Designs Or Analyses:**

See above.

**Methods And Evaluation Criteria:**

See above.

**Other Comments Or Suggestions:**

* In Corollary 3.3, line 324, why there is a $p$ in the denominator? The SAGA  estimator no longer needs probability $p$.

* A table summarization of existing work with different assumptions and convergence criteria could be provided.

**Other Strengths And Weaknesses:**

Some other weaknesses include:

* In both VFR-SVRG and VFR-SAGA algorithms, the minibatch size is quite large. It is of the order $\mathcal{O}(n^{2/3})$, which is larger than most minibatch sizes of the stochastic methods solving nonconvex optimization.

* The choice of $\gamma$ is quite restrictive. It must belong to the range of $(1/2, 1)$, so it cannot recover the classic FRBS and optimistic gradient methods. Could the authors provide some intuition as to why $\gamma$ cannot be chosen $1/2$ for the proposed approaches?

**Questions For Authors:**

See above.

**Relation To Broader Scientific Literature:**

It is an interesting work for solving large-scale nonmonotone equations. Overall, the idea of applying variance reduction on the intermediate object is novel and interesting.

**Theoretical Claims:**

See above.

---

> ### Author Rebuttal · Authors · 2025-03-25
>
> First of all, we acknowledge the reviewers for his/her comments and feedback on our work. Below is our response to each point.
>
> P1. Weakness:
>  + Q1.1: Paper organization is not good enough. The comparison with (Cai et al. 2023) can be presented in the main text instead. The comparison with existing work can be summarized in a table. The author may consider highlighting the co-coercive condition presented in (Cai et al. 2023) in the main text.
> > R1.1: Thank you for your comment. We will try to improve the organization of the paper as suggested. We will bring the comparison with [Cai et al 2023] back to the main text, and highlight the co-coercivity condition used in [Cai et al 2023] vs. our assumptions. This co-coercivity condition is stronger than monotonicity+Lipschitz, and quite limited since it does not cover bilinear matrix games. We will also add a table summarizing the comparison with existing works.
> + Q1.2: The convergence rate of the proposed methods is inferior to the methods obtained in (Cai et al. 2023). Could the authors recover the convergence rate if the co-coercive assumption is imposed?
> > R1.2: In terms of convergence rates, our rate is slower than [Cai et al. 2023] ($1/k$ vs. $1/k^2$) when solving co-coercive equations. However, since our assumptions are significantly weaker, it is unclear if this theoretical comparison is fair.
> Note that Halpern's method in [Cai et al 2023] is a type of accelerated method, while our method is not. We believe that $O(1/k)$ is the best rate one can get for non-monotone problems, similar to nonconvex optimization methods, though we do not have a lower bound result to compare at this moment. In nonconvex optimization, this rate is optimal (up to a constant factor).
> + Q1.3: In Theorem 3.1, the choice of $\kappa$ seems restrictive.
> > A1.3: So far, the range of $\kappa$ is indeed restrictive since we did not focus on optimizing $M$ and $\gamma$ in Theorems 3.1 and 4.1. In deterministic methods, the largest range of $L\kappa$ is $L\kappa < 1$. We believe that our range on $L\kappa$ can also be improved, but it requires substantial changes in the algorithmic design and analysis.
> + Q1.4: In the experiments, the author may consider comparing with the methods proposed in (Cai et al. 2023).
> > A1.4: Yes, we will add some examples to compare with [Cai et al 2023] for co-coercive problems as suggested.
>
> P2: Other Strengths And Weaknesses:
> + Q2.1: In both VFR-SVRG and VFR-SAGA algorithms, the minibatch size is quite large. It is of the order $\mathcal{O}(n^{2/3})$, which is larger than most minibatch sizes of the stochastic methods solving nonconvex optimization.
> > A2.1: This choice of mini-batch (i.e., $b = \mathcal{O}(n^{2/3})$) is to attain the complexity of $\mathcal{O}(n + n^{2/3}\epsilon^2)$.
> Our theory works with any batch size $b \in [1, n]$. However, the choice of $b$ affects the final oracle complexity. Therefore, in practice, we can choose any batch size, and our method still converges, but the overall oracle complexity will be worse than the best one $\mathcal{O}(n + n^{2/3}\epsilon^2)$. We will add a brief discussion on this aspect in the revision.
>
> + Q2.2: The choice of $\gamma$ is quite restrictive. It must belong to the range of $(1/2, 1)$, so it cannot recover the classic FRBS and optimistic gradient methods. Could the authors provide some intuition as to why cannot be chosen for the proposed approaches?
> > A2.2: As can be seen from (22) of Lemma C.1., if $\gamma = 1/2$, then we must impose $\mu = 0$ and $\kappa = 0$, leading to star-monotonicity in Assumption 1.4. Our theory still works, but no longer covers Assumption 1.4. with $\kappa > 0$, but covers monotone+Lipschitz problems. As far as we know, existing FRBS methods require monotonicity, which is consistent with our theory when choosing $\gamma = 1/2$. We will add a discussion of this special case.
>
> P3: Other Comments Or Suggestions:
> + Q3.1: In Corollary 3.3, line 324, why there is a  in the denominator? The SAGA estimator no longer needs probability.
> >A3.1.: Thank you. It is indeed a typo. We will correct it.
> + Q3.2: A table summarization of existing work with different assumptions and convergence criteria could be provided.
> >A3.2: We will add a table to summarize existing works with assumptions and convergence criteria as suggested.
>
> P4: Overall Recommendation: 3: Weak accept (i.e., leaning towards accept, but could also be rejected)
> >A4: We hope we have addressed your concerns. We will implement what we have promised above.
> Given our novel methods and theoretical contributions, and the effort of addressing your concerns, your re-evaluation is highly appreciated.

---

### Official Review · Reviewer_9taN · 2025-03-09

**Overall Recommendation:** 3

**Summary:**

1. inspired by  SVRG & SAGA, construct two variance-reduced estimators for the forward-reflect operator
2. show that VFR and VFRBS methods achieve SOTA oracle complexity for non-monotone operator splitting problems

**Claims And Evidence:**

1. Does the convergence of your splitting algorithm have strong connection with your estimator? If I replace SVRG/SAGA with anothor variance-reduced estimators, will the algorithm converge?
2. Could you provide a table to compare with the existing literature, in term of problem setup and sample complexity?

**Essential References Not Discussed:**

N/A

**Experimental Designs Or Analyses:**

1. Solving non-monotone problem is one of the main differences compared to the existing works. However, those examples didn't explain why they are non-monotone.
2. I understand that it is hard to do theoretical comparison to those works requiring monotonicity assumption, e.g. [Cai et al., 2023]. Could you provide some empirical comparison? Since the monotonicity assumption is violated for those approaches, one can expect slow convergence or even diverge behavior.

**Methods And Evaluation Criteria:**

The authors claimed their work is to solve non-monotone, non-smooth, and large-scale problem arising from generative ML, adversarial learning, and robust learning. However, their numerical examples are limited to synthetic or toy example whose scale is very small.

**Other Comments Or Suggestions:**

I think this paper is too dense in technique presentation while leaving very little room for its possible applications in ML. For example, what kind of non-convex problem satisfying Assumption 1.4?

**Other Strengths And Weaknesses:**

Strength:
1. solid theoretic work

Weakness:
1. very dense theory and very weak experiments
2. the motivating ML applications, e.g. GAN and adversarial training, are not the main line for generative modeling and robust deep learning.

**Questions For Authors:**

N/A

**Relation To Broader Scientific Literature:**

Exploring the application of operator splitting in non-monotone and stochastic setup could be useful for broader engineering problems than ML learning.

**Theoretical Claims:**

I didn't check the proofs.

---

> ### Author Rebuttal · Authors · 2025-03-25
>
> First of all, we highly acknowledge the comments and questions from the reviewer.
> Below is our detailed response.
>
> Q1. "Does the convergence ... estimator? If I replace SVRG/SAGA ...  converge?
> >R1. The answer is "no". In lines 233-240 and 297-304, we have stated that any estimator $S^k$ satisfies Definition 2.1. can be used in our methods, covering a wide class of unbiased estimators. The convergence guarantees are stated in Theorems 3.1 and 4.1 without specifying $S^k$. Therefore, any estimator satisfying Definition 2.1. can be used in our methods to have convergence guarantees as in these theorems. SVRG and SAGA are two concrete instances of $S^k$. However, the oracle complexity depends on the specific estimator since we do need to know the cost of constructing such an estimator (See Subsections 3.2 and 4.2).
>
> Q2. "Could you provide a table to compare with the existing literature, in terms of problem setup and sample complexity?"
> >R2.  Thank you for the request. Yes, we will provide a table summarizing existing results and comparing them with ours.
>
> Q3. "The authors claimed ... very small."
> >R3. Our paper primarily focuses on new algorithms, theoretical convergence, and oracle complexity rather than experiments. Our experiments are used to validate the theory and we did not specifically focus on concrete ML applications. Nevertheless, based on your suggestion, we will increase the sizes of our experiments in the revision.
>
> >Regarding nonsmooth and non-monotone problems, Supp. E2 provides nonsmooth and non-monotone problems since the underlying matrix is not necessarily positive semidefinite (non-monotone), and they have constraints (nonsmoothness).
> Example 2 tackles a nonlinear minimax problem, which is nontrivial compared to the quadratic minimax one in Example 1. As mentioned above, we will substantially increase the size of our experiments as you suggested.
>
> Q4: "Solving non-monotone problem ... are non-monotone."
> >R4: Example 1 allows $\mathbf{G}$ to be non-positive semidefinite. Therefore, it violates the condition $\langle Gx - Gy, x - y\rangle \geq 0$ (since $\langle Gx - Gy, x - y\rangle = (x-y)^T\mathbf{G}(x - y)$).
> This means that $G$ is non-monotone. We will clearly explain this in the revision.
>
> Q5: "I understand that it is hard to do a theoretical comparison ... even diverge behavior."
> >R5: The work by (Cai et al 2023) relies on Halpern's fixed-point iteration, and indeed requires stronger assumptions than ours. We will conduct numerical examples to compare both methods, at least in the co-coercivity case. We are not sure if it will diverge in the non-monotone case, but there is no theoretical result to guarantee convergence for this method under Assumption 1.4.
>
> Q6: Weakness:
> -- very dense theory and very weak experiments
> -- the motivating ML applications, e.g. GAN and adversarial training, are not the main line for generative modeling and robust deep learning.
> >R6.1: We agree with the reviewer that the paper is dense, and we will try to improve its presentation as suggested. It was significantly challenging for us to conduct a 10-page paper with both strong theories and experiments. Therefore, we chosed to focus on methodology and theoretical results in our submission. However, as we discussed above, additional experiments will be added to the supplementary in the revision.
>
> >R6.2: We will rephrase some inaccurate statements related to robust deep learning and generative modeling as suggested.
>
> Q7: "I think this paper is too dense ... applications in ML.
> For example, what kind of non-convex problem satisfies Assumption 1.4?"
> >R7: We will improve the presentation of our paper as suggested. We will add more experiments to the second example to handle nonconvex regularizers such as CAD, to obtain non-monotone problems.
> Regarding Assumption 1.4., one simple example is to consider a nonconvex quadratic problem $\min_x \frac{1}{2}x^TQx + q^Tx$, where $Q$ is symmetric and invertible, but not necessarily positive semidefinite. The optimality condition is $Qx + q = 0$.
> If we define $Gx = Qx + q$ and $T = 0$, then $\langle Gx - Gy, x - y\rangle = (x-y)^TQ(x-y) \geq \lambda_{\min}(Q^{-1})\Vert Q(x-y)\Vert^2 =  \lambda_{\min}(Q^{-1})\Vert Gx - Gy\Vert^2$. Hence, if $\lambda_{\min}(Q^{-1}) < 0$, then it satisfies Assumption 1.4. with $\rho := -\lambda_{\min}(Q^{-1}) > 0$.
> Note that one can verify that our examples in Section 5 and SupDoc. E satisfy Assumption 1.4. We will add a paragraph to explain this case.
>
> Q8: Overall Recommendation: 2: Weak reject (i.e., leaning towards reject, but could also be accepted)
> >R8: We hope the reviewer will read our responses above, and re-evaluate our work based on new algorithms (especially, the one for inclusion) and theoretical contributions. We promise to improve our paper and add a table for comparison, larger experiments, and a comparison to [Cai et al 2023] as requested.

---

### Official Review · Reviewer_tBMV · 2025-03-13

**Overall Recommendation:** 4

**Summary:**

This paper studies the forward-reflected operator in two types of variance-reduced estimators: SVRG and SAGA. Using these estimators, the authors propose the Variance-Reduced Forward-Reflected Method and the Variance-Reduced Forward-Reflected-Backward Splitting, which solve nonlinear and generalized equations, respectively. Through Lyapunov analysis, the authors establish an complexity of $O(n+n^{2/3}\epsilon^{-2})$ to obtain $\epsilon$-residual error, matching the best-known results. Theoretical findings are further validated through numerical experiments.

**Claims And Evidence:**

The authors state that their results match the best known results. I request that authors verify this by providing a table or a comparison with prior works in terms of complexity and assumptions. This is necessary and would enhance the novelty and clarity.

**Essential References Not Discussed:**

I believe this papers address prior works.

**Experimental Designs Or Analyses:**

Analyses and experiments look sound.

**Methods And Evaluation Criteria:**

Yes, methods and evaluation criteria makes sense.

**Other Comments Or Suggestions:**

Please refer to the Questions for Authors section.

**Other Strengths And Weaknesses:**

This paper systematically studies variance-reduced estimators, SVRG and SAGA estimators of forward-reflected operator and proposes new algorithms with theoretical complexity matched to the best known results. The paper is well organized, and the experiments demonstrate the theoretical results. Furthermore, it shows that proposed algorithms perform better than ones in prior work. I believe this work provides a valid contribution to the literature on stochastic approximation and nonmonotone inclusion problems, since, although their theoretical results do not improve upon prior work, this paper introduces an interesting new operator with symmetrical variance reduction analyses and demonstrates its practical potential.

**Questions For Authors:**

Do the complexities of the proposed methods in this paper indeed match the best-known results? (As I wrote in Claims and Evidence section, please make table or a comparison with prior works in terms of complexity and assumptions.)

**Relation To Broader Scientific Literature:**

As the authors introduced, this work falls within the literature on stochastic approximation and nonmonotone inclusion problems.

**Theoretical Claims:**

I briefly reviewed the proofs in both the main text and the supplementary material.

---

> ### Author Rebuttal · Authors · 2025-03-25
>
> First of all, we highly acknowledge the reviewer for constructive comments and feedback. Below is our response to each point.
>
> Q1: The authors ...  results. I request that authors verify this by providing a table or a comparison with prior works in terms of complexity and assumptions. This is necessary and would enhance the novelty and clarity.
> >R1: Thank you! Yes, we will provide a table to summarize and compare our work with existing results. We will also highlight our novelty using this summarized table as suggested.
>
> Q2: This paper systematically studies variance-reduced estimators, SVRG and SAGA estimators of forward-reflected operator and proposes new algorithms with theoretical complexity matched to the best known results. The paper is well organized, and the experiments demonstrate the theoretical results. Furthermore, it shows that proposed algorithms perform better than ones in prior work. I believe this work provides a valid contribution to the literature on stochastic approximation and nonmonotone inclusion problems, since, although their theoretical results do not improve upon prior work, this paper introduces an interesting new operator with symmetrical variance reduction analyses and demonstrates its practical potential.
> >R2: We appreciate your adequate summary of our paper and contribution. The complexity we achieved in this paper is similar to the one in variance-reduced methods for nonconvex optimization using SVRG and SAGA, and this complexity is unimprovable without using enhancement tricks such as restart or nested loop.
>
> Q3: Do the complexities of the proposed methods in this paper indeed match the best-known results?
> >R3: To our best knowledge, our oracle complexity results are the best-known so far for non-monotone problems (i.e., covered by Assumptions 1.3-1.4) using SVRG and SAGA. We will add a table to compare as suggested.

---

> > ### Comment · Reviewer_tBMV · 2025-04-07
> >
> > Thank you to the authors for their response. I will maintain my score

---

### Official Review · Reviewer_bar9 · 2025-03-14

**Overall Recommendation:** 4

**Summary:**

In this paper, the author proposes two novel algorithms for solving a class of non-monotone equations, building upon the Forward-Reflected Backward Splitting framework and incorporating variance reduction techniques such as SVRG and SAGA. The proposed methods are accompanied by rigorous convergence guarantees and achieve state-of-the-art complexity bounds. To validate the theoretical results, the authors conduct a series of numerical experiments demonstrating the practical effectiveness of the algorithms.

**Claims And Evidence:**

The claims in the submission are well-supported by rigorous theoretical analysis and empirical validation, with no evident inconsistencies.

**Essential References Not Discussed:**

No, all essential related works appear to be appropriately cited and discussed.

**Experimental Designs Or Analyses:**

I find the experiments to be well-designed and informative, with no apparent issues in their methodology or analysis.

**Methods And Evaluation Criteria:**

Yes, the proposed methods and evaluation criteria are appropriate and well-aligned with the problem and application context.

**Other Comments Or Suggestions:**

On lines 140–142 of the second column, please ensure that the citation of “Diakonikolas et al.” is formatted correctly in accordance with the appropriate referencing style.

**Other Strengths And Weaknesses:**

The manuscript is well-composed, and the results are presented in a clear and coherent manner.

**Questions For Authors:**

I do not have any questions to raise at this point.

**Relation To Broader Scientific Literature:**

The proposed approach and novel algorithms can be helpful for optimization community to create new algorithms or can be useful for practical tasks.

**Theoretical Claims:**

Yes, I reviewed the proofs and identified issues related to the assumptions and justification of some theoretical claims.

---

> ### Author Rebuttal · Authors · 2025-03-25
>
> First of all, we highly acknowledge the reviewer for comments and feedback. Below is our response to each point.
>
> C1: In this paper, the author proposes two novel algorithms for solving a class of non-monotone equations, building upon the Forward-Reflected Backward Splitting framework and incorporating variance reduction techniques such as SVRG and SAGA. The proposed methods are accompanied by rigorous convergence guarantees and achieve state-of-the-art complexity bounds. To validate the theoretical results, the authors conduct a series of numerical experiments demonstrating the practical effectiveness of the algorithms.
>
> >R1: Thank you for a great summary of our work and contribution.
> Unlike existing variance-reduction methods in the literature which primarily address co-coercive or monotone problems, we instead consider a class of problems satisfying a weak-Minty condition, which is possibly non-monotone. Moreover, our methods are new and significantly different from the existing ones due to the use of $S^k$. The complexity $\mathcal{O}(n + n^{2/3}/\epsilon)$ is the best-known (but not optimal) for SVRG and SAGA in nonconvex optimization (without using enhancement tricks, like researt or nested loops).
>
> C2: On lines 140–142 of the second column, please ensure that the citation of “Diakonikolas et al.” is formatted correctly in accordance with the appropriate referencing style.
>
> >R1: Thank you. We will correct the issue you identified.

---

### Decision · Program_Chairs · 2025-05-01

**Decision:**

Accept (poster)

**Comment:**

For monotone inclusion problems, the paper develops novel stochastic variance-reduction methods, which combine the ideas from the forward-reflected-backward splitting method and unbiased variance-reduced estimators. By proposing a new definition for variance-reduced estimators, a quite generic theoretical analysis is delivered. From the reviews, the reviewers reach a consensus that the paper contains interesting result and novel contribution to the community. In the meantime, the paper contain the following issues
 - Paper presentation. The reviewers raised several points to improve the readability of the paper.
 - Weak numeric experiments, and lacks comparison against some existing methods.

The authors are encouraged to modify the paper accordingly.